# A coastally improved global dataset of wet tropospheric corrections for satellite altimetry

Clara Lázaro[1,2], Maria Joana Fernandes[1,2], Telmo Vieira[1,2], Eliana Vieira[1]

[1]Faculdade de Ciências, Universidade do Porto, 4169-007 Porto, Portugal

[2] Centro Interdisciplinar de Investigação Marinha e Ambiental (CIIMAR/CIMAR), Universidade do Porto, 4450-208 Matosinhos, Portugal

*Correspondence to*: Clara Lázaro (clazaro@fc.up.pt)

**Abstract.** The accuracy of satellite radar altimetry (RA) is known to deteriorate towards the coastal regions due to several reasons, amongst which the improper account for the wet path delay (WPD) can be pointed out. The most accurate WPDs for

RA are derived from the on-board microwave radiometer (MWR) radiance measurements, acquired simultaneously as the altimeter ranges. In the coastal zone, however, the signal coming from the surrounding land contaminates these measurements and the water vapour retrieval from the MWR fails. As meteorological models do not handle coastal atmospheric variability correctly yet, the altimeter measurements are rejected whenever MWR observations are absent or invalid. The need to solve this RA issue in the coastal zone, simultaneously responding to the growing demand of data in these regions, motivated the

development of the Global Navigation Satellite Systems (GNSS) derived Path Delay (GPD) algorithm.

The GPD combines WPD from several sources through objective analysis (OA) to estimate the WPD or the corresponding RA correction accounting for this effect, the wet tropospheric correction (WTC), for all along-track altimeter points for which this correction has been set as invalid or is not defined. The current GPD version (GPD Plus, GPD+) uses as data sources WPD from coastal and island GNSS stations, from satellites carrying microwave radiometers, and from valid on-board MWR

measurements. The GPD+ has been tuned to be applied to all, past and operational, RA missions, with or without an on-board MWR. The long-term stability of the WTC dataset is ensured by its inter-calibration with respect to the Special Sensor Microwave Imager (SSM/I) and SSMI/I Sounder (SSM/IS). The dataset is available for TOPEX/Poseidon (T/P), Jason-1 and Jason-2 (NASA/CNES), Jason-3 (NASA/EUMETSAT), ERS-1, ERS-2, Envisat and CryoSat-2 (ESA), SARAL/AltiKa (ISRO/CNES) and GFO (U.S. Navy) RA missions. The GPD+ WTC for Sentinel-3 (ESA/EUMETSAT) shall be released soon.

The present paper describes the GPD+ database and its assessment through statistical analyses of Sea Level Anomaly (SLA) datasets, calculated either with GPD+, ECMWF ReAnalysis Interim (ERA Interim) model or MWR-derived WTCs. Global results, as well as results for three regions (North American and European coasts and Indonesia region), are presented for the recent ESA's Envisat Full Mission Reprocessing (FMR) V3.0. Global results show that the GPD+ WTC leads to a reduction in the SLA variance of 1-2 cm$^2$ in the coastal zones, when used instead of the ERA WTC, which is one of the WTC available

in these products and can be adopted when the MWR-derived WTC is absent/invalid. The improvement of the GPD+ WTC over the ERA WTC is maximum over the tropical oceans, particularly in the Pacific Ocean, showing that the model-derived

WTC is not able to capture the full variability of the WPD field yet. The statistical assessment of the GPD+ for the North American coast shows a reduction in SLA variance, when compared to the use of the ERA-derived WTC, of 1.2 cm$^2$, on average, for the whole range of distances from coast considered (0-200 km). Similar results are obtained for the European coasts. For the Indonesia region, the use of GPD+ WTC instead of that from ERA leads to an improvement, on average, of the order of 2.2 cm$^2$ for distances from coast up to 100 km. Similar results have been obtained for the remaining missions, particularly for those from ESA. Additionally, GPD+ recovers the WTC for a significant number of along-track altimeter points with missing or invalid MWR-derived WTCs, due to land, rain and ice contamination and instrument malfunctioning, which otherwise would be rejected. Consequently, GPD+ database has been chosen as the reference WTC in the Sea Level Climate Change Initiative (CCI) products; the GPD+ has also been adopted as reference in CryoSat-2 Level 2 Geophysical Ocean Products (GOP). Strategies to further improve the methodology, therefore enhancing the quality of the database, are also discussed. The GPD+ dataset is archived on the homepage of the Satellite Altimetry Group, University of Porto, publicly available at the repository https://doi.org/10.23831/FCUP_UPORTO_GPDPlus_v1.0 (Fernandes et al., 2019).

## 1 Introduction

Since the early 1990s, satellite radar altimetry (RA) missions have been observing the oceans, measuring global and regional mean sea level, as well as its change. Altimeters on board RA missions measure the sea surface height (SSH) by subtracting the measured altimeter range, the nadir-measured distance between the satellite and the sea surface, from the satellite altitude (H) above a reference ellipsoid of a terrestrial (geocentric) reference frame, currently known with a centimetre-level radial error (Rudenko et al., 2017). In the computation of accurate SSH, a multitude of well understood effects must be properly considered: those that introduce errors in the measured range, e.g. atmosphere propagation delay, and those that induce SSH variability other than that under analysis over time, e.g. ocean tides and atmospheric pressure. Sea surface height anomalies, or sea level anomalies (SLA), are computed subtracting a mean sea surface (MSS) from the corrected SSH measurements. Range corrections are required to account for the delay the microwave pulses suffer, as they propagate through the atmosphere (ionospheric and tropospheric corrections, the latter including the effect of the neutral atmosphere) and for the interaction with the sea surface (sea state bias); geophysical corrections account for the sea level variability due to tides (ocean, solid earth and polar tides, as well as loading effects) and for the ocean's response to atmospheric pressure (dynamic atmospheric correction, a combination of a high-frequency signal with the low-frequency inverted barometric response of the ocean); if needed to homogenize and inter-calibrate multi-mission data, the reference frame offset correction is applied, accounting for instrument-dependent effects and biases between missions (Fernandes et al., 2014). A detailed description of the corrections is given in Chelton et al. (2001) and Escudier et al. (2017).

This may be expressed as:

$$SLA = H - R_{corr} - MSS \tag{1}$$

where $R_{corr}$ is the altimeter range ($R$) corrected for all instrument ($\Delta R_{inst}$), range ($\Delta R_{range}$) and geophysical ($\Delta R_{geoph}$) effects:

$$R_{corr} = R + \sum(\Delta R_{range} + \Delta R_{geoph} + \Delta R_{inst}) \tag{2}$$

The quality of the SLA measurements has considerably improved over time, essentially because new models and corrections have become available, and satellite orbit determination as well as radar processing have improved in absolute accuracy. This is particularly true over open ocean, where altimeter waveforms do not depart from the expected shape given by the Brown model and geophysical and range corrections can be accurately estimated (Chelton et al., 2001; Escudier et al., 2017; Fernandes et al., 2006).

The total tropospheric path delay for microwaves can be divided into two components, one depending only on the surface pressure, the hydrostatic term, and a remainder that mainly depends on water vapour abundance, commonly termed wet path delay (WPD) (Askne and Nordius, 1987). The dry tropospheric correction (DTC) accounts for the hydrostatic term that, despite having an absolute value as large as 2.3±0.2 m in the zenith direction at sea level, over the ocean it can be calculated with millimetre-accuracy, provided the sea level atmospheric pressure is known at each location (Fernandes et al., 2014). From here

onwards, the terms DTC and WTC are used to refer to the dry and wet tropospheric corrections (negative values), respectively, applied to RA measurements and, accordingly, DPD and WPD to the corresponding absolute values. The DTC computation can be carried out using sea level pressure fields given by numerical weather models (NWMs), as described e.g. in Fernandes et al. (2013a). Ranges are corrected for the wet path delay through the wet tropospheric correction (WTC), possessing an absolute value less than 0.50 m (Chelton et al., 2001). In opposite to the estimation of the DTC, the WTC retrieval requires

the knowledge of the full water vapour and temperature profiles, which are known to be highly variable, both temporally and spatially (Dousa and Elias, 2014; Vieira et al., 2019a). Therefore, to properly account for the microwave propagation delay through the troposphere, RA missions carry aboard a passive microwave radiometer (MWR), nadir-looking instruments capable of measuring both the water vapour and the cloud liquid water components of the wet path delay, from brightness temperatures in appropriate bands of the microwave spectrum.

Radiometers embarked on RA missions can be divided into two main groups (Steunou et al., 2014). Two-channel MWR, operating at frequencies 21–23.8 GHz, the primary water vapour sensing channel, and 34–37 GHz, carried by the European Space Agency (ESA) ERS-1, ERS-2 and Envisat, by ESA/EUMETSAT Sentinel-3, by US Navy's mission Geosat Follow-On (GFO) and by the joint Indian Space Research Organization (ISRO) and Centre National d'Études Spatiales (CNES) SARAL (Satellite with ARgos and ALtiKa) missions; three-channel MWR carried by NASA's missions TOPEX/Poseidon (T/P), Jason-

1, Jason-2 and Jason-3, with an additional channel operating at 18–18.7 GHz. MWR footprints vary in the range of from 20 to 45 km, depending on the instrument and frequency except for the one embedded within SARAL's altimeter, for which the dual frequency radiometer has a footprint of diameter less than 12 km (Steunou et al., 2014).

Accurate measurements of the integrated amount of water vapour and cloud liquid in the atmosphere are achievable in open ocean, but difficult to perform in coastal regions where the precise estimation of the WTC is still challenging. Nearly a decade

ago, the RA community started developing new algorithms and methodologies aiming at recovering altimetric data in the

coastal region, leading to a more mature status of the emerging, at that time, field of coastal altimetry. Altimetric data in the closest 50 km away from the coast are usually flagged as invalid, being therefore discarded, or non-existent due to several reasons. On the one hand, the shape of the waveforms no longer can be described by the Brown model and this is overcome using specific retracking techniques; on the other hand, the accurate modelling of some corrections is difficult. This is

particularly true for the estimation of the wet path delay, and consequently of the WTC, since in coastal areas the measurements of the MWR are in general contaminated by land, in part due to the large diameter of its footprint. Also important, is the fact that the WTC retrieval algorithms are designed for open-ocean conditions, thus assuming surface emissivity values corresponding to open-ocean conditions; however, surface emissivity can be highly variable when the surrounding land surfaces contribute partially to the returning signal, causing a failure of the retrieval algorithms. Different strategies have been

proposed in the last years to accomplish the estimation of the wet tropospheric correction in coastal areas, which are summarized in Cipollini et al. (2017). One of these is the GNSS (Global Navigation Satellite System) derived Path Delay (GPD) algorithm. The GPD was developed by the University of Porto (UPorto) in the scope of the ESA's funded project COASTALT (Vignudelli et al., 2009) to estimate the WTC for correcting the altimetric data in the coastal European region. It has evolved over the last years reaching a mature status recently (GPD Plus, GPD+), with the global computation of a WTC

dataset for all operational and past RA missions that has been adopted as reference to derive the ESA Climate Change Initiative Sea Level dataset (Quartly et al., 2017, Legeais et al., 2018).

With this article, it is aimed to inform GPD+'s current and potential new users about the content and the services that the GPD+ database provides. The methodology, the input data sources and the supplied GPD+ products are presented and described. The WTCs provided in the GPD+ products have been assessed through various SLA variance statistical analyses,

inspecting simultaneously the impact of the correction on sea level variability. Results are provided for the recently reprocessed and released Envisat Geophysical Data Records (GDR) V3.0 dataset, both globally and for three selected regions (North American and European coasts and Indonesia region), aiming at showing the improvement in the description of the SLA field when the GPD+ WTC is adopted instead of the MWR- or ERA-derived WTCs provided in the Envisat GDR V3.0. A summary of the results for the remaining satellite altimetry missions is also provided. For more details concerning these results, readers

may refer to Fernandes et al. (2015) and Fernandes and Lázaro (2016, 2018). To ensure the long-term stability of the GPD+ WTC, an important issue when trends in sea level change are calculated, the large set of radiometers used in this study have been previously inter-calibrated through the inter-comparison of the various datasets. The calibration parameters of this analysis are presented for all satellite altimetry missions. Additionally, strategies to further improve the methodology, aiming at enhancing the quality of the GPD+ products publicly available in the database, are shared.

This paper is organised in five sections. The input data are described in Sect. 2.1, the technical description of the algorithm is presented in Sect. 2.2, and the generated WTC database in Sect. 2.3. Section 3 describes the results obtained globally and for three zoomed-in regions, selected to show the performance of the database in coastal regions, and includes their discussion. Section 4 describes the availability of the GPD+ products. The conclusions are provided in Section 5.

## 2 The GPD+ algorithm and the GPD+ WTC database

The GPD+ algorithm has been developed to estimate the WTC over coastal regions, where MWR-derived WPDs, if available, are usually anomalous values due to land contamination both in the altimeter and MWR observations. If uncorrected, this leads to a rejection of a large number of points in these regions. To accomplish this task, WPD datasets from different sources are combined through an optimal interpolation scheme.

### 2.1 Input WPD observations

In the most recent version of the algorithm (GPD+), WPDs from the following sources are used as input: 1) tropospheric zenith total delays (ZTDs) computed at a dense GNSS network of stations distributed globally along the coastline; 2) total column water vapour (TCWV) products generated from measurements from passive imaging MWR on board environmental and meteorological Earth observation satellites; 3) along-track WPD (the absolute value of WTC) measurements from the on-board MWR, before they become invalid when approaching the coast. The algorithm also provides valid WTC estimates for offshore and open-ocean measurements for which invalid WTC are detected, provided WPD observations are available at those geographical locations. In this way, the algorithm attempts to eliminate measurements contaminated by heavy rain and ice, as well as faulty measurements due to, e.g., instrument malfunctioning.

#### 2.1.1 WPD from GNSS-derived ZTD

Tropospheric propagation delays are a source of error in GNSS positioning as well, being therefore estimated, at each GNSS station, for each observation. The quantity computed at each station is the slant total delay (STD) between the satellite and the ground-based station. Provided *a priori* value for the zenith hydrostatic delay (ZHD or DTC in satellite altimetry terminology), computed from knowledge of surface atmospheric pressure either measured locally or NWM-derived meteorological data, and mapping functions for hydrostatic and wet components are known, the ZTD at station height can be computed with millimetre accuracy (Pany et al., 2001; Fernandes et al., 2013a, 2015). Mapping functions relate the delay in the station zenith direction, ZTD, with that in the actual satellite-station direction, STD. While the wet delay varies in time in an unpredictable way, the ZHD can be derived with millimetre accuracy from e.g. NWMs (Pany et al., 2001). Therefore, an *a posteriori* more accurate ZHD can be computed and subtracted from the estimated ZTD, yielding the wet delay in the zenith direction (zenith wet delay, ZWD or WPD in satellite altimetry terminology). ZHDs, computed with millimetre accuracy at station height from ZHDs at sea level derived from sea level pressure (SLP) fields from an NWM (e.g. European Centre for Medium-Range Weather Forecasts ReAnalysis (ERA) Interim or ECMWF operational models) and further reduced to station height using an adequate height reduction procedure, are used to derive WPD from GNSS (Fernandes et al., 2013a, 2015). The WPDs obtained this way are given at station height and therefore at a level different from that of interest in satellite altimetry, which is the mean sea level. Therefore, the height reduction of the WPD is required. This has been performed using an exponential decay function, empirically obtained by Kouba (2008), valid for WPD height reductions for heights below ~1000 m (Vieira et al., 2019b).

This summarises the methodology adopted by UPorto in the computation of WPD from GNSS measurements. A complete description of the methodology and its assessment can be found in Fernandes et al. (2013a, 2015) or Vieira et al. (2019b). Zenith total delays (ZTD) estimated at UPorto, along with those available online from international GNSS services (IGS (International GNSS Service), EPN (EUREF Permanent Network) and SuomiNet) and from several stations located at the German Bight, provided to UPorto by the Technische Universität Darmstadt (TUD) in the scope of ESA's Climate Change Initiative (CCI) project, have been used. More than 800 coastal (at distances from the coast less than 100 km) and offshore GNSS stations, with altitude below 1000 m, are being used at the time of writing. Figure 1 shows the increase both in the number of GNSS stations and GNSS observations used as input in the GPD+ algorithm. The number of stations almost duplicates, in 2008.5, relatively to the number of stations in the beginning of the period and have been continuously increasing until present. Figure 2 shows the location of the coastal and island GNSS stations used as input in the GPD+ and the standard deviation (SD) of the WTC field from ERA Interim for the along-track point of Envisat cycles 96-108 (11/2010-11/2011). WTC SD ranges from 1–15 cm and has a strong dependency with latitude. Maximum values of WTC SD can be seen in the tropical southern and eastern Asia, in the north of Australia, and around Mexico and southwest USA, due to variable precipitation determined by the monsoon regime (Vieira et al., 2019a). Over the polar regions, minimum WPD SD values are found (values < 3 cm).

### 2.1.2 WPD from scanning imaging MWR

The methodology developed by UPorto to calculate the WTC from TCWV products from passive imaging MWR on board Earth observation satellites is discussed in detail in Fernandes et al. (2013b, 2015). Due to their large spatial and temporal resolutions and spatial coverage, SI-MWR data increase the number of observations to use as input in GPD+ over the ocean, thus allowing the recovery of e.g. entire tracks for which the MWR-derive WTCs are missing due to instrument malfunctioning (i.e. where MWR- and GNSS-derived observations are not available). For this reason, their use improves the description of the WPD field. Additionally, these data are of extreme importance since they provide the unique possibility of computing the WTC over open ocean for those RA missions that do not possess an MWR, like e.g. CryoSat-2 (CS-2). In fact, GPD+ is an upgrade from the GPD methodology, which was developed to compute the WTC only for coastal points, relying only on GNSS and valid on-board MWR measurements. Motivated by the need to compute an improved correction for CS-2, the SI-MWR data set was included and the focus of the correction extended to open ocean.

TCWV datasets from 20 scanning imaging (SI) passive MWR (SI-MWR), available at NOAA Comprehensive Large Array-Data Stewardship System (CLASS) and at Remote Sensing Systems (RSS) have been selected. CLASS includes data from the AMSU-A (Advanced Microwave Sounding Unit-A) on board NOAA-16, -17, -18, -19, MetOp-A and MetOp-B satellites. RSS delivers datasets for several sensors, namely SSM/I (Special Sensor Microwave Imager) and SSM/IS (SSM/I Sounder) on board DMSP (Defense Meteorological Satellite Program) satellites (F08, F10, F11, F13, F14, F16, F17 and F18), WindSat aboard Coriolis, Tropical Rainfall Measuring Mission's (TRMM) radiometer TMI (TRMM Microwave Imager), Global Precipitation Measurement's (GPM) Microwave Imager (GMI), AMSR-E (Advanced Microwave Scanning Radiometer for

EOS) on board AQUA and AMSR-2 in the Japanese Global Change Observation Mission – Water Satellite 1 (GCOM-W1).

Two types of TCWV products have been used: Level-2 swath products in HDF-EOS2 format (near real time products, 14-15 orbital swaths per day available for each instrument) from all data sources except RSS, and Level-2 gridded products (two grids per day, each containing the ascending/descending passes) in binary format from RSS. Table 1 shows the availability of the TCWV products used as input in GPD+ and their main characteristics (spatial and temporal resolution and availability). Figure 3 shows the number of SI-MWR along time for each RA mission. For the Envisat mission, for example, the number of

SI-MWR increased from 4 to 11, from the beginning (05/2002) to the end (03/2012) of the mission, respectively.

The calculation of the path delay from TCWV can be performed knowing that the quotient between WPD and TCWV is modelled by a decreasing function of WPD of the type

$$\frac{WPD}{TCWV} = a_0 + a_1 TCWV + a_2 TCWV^2 + a_3 TCWV^3 \tag{3}$$

with constants $a_0 = 6.8544$, $a_1 = -0.4377$, $a_2 = 0.0714$, and $a_3 = -0.0038$, for TCWV in the right-hand side of the equation in

centimetres (Stum et al., 2011).

It is known that, in addition to TCWV, WPD also depends on temperature. Expressions such as Eq. (3) account for an implicit modelling of this dependence. Fernandes et al. (2013b) have shown that this expression leads to similar results as those obtained by adopting formulae that make use of explicit values of atmospheric temperature given e.g. by an NWM, as the one adopted by Bevis et al. (1994). The authors show that after sensor inter-calibration, a crucial step to guarantee datasets consistency, the

WTC derived from both methods are equivalent, with differences within ± 2 mm.

### 2.1.3 WPD from along-track MWR

The provenience of the MWR-derived WTC used as input in the GPD+ is the Radar Altimeter Database System (RADS) (Scharroo et al., 2012), except for Envisat, as this mission has been recently reprocessed, and SARAL before cycle 30 (for cycles 1 to 30, the MWR-derived WTC provided in the products from the Prototype for Expertise on AltiKa for Coastal,

Hydrology and Ice (PEACHI) project (Valladeau et al., 2015), available through AVISO+, have been used). It is recalled that the WPD is the absolute value of the WTC, the quantity of interest in satellite altimetry. RA data necessary to compute the SLA datasets used to validate the GPD+ WTC are also extracted from RADS. For each RA mission, only valid MWR-derived WTC are input in the algorithm, therefore the correct identification of valid/invalid measurements is of crucial importance. Exception made for CryoSat-2 (for which, in the absence of an on-board MWR, a WTC is generated for all along-track

altimeter points), GPD+ estimates a WTC for those points with an invalid MWR-derived WTC only. In this way, the valid observations from the on-board MWR are preserved.

Invalid measurements are usually detected using a set of flags, some of them provided in the products, as the radiometer flag for the surface type or the ice flag. If different from 0, these flags indicate invalidity due to land contamination or instrument malfunctioning, or ice, respectively. MWR-derived WTCs outside the range -0.5 m ≤ WTC < 0.0 m, generally due to heavy

rain or ice, are also discarded. A validity criterion based on the distance from coast is also applied: if the location of a certain

MWR measurement is such that its distance from the coast is less than a threshold value, then this measurement is most certainly contaminated by land. Threshold values used in this criterion depend on the RA mission. Adopted values are based on the known characteristics of each instrument and on an independent assessment of the on-board MWR observations using GNSS-derived WPDs in the coastal zones (Vieira et al., 2019b). Results for ESA missions are alike, showing that land contamination occurs at distances from coast less than 30 km; the same threshold has been used for GFO and T/P. In relation to the remaining NASA missions, values of 15 km have been used for Jason-1/2/3. For SARAL, a threshold value of 15 km was adopted. Also, noisy MWR measurements are discriminated using median filters based on statistical analysis of the differences to the NWM-derived WTC on the same along-track point and neighbouring points. Invalid measurements are detected if: 1) radiometer surface type flag is different from 0; 2) ice flag is different from 0; 3) do not satisfy the defined statistical criteria or are outside WTC limits, 4) are at a distance from coast less than the threshold established for that mission. Figure 4 shows all the along-track points flagged as invalid for Envisat cycle 12, which reach 29.5%. As it will be shown in Sect. 3 for Envisat, per cycle and on average, approximately 30% of the oceanic points have an invalid WTC value; for these points, an SLA value cannot be computed due to the invalidity of the WTC or of other corrections, or because certain criterion is not met (e.g., number of 18 Hz measurements to compute the 1 Hz values used less than the imposed minimum). For approximately 10% of all oceanic points (including the coastal zone), the WTC is the only correction that prevents the computation of the SLA. This is, on average, the percentage of points with a valid SLA value recovered by the GPD+ algorithm for a mission such as Envisat. For other missions, this percentage depends on instrument type, band of latitudes covered by the mission (which determines the amount of ice contamination) and instrument performance, and is summarised in the conclusions.

### 2.1.4 Radiometer Calibration

Global mean sea level is a valuable proxy to understand climate change and how it operates, since it includes the response from various components of the climate system. Also important in the analysis of trends in sea level change, which requires a 0.3 mm/yr error level set by the Global Ocean Observing System (GOOS), is the stability of the altimetry dataset. Therefore, the examination, and consequent accounting for, of drifts in the corrections, particularly in the WTC, is necessary to ensure that the corrections are stable in time and do not introduce spurious trends in the SLA.

To ensure long-term stability of the GPD+ WTC, the large set of radiometers used in this study have been previously inter-calibrated through the inter-comparison of the various datasets. Data from the reference missions for sea level investigations, such as the T/P and Jason series (and soon Sentinel-6), have been calibrated against those of the Special Sensor Microwave Imager (SSM/I) and the SSM/I Sounder (SSM/IS) by selecting matching points from each pair of missions operating simultaneously with a difference in time and location less than 45 minutes and 50 km, respectively (Fernandes et al., 2013b). The time-series of these matching points was used with a 3-parameter model to adjust offset (*a*), scale factor (*b*) and linear trend (*c*) for each mission (Fernandes and Lázaro, 2016):

$$Y = a + bX + c(T - T_0), \ \ T_0 = 1992 \tag{4}$$

The remaining altimetry missions were then inter-calibrated to these calibrated datasets from the reference missions since orbits of most all remaining missions are sun-synchronous with different times for the Equator crossing than those of the SSM/I(S), with a small number of matchups mostly found at high latitudes, not representative of the WTC variability. For these missions, data were analysed at crossover points and the same adjustment parameters were obtained from the time-series. For the crossover analysis, only data with difference in time less than 180 minutes were used. As an example, the calibration parameters have been obtained for Envisat are $a$=-6.82 mm; $b$=0.991 and $c$=-0.0028 mm/yr, showing that the trend is negligible and indicating that the dataset is well aligned with the altimeter reference missions and with SSM/I and SSM/IS. The small offset and scale factor have the impact of making the correction more negative by 6-7 mm. The calibration parameters (offset, scale factor and linear trend) and their formal errors, obtained for all satellite altimetry missions with an on-board MWR available in the GPD+ database, are presented in Table 2. For more details concerning the calibration of the radiometers, the readers are advised to see Fernandes and Lázaro (2016).

### 2.1.5 WPD from NWM

Space-time collocated WTCs from NWM grids are adopted in the OA as first guess. Usually two models from ECMWF are used: ERA model, provided each 6 hours with $0.75° \times 0.75°$ spatial resolution, used for missions prior to 2004, and ECMWF Operational Model (ECMWF Op., 6-hour time interval, $0.125° \times 0.125°$ spatial resolution) for missions after this period. Since the ECMWF Op. has undergone several updates, not having the same accuracy over time, for all missions with data before 2004 (T/P, Jason-1, ERS-1, ERS-2, Envisat and GFO) ERA Interim is used in GPD+, while for the most recent missions the ECMWF Op. model is adopted. Therefore, in the absence of observations to improve the first guess, a WTC estimate from ERA Interim or ECMWF Op. is output from GPD+. This is normally the case for the northernmost latitudes. In addition, to reduce data discontinuities, output values solely based on model data are adjusted to the valid MWR measurements of each cycle by solving for the mean difference, of the order of a few millimetres, between the two datasets for all points with a valid MWR-derived WTC.

### 2.2 Algorithm description

The GPD+ algorithm is based on objective analysis and estimates the wet path delay, given measurements from different sources of the variable under study at a restricted number of data points. The statistics of the field are estimated in the form of a correlation function and of the measurement errors associated with each type of observation. The expected error associated to this estimate is also derived. The technique for the objective analysis is fully described in Bretherton et al. (1976).

The algorithm has been originally implemented to calculate the WTC in the coastal zone, where the retrieval of the wet path delay from on-board MWR measurements become invalid. Later, it has evolved to provide the correction also over open ocean, providing the correction during, for example, instrument malfunctioning, and inland waters.

For the altimetry missions carrying an on-board MWR (all but CryoSat-2), a GPD+ WTC estimate is calculated for all along-track points with an MWR-derived WTC deemed as invalid, using valid WTC observations from different sources at the nearby location and within a time interval, defined by the spatial and temporal radiuses of influence used in the computation. In the current GPD+ version, these radiuses have been set equal to the correlation spatial and temporal scales. Whilst the spatial correlation scale varies spatially, both with longitude and latitude (Fernandes and Lázaro, 2016), the temporal correlation scale

has been set to 100 minutes (Bosser et al., 2007). For the CryoSat-2 mission, since it does not carry a passive microwave radiometer, a GPD+ WTC estimate is computed for every along-track point using third-party data (WTC observations, other than those from the on-board MWRs) only. The location and time of each along-track are those provided in the GDR products present in RADS. Due to the temporal difference between adjacent satellite tracks, in practice only along-track valid on-board MWR measurements from the track to which the point of estimation belongs are used.

Regarding the accuracy of the observations, a constant value of 0.5 cm has been set for the white noise of the GNSS- and MWR-derived wet path delays, while for the SI-MWR observations a value between 0.7 cm and 1.1 cm, depending on the mission, has been used (Fernandes et al., 2013b).

The procedure for finding a good estimate of the WTC starts with the definition of the first guess or *a priori* value for the field. In the current version of the algorithm, the first guess is the space-time collocated NWM-derived wet path delay from ERA

Interim or ECMWF-Op, the most suitable depending on purpose and time period. Therefore, in the absence of observations, the GPD+ WTC equals the NWM-derived WTC. In the presence of observations, its input number is limited to 15 in order to decrease computational burden; the chosen observations are those for which the statistical weights are larger, meaning that for these measurements the differences in acquisition time and distance to the point where the estimate is being calculated are the smallest.

The estimates for those missions that embark an MWR rely on the valid MWR-derived WPD values. Therefore, one of the core competencies of the GPD+ methodology is its ability to detect corrupted WTC values, which is achieved through the definition of improved criteria for their detection. Measurements flagged as invalid are those that: - have the radiometer surface type flag set as 1; - are contaminated by ice; - are contaminated by rain; - are outside the range [-0.5 m, 0.0 m]; have mission-dependent flags (e.g., radiometer along-track averaging flag for Envisat) set as 1; - do not satisfy several statistical criteria

based on the differences between adjacent measurements and between MWR and NWM values; - are at distances from coast less than 15 or 30 km, depending on being a reference and SARAL or ESA mission.

A general Gaussian space-time correlation function of the form

$$G(r, \Delta t) = e^{-\frac{r^2}{C^2}} \cdot e^{-\frac{\Delta t^2}{T^2}}$$ (5)

where $r$ and $\Delta t$ represent the distance and the time interval between acquisitions of each pair of points, and $C$ and $T$ are the spatial and temporal correlation scales, respectively, has been adopted to account for the spatial and temporal variability. A diagram showing the workflow of the GPD+ algorithm is shown in Fig. 5.

## 2.3 GPD+ WTC files description and nomenclature

As the impact of the correction is mainly in ocean studies, in the current version, the final GPD+ WTCs are continuous products over the ocean and coastal regions. To prevent the loss of points when interpolating to 20 Hz points, in addition to ocean points, the closest point over land is included, provided it is within a distance less than 50 km from the ocean. This guarantees that observations over ocean necessary to compute the WTC for this location are still available within the radiuses of influence centred on the point. The WTC estimated for the closest points over land are also estimated at sea level. For Envisat, as this mission has been recently reprocessed (Version 3.0), the GPD+ WTC covers the whole range of latitudes and surfaces, including land. Corrections are currently publicly available for ten RA missions: T/P, Jason-1, Jason-2, Jason-3, GFO, ERS-1, ERS-2, Envisat, SARAL and CryoSat-2. Figure 6 gives an example of the GPD+ WTC for Envisat's cycle 12, showing global coverage (top panel) and over ocean regions with valid sea level anomaly values (bottom panel). As stated above, the correction has its main impact over the ocean since it is meant to be used to improve satellite altimetry. Over non-oceanic surfaces, the correction has been set equal to the ECMWF ERA Interim or Operational models, depending on the mission, as previously explained (Sect. 2.1.5). This has already been done for Envisat, therefore ensuring that Envisat GPD+ products are continuous over ocean and non-ocean surfaces. Future versions of the GPD+ correction for the remaining missions will cover all surface types as well. In addition, it is envisaged to improve the GPD+ methodology so that GPD+ WTCs will be estimated over non-oceanic regions, provided WPD observations exist (e.g. from MWR over large lakes or from GNSS). The GPD+ WTC products, which content is described in Table 3, are provided for each cycle of the mentioned altimetric missions. For the time and location of each altimeter measurement, specified by the variables 'time_01' in UTC seconds since 2000-01-01 00:00:00.0 and 'geodetic lat_01' and 'lon_01' in degrees as given in each GDR file, the GPD+ wet tropospheric correction, in metres, and its associated validity flag, fields 'GPD_wet_tropo_cor_01' and 'GPD_wet_tropo_cor_qual_01' respectively, are provided at 1 Hz. The sign convention adopted is that the WTC should be added to the range measured by the altimeter to correct it for the range delay. The data-quality flag can take the following values:

- 0: the MWR-derived WTC is valid and, in this case, the GPD+ correction is equal to the MWR-derived WTC, after applying calibration factors, therefore preserving the high accuracy of these data;
- 1: the invalid MWR-derived WTC has been replaced by a valid GPD+ estimate based on observations;
- 2: no observations were available for the computation and the GPD+ estimate is the first guess (i.e., ERA Interim for TOPEX/Poseidon, ERS-1, ERS-2, Envisat, Jason-1 or ECMWF Op. for OSTM/Jason-2, Jason-3, Cryosat-2, SARAL/AltiKa) with possible small bias applied.

- 3: GPD + estimate is outside the valid range ([-0.5, 0.0]), and either the value -0.5 or 0.0 was attributed to the output value (in the most recent implementation this never occurs, as these are replaced by the NWM values).

By using this flag, a knowledgeable user can select the data most suitable for a given application: a continuous correction e.g. for coastal studies, solely the valid measurements for the on-board MWR (e.g. for calibration purposes or global climate studies) or exclude the points solely based on NWM values.

NetCDF files include self-documenting variables and common attributes.

The nomenclature selected for the GPD+ dataset is:

< MISSION>_c<CYCLE_NUMBER>_gpd.nc

where <MISSION> is two-letter code that depends on the mission (see Table 4) and <CYCLE_NUMBER> is a three-digit number indicating the cycle number of <MISSION>. In all cases, the RADS cycle number convention has been adopted. In cases such as Jason-1 geodetic phase (phase c), cycle numbers are different from those adopted by AVISO. For CryoSat-2, sub-cycle numbers of 27 or 29 days are used according to RADS convention. The availability of GPD+ WTC for each mission is presented in Table 4 (Fernandes et al., 2019).

## 3    Results and Discussion

The results here provided have been obtained in the scope of several ESA-funded research projects and present new scientific findings that have not been published before. For Envisat, the GPD+ WTC was computed for inclusion in the newly reprocessed Envisat Geophysical Data Records (GDR) V3.0 in the ambit of the ESA second Envisat Altimetry Full Mission Reprocessing (FMR). Results concerning the remaining RA missions are summarised in the conclusions. For more details, the reader is advised to consult Fernandes and Lázaro (2018) for Sentinel-3, Fernandes and Lázaro (2016) for Cryosat-2 and GFO, and Fernandes et al. (2015) for T/P, Jason-1 and -2 and ESA missions, however the latter results were obtained with a previous version of GPD+, the so-called GPD algorithm.

### 3.1 GPD+ WTC for Envisat Mission

Results for Envisat cover the period May 2002 to April 2012, cycles 6 to 113, which corresponds to the whole Envisat FMR V3.0 dataset released in July 2018 (ESA, 2019). The GPD+ WTC is here compared with the ECMWF Reanalysis WTC (ERA Interim, field mod_wet_tropo_cor_reanalysis_01) and with the WTC derived from the on-board MWR (field 'rad_wet_tropo_cor_sst_gam_01'), both present in the FMR GDRs. The Envisat V3.0 reprocessed data have been improved, comparatively to the previous version, in many aspects, among which is an increased availability of the data acquired by the MWR, particularly at the beginning of the mission.

In the estimation process, the ERA Interim WTC was selected as first guess, being therefore the adopted values in the absence of measurements, as those occurring over land. Anomalies in this field have been found, with the field out of limits in a set of points, most of them concentrated on certain passes. This is due to the fact that this correction has been computed from 3D

model fields at the altimeter measurement altitude. Therefore, whenever the altimeter-derived surface height is not set (Not a Number value, NaN), the corresponding model-derived WTC will also be NaN. As our goal is to be able to provide continuous WTC, without data gaps, this field is unsuitable for use in the GPD+ estimations. For this reason, abnormal values present in

the products were replaced by those computed from ERA Interim single layer fields of TCWV and 2-metre temperature, with the formulation used by Fernandes and Lázaro (2016).

The MWR-based correction used in the generation of the GPD+ WTC products ('rad_wet_tropo_cor_sst_gam_01' GDR field) is hereafter called 'on-board MWR-derived WTC'. Figures 7 and 8 show the GPD+ WTC for some Envisat tracks, with different WTC variability conditions, exemplifying several issues commonly encountered in the on-board MWR-derived WTC

that no longer exist in the GPD+ WTC: unavailability of the correction (Fig. 7a); correction contaminated by ice (Fig. 7b and Fig. 8a, at latitudes above ±60°); existence of outliers (red points over open ocean at latitudes 30°S-40°S in Fig. 8a); and correction contaminated due to land proximity (red points around coastal regions in all panels except Fig. 7a). The improvement in the description of the WTC signal in terms of small spatial scales when compared to the ERA Interim WTC (in blue) is clear in the panel (a) of Fig. 7 (e.g. around latitudes 10°S-10°N). It is important to refer that the corrections are

shown only for points for which a valid SLA value can be computed after recovering the WTC, as explained in what follows. Figure 9 summarizes the results for the whole Envisat period (cycles 06 to 113). The percentage of points, for each Envisat cycle, with a rejected MWR-derived WTC, for which a GPD+ estimate has been computed are represented in pink and are seen to be around 30%. Figure 4 shows an example of the geographical location of these invalid MWR-derived WTCs for Envisat Cycle 12. For this cycle, the percentage of ocean points with invalid WTC is 29.5% and the corresponding number

when only points with valid SLA are selected is 10.9%. By way of example, for the same cycle, the percentage of points recovered due to land, ice and rain contamination, this latter also including outliers, is 8.9%, 17.4% and 3.2%, respectively. The corresponding percentage of points for which a valid SLA value could be computed after the estimation of the WTC by the GPD+ is shown in green. The number of points with valid SLA values (in grey) per cycle is also represented. This figure allows us to show that the GPD+ algorithm leads to the recovery of approximately 10% of the points with valid SLA value. In

some cycles this value can reach 20% or more, most of these points are located at high latitudes and in coastal regions. Keeping in mind that ESA missions are near-polar missions with an inclination of ~98.5°, they have the great advantage, when compared to the reference missions, of acquiring measurements at high latitudes. The recovery of data in these regions, besides along the coastal regions, can be considered one of the greatest advantages of the GPD+ methodology. The given figures show that for around 20% of the altimeter measurements, an SLA value could not be computed due to a reason other than the invalidity of

the WTC. This means that if in future FMR the issues that prevent the SLA computation are totally or partially solved, the percentage of data recovery will increase up to a maximum of 30% when the GPD+ WTC is used. Despite being provided continuously, the GPD+ WTC has its largest impact over ocean.

GNSS data cannot be considered independent from the GPD+ WTC since they have been used in their computation. Therefore, these data are not adequate to use in the GPD+ validation. However, the analysis of the root mean square (RMS) value of the

WTC differences, function of distance from coast, can be valuable to inspect the correction in coastal regions, where the

methodology is committed to ameliorate the WTC. For this assessment, GNSS-derived WTC have been computed at a network of 60 GNSS stations using the methodology explained in Vieira et al. (2019b). This network has a good geographical distribution and covers regions around the world with different atmospheric variability conditions. This data set consists of WTC measurements at each station location for the whole period of observations available for that station, allowing a non-

collocated comparison with WTC estimations at MWR points. Differences between these GNSS-derived WTC and the on-board MWR and the GPD+ WTC retrievals, respectively, have been computed and analysed for the whole Envisat mission.

For the acquisition instant of each MWR-derived WTC, a GNSS-derived WTC is interpolated, at the station location, for the same instant and is further reduced to sea level; at the same acquisition epoch and location of each MWR-derived WTC, the GPD+ WTC is also available, being the two collocated both in time and space and over ocean. For each pair of WTCs (MWR

and GNSS-derived WTCs and GPD+ and GNSS-derived WTCs, relative to the same instant), the distance from coast of each altimeter point is computed. This process is repeated for each GNSS station with surrounded altimetry measurements and then the whole set of stations is considered, to obtain representative results for the whole globe. Differences are binned into 5-km intervals and the RMS values computed function of distance from coast. The results are shown in Fig. 10, for distances up to 65 km from the coast, where red and grey bars represent the number of measurements used to compute the RMS of the

differences GNSS-MWR and GNSS-GPD+, respectively. The number of differences is not the same for each case, since the number of invalid MWR-derived WTCs increases as the tracks approach coast, being discarded from the analysis, while the same along-track points have valid WTC estimates from GPD+. For the comparison GNSS-GPD+, only WTC retrieved from the observations have been selected (i.e. those estimated from the model where discarded); for the comparison GNSS-MWR, valid MWR values and those that would be rejected solely based on the criteria of distance from coast were selected (otherwise

the invalid measurements due to e.g. ice or rain contamination would overestimate the results). Consequently, the number of GNSS-MWR differences is generally smaller than the number of GNSS-GPD+ differences.

The increase in the RMS value of the GNSS-MWR differences in the closest 25 km of the coast, seen in Fig. 10, is a clear indication of the loss of accuracy of MWR-derived WTCs in this coastal strip. This also shows that when all rejection criteria except the one related with the distance from coast are applied, land contamination exists and is therefore necessary to set up

a criterion based on distance from coast. Therefore, all MWR-derived WTCs within distances from coast lesser than this threshold value are flagged as invalid in the GPD+ methodology (even if they are set as valid in the GDR) and not used as observations. Consequently, this threshold value can be useful in forthcoming GPD+ versions to estimate the WTCs for all points within this distance from coast. Figure 10 shows that the RMS of the differences GNSS-GPD+ decreases when approaching coast, where the stations and the number of differences generally increase, indicating that the GPD+ WTCs

estimates are valid up to the coastline and that these WTC values are recovered at all along-track points without valid MWR-derived WTCs. Moreover, Fig. 10 shows that the GPD+ methodology recovers the WTC not only along the coastal areas, but also offshore.

For other missions, results have been presented in Vieira at al. (2019b) and in Fernandes and Lázaro (2018) and are summarised here. For the 2-band radiometers, land contamination on the MWR observations occurs for points at distances from coast

smaller than 25-30 km (ERS-1 and ERS-2), 20-25 km (Sentinel-3) and 15-20 km (GFO and SARAL), the latter in agreement with the smaller radiometer footprint of the SARAL MWR. Similar analysis shows that land contamination is observed up to 25-30 km from the coast for T/P and Jason-1 and up to 20-25 km for Jason-2 and Jason-3. These numbers are function both of the instrument footprint size and of the efficiency of the criteria used to detect valid/invalid MWR observations, since in these plots only MWR values that passed all validation criteria, except for the distance from coast, have been used. In summary, for 455 each mission, these analyses show the distances from coast up to which the MWR observations are contaminated by land and must be discarded. Moreover, they also show that GPD+ is efficient in removing this effect.

## 3.2 Performance assessment of the Envisat GPD+ WTC

Water vapour content can be accurately obtained by radio sounding data that could ideally be employed to validate the GPD+ estimates. Despite having high vertical resolution, radiosonde measurements are distributed only over limited areas, i.e., 460 regions where stations are located, do not cover oceanic regions and are very scarce over the Southern Hemisphere (Ye et al., 2017). Therefore, their low temporal and spatial resolutions have reduced their use as a validation tool in the context of satellite altimetry.

For this reason, the GPD+ products have been assessed through various SLA variance statistical analyses, analysing simultaneously the impact of the correction on sea level variability. The reasoning for adopting this analysis is that the larger 465 the variance reduction in the SLA signal when using a certain WTC, the better is the correction, i.e., the larger is the reduction in the SLA error, and closer to a pure oceanic signal is the SLA dataset that uses that correction. Therefore, three SLA datasets of collocated along-track points were derived using the same standard corrections (Sect. 1) but the WTC, which can be the GPD+, the MWR-derived or the ERA Interim WTCs. The criteria to select valid SLA are those recommended in the literature and adopted in the standard RADS processing (Scharroo et al., 2012) and include: application of thresholds for all involved 470 fields (satellite orbit above reference ellipsoid, altimeter range, all range and geophysical corrections), altimeter ice and rain flag (whenever set) and SLA within ±2m.

In the comparisons with the ERA Interim, all points with valid SLA have been selected, including points over ocean, coastal and polar regions. However, in the comparisons with the on-board MWR, only points for which the MWR-derived WTC is available and within the -50 cm - 0 cm range are used. Therefore, in the latter case, points with WTCs from the on-board MWR 475 which values are outside this range or are absent, have been discarded from the analyses. For Envisat cycle 12 (Fig. 4), these points are represented in dark green and correspond mainly to entire tracks for which no MWR-derived WTCs are available. Consequently, the number of points used in the WTC comparisons between GPD+ and ERA and GPD+ and MWR is different, however quite similar for both comparisons as it can be seen in Fig. 11 below.

Differences between each pair of SLA data sets are computed along track and at crossovers and the weighted variance estimated 480 for the time span of the whole Envisat period, with latitude-dependent weights (i.e., weights are function of the co-sine of latitude). Variance differences have been calculated in such a way that negative values represent an improvement in the description of the SLA field when the GPD+ WTC is used for its generation. For the computation of the crossovers, only

measurements with a temporal difference less than 10 days were used. Besides the temporal analysis, the variance differences, both calculated along-track and at crossovers, are also mapped for the analysis of their spatial distribution. In this latter case, the variances of the SLA differences are gridded onto 4-degree spatial resolution cells. Along-track SLA variance differences are also computed as function of latitude and distance from coast, where the variance for the whole Envisat period is computed over bins of latitude and distance from coast. Sub-section 3.2.1 shows the results obtained from the global analysis. Sub-section 3.2.2 shows the results zoomed into three different geographical domains: North American and European coasts, selected due the existence of the great quantity of GNSS stations, and Indonesia region, a challenging region in terms of coastal satellite altimetry.

### 3.2.1 Global Analysis

Figure 11 illustrates the results obtained for the period of the whole Envisat mission. From this figure, it is observed that the GPD+ WTC for Envisat represents, in general, a significant improvement when compared to the other WTCs selected for this assessment.

Usually the SLA variance reduction is analysed at crossover locations, however since oceanic variability with periods lower than 10 days is neglected when doing this analysis, whilst preserved in the along-track differences, both diagnostics are considered complementary. Figure 11 shows the results for both diagnoses: variance differences calculated along-track are shown in yellow, while variance differences at crossovers are represented in blue.

Using the GPD+ WTC instead of the MWR-derived WTC (Fig. 11a) leads, in the along-track analysis, to an improvement in the variance of the oceanic signal of 0.3 cm$^2$ on average, increasing in the second half of the period where values of 2 cm$^2$ can be reached in some cycles, most probably due to the globally poorer performance of both the MWR and the altimeter towards the end of the mission. For the GPD-MWR comparison, the SLA reduction is more noticeable in the along-track analysis than in the crossover analysis. Smaller variance differences are expected in this later analysis, since the GPD+ generally equals the MWR-derived WTC in open ocean, where most crossovers are located. Adopting the GPD+ WTC instead of the ERA Interim one (Fig. 11c) leads to a reduction in SLA variance which, on average, is in the range of 1 and 2 cm$^2$, for the analysis along the tracks, reaching a maximum value of 3 cm$^2$ in the analysis at crossovers. Therefore, it is expected that the GPD+ WTC leads to a reduction in the SLA variance over open ocean too. Figures 11b and 11d show the number of crossovers (in blue) and along-track pairs (yellow) used, per cycle, in the comparison of the GPD+ with the MWR-derived and ERA WTCs, respectively. A large amount of Envisat data was lost in the period corresponding to cycles 94 and 95, since a new orbit configuration (30-day repeat cycle) for the mission was implemented in October 2010, corresponding to a change from Envisat Phase b to Phase c.

Figure 12 shows the reduction in SLA variance globally, after being spatially averaged and gridded onto 4-degree spatial resolution cells, estimated at crossovers for the differences GPD+ and MWR-derived WTCs, and GPD+ and ERA WTCs, on top and bottom plots, respectively. In these plots, blueish colours represent an improvement in the SLA dataset by reducing the SLA variance. The improvement of the GPD+ WTC over the model WTC (Fig. 12b) is clear, with maximum values of

variance reduction in the tropical oceans, particularly over the Pacific Ocean. The improvement over the Southern Ocean and around the coast of Antarctica shows that the model WTC is not able to capture the full variability of the WPD field yet. Figure 12a shows that the GPD+ and the MWR-derived WTCs are equal over the eastern oceanic basins (SLA variance close to zero, represented by the green colour) as expected, since the GPD+ preserves the valid MWR-derived WTC over open

ocean. However, despite the SLA improvement when using GPD+ WTCs being smaller than that when the ERA WTCs are used, it can be emphasized that the improvement is not limited to the coastal regions, being clear over e.g. the regions where the western boundary currents flow. Therefore, the use of third-party, mainly SI-MWR, data can help the description of the WPD field. Over the Southern Ocean, for latitudes 80ºS-60ºS, some degradation is visible when the GPD+ is used. This could probably be due to the existence of ice contamination in the radiometer-derived (both along-track and image) WTCs. However,

it is recalled that, over this region, the MWR-derived WTC is usually missing (default value or NaN) or out of range, and that these points, for which a GPD+ estimate would be computed otherwise, have been removed from the analysis. Therefore, it must be emphasized that these results for the comparison GPD+ and MWR-derived WTCs provide underestimated results for the GPD+.

SLA variance differences have also been analysed as function of latitude and distance from coast and the results are shown in

Fig. 13. Both the differences between GPD+ and ERA WTCs and GPD+ and MWR-derived WTCs are represented. The variance of the SLA dataset is reduced when GPD+ is used instead of the ERA WTC for all latitudes (Fig. 13a). The improvement of the GPD+ WTC with respect to the model one, with an average value of 1.3 cm$^2$ is maximum over latitudes where maximum atmospheric water content can be found, namely over the subtropical ocean and over latitudes where the western-boundary currents flow, particularly in the northern hemisphere where the variance reduction surpluses 2 cm$^2$. As

expected, the improvement is smaller for the comparison with the MWR-derived WTC, since this analysis includes open-ocean points where both corrections are equal. Leading to an improvement in the SLA variance of 0.32 cm$^2$ on average, the GPD+ WTC has its best performance against the WTC from the radiometer in the extratropical ocean, especially in the northern one. The increase in the reduction of the SLA variance at these latitudes is associated to a better description of the WPD field in the coastal regions northwards of the regions where the western boundary currents flow (off Newfoundland and in the Sea

of Okhotsk), as can be concluded from the maps showing the reduction in SLA variance for the difference GPD+ and MWR-derived WTCs, computed along-track and spatially averaged at each 4-degree cell (not shown). The SLA dataset is also improved over the coastal regions when the GPD+ WTC is applied (Fig. 13b). The improvement over the ERA WTC is, on average, 0.77 cm$^2$ in the 30 km closest to land, increasing to ~1.4 cm$^2$ for larger distances. This means that a better description of both the WTC and SLA fields is obtained over open ocean when the GPD+ WTC is adopted (cf. Fig. 12). The improvement

over the WTC from the on-board MWR is larger in the nearest 20 km to the coast, where the reduction in variance can reach 3.3 cm$^2$ (average value is 2.0 cm$^2$). As the distance to shore increases, the reduction in variance decreases, although still negative and around -0.60 cm$^2$ on average. This result is expected, since the number of invalid MWR-derived WTCs decrease offshore and therefore the GPD+ WTCs equal those retrieved from the MWR measurements.

The improvement obtained when the GPD+ methodology is applied to coastal areas is unfortunately not completely evident in

the presented results, since the MWR-derived WTC for those points for which this correction is missing or outside limits have not been in the analyses. For these points, if available, the MWR-derived WTCs are expected to be significantly worse than the GPD+ one.

### 3.2.2 Coastal Analysis

This section shows zoomed-in results for three different regions: North American and European coasts, and Indonesia region. The first two regions have been selected due to the great quantity of GNSS stations available along the coast (shown by the red dots in Fig. 2), while the third has been selected since it is recognised as being quite challenging for satellite altimetry. The results have been obtained for the whole period of the Envisat mission and all along-track points within the geographic limits have been considered. As already described in the previous section, points with MWR-derived WTC out of the range -50 cm

- 0 cm and those for which the WTC is not defined in the altimeter products are rejected from the comparisons with the on-board MWR, while in the comparisons with ERA all points with valid SLA are selected.

Results are illustrated in Fig. 14. Left panels show the SLA variance difference (in cm$^2$) function of distance from coast calculated along the satellite tracks, where negative variance differences represent an improvement in the description of the SLA field when the GPD+ WTC is used. Right panels show the spatial distribution of the weighted SLA variance differences

(in cm$^2$) computed along the satellite tracks, after being spatially averaged and gridded onto 4-degree spatial resolution cells. In these latter plots, blueish colours represent an improvement in the SLA dataset (reduction in the SLA variance) when the GPD+ WTC is used.

All the regions show that the SLA variance is reduced along the coasts when the GPD+ WTC is used rather than the MWR- (in green) or the ERA-derived (in blue) WTCs. For the North American coast (Fig. 14a, left panel), the improvement is clear

up to 100 km off the coast. For distances up to 40 km off the coast, the reduction in SLA variance is, on average, 8.7 cm$^2$, being ~3.4 cm$^2$ when averaged for distances between 40-100 km off the coast. For larger distances, the differences tend to zero, since the GPD+ preserves the valid MWR-derived WTC and therefore both corrections are equal. The comparison with the ERA-derived WTC shows an averaged SLA variance difference of -1.2 cm$^2$ (GPD+ reducing the variance) for the whole range of distances. The right panel of Fig. 14a shows that the reduction in SLA variance, when the GPD+ correction is used

instead of the ERA-derived one, is larger along the eastern coast, where the WTC variability is larger (cf. Fig. 2), and that the improvement is not limited to the coastal zone, but is also clear over open ocean. This result can be extended to the three selected regions.

For the European region (Fig. 14b, left panel), an improvement of 1.5 cm$^2$ is, on average, obtained for the comparison GPD+ and MWR-derived WTCs for the 20 km closest to the coast. For larger distances, and up to 100 km off the coast, the averaged

reduction in SLA variance is 0.67 cm$^2$. The comparison with the ERA-derived WTC shows an SLA variance difference of -1.2 cm$^2$ (GPD+ reducing the variance), on average, for the whole range of distances. SLA variance reduction is notorious over the Mediterranean region (Fig. 14b, right panel).

For the Indonesia region, the improvement of the GPD+ WTC with respect to the MWR-derived one is mainly achieved in the 20 km closest to the coast, where the SLA variance reduction is, on average, 1.4 cm$^2$. The use of GPD+ WTC instead of ERA-derived WTC leads to an improvement that, on average, is of the order of 2.2 cm$^2$ for the whole range of distance from coast. This reduction is observable over almost the whole region, being larger in its northern part.

The results obtained for the comparison with the ERA WTC are a clear indication that current NWM do not correctly represent the WTC field variability yet. This result can also be extracted from Fig. 7 and Fig. 8, where it is seen that the NWM-derived WTC does not exhibit the small spatial scales as well as the MWR-derived, and consequently, GPD+ WTCs.

Once again, it is worth noticing that, in these results, the improvement obtained when the GPD+ methodology is applied to coastal areas is underestimated, since the MWR-derived WTCs for those points for which this correction is missing or outside limits have not been used in the performed analyses. For these points, the MWR-derived WTC, if available, would probably be contaminated by land and would degrade the MWR-derived dataset.

## 4  Data Availability

The GPD+ WTCs are freely available in NetCDF format at the UPorto's Satellite Altimetry repository https://doi.org/10.23831/FCUP_UPORTO_GPDPlus_v1.0 (Fernandes et al., 2019) and at the AVISO (Archiving, Validation and Interpretation of Satellite Oceanographic data) webpage (https://www.aviso.altimetry.fr/en/data/products/auxiliary-products/gpd-wet-tropospheric-correction.html).

## 5  Conclusions

The wet tropospheric correction (WTC) is still considered an important source of error in satellite altimetry, particularly in coastal and polar regions, where the retrieval of the wet path delays from the microwave radiometer (MWR) measurements on board the altimetry missions leads to invalid values. During the data processing aiming at deriving the sea level anomaly, altimeter measurements are discarded if the WTC is absent, which is frequent in coastal and polar regions. In the last years, a huge effort has been made to develop methodologies capable of computing WTC estimates where the correction is absent, while keeping the high-accuracy of MWR-derived WTC values. A few methodologies emerged, among which the GPD and its most-updated version GPD+ have proven to be the most effective in reducing the SLA variability due to non-ocean phenomena, simultaneously leading to the recovery of a significant number of measurements.

This paper describes the GPD+ WTC database and exemplifies the results using as input the Envisat FMR V3.0. The GPD+ WTC equals the MWR-derived WTC whenever this latter is valid, thus preserving its accuracy. For those MWR-derived WTCs detected by the algorithm as anomalous, a new estimate and its associated mapping error are computed. The GPD+ algorithm has been trained to detect land, ice, and outlier-contaminated measurements, besides those identified in the GDR

data already. On top of preserving the accuracy of the WTC derived from the on-board MWR measurements, the GPD+ algorithm guarantees the continuity and consistency of the output WTC globally and, in particular, in the coastal zone.

Prior studies using a previous GPD+ version (e.g., GPD algorithm cf. Fernandes et al. (2015)) show that the GPD WTC led to a significant improvement of the SLA dataset for T/P and ESA-funded missions, since these, particularly the latter, had on-board MWRs which retrieval algorithms output very noisy values in coastal and ice contaminated regions. For these missions, the GPD WTC was proven to be the preferred WTC to be used in the definition of the SLA field, when compared to the baseline MWR one, the model-derived one and the AVISO reference composite correction, provided in their products (Legeais et al., 2018). The main advantage of the methodology when applied to the T/P mission is the correction of several TOPEX/Poseidon Microwave Radiometer (TMR) anomalies present in the second part of the mission, particularly noticeable in the Indian Ocean, which would otherwise seriously affect the calculation of the mean sea level at regional scales (Fernandes et al., 2015).

The GPD+ WTCs for GFO and CryoSat-2 missions have been described in Fernandes and Lázaro (2016). Despite the MWR on board GFO mission being considered a stable and accurate instrument, it had periods of malfunctioning, particularly in the last years of the mission. In addition to improving the derived SLA dataset, by reducing the error associated with non-pure oceanic signal, the GPD+ recovers the WTC for the periods during which the GFO MWR was defective. For CryoSat-2 mission, without an on-board MWR and therefore without a WTC relying on observations, the GPD+ is computed for all along-track points. GPD+ WTC thus replaces the NWM-derived WTC that otherwise would have to be used instead. For this mission, the exploitation of third-party data has been proven to be very effective. As the results in this paper show, the NWM-derived WTCs are still inaccurate since they are limited to a poor spatial and temporal resolution.

Products available for Jason missions already possess a coastally improved WTC (Brown, 2010). Still, although small, some improvement, particularly at high latitudes and mainly for Jason-1 can be achieved when the GPD+ correction is used in the generation of the SLA dataset (Fernandes et al., 2015). The current version of the correction (GPD+) for the reference missions leads to more accurate retrievals than before, due to several improvements (e.g. the inclusion of WPD third-party observations from imaging radiometers and a better screening for anomalous MWR-derived WTCs). Due to the fact that, contrary to Jason missions, T/P products do not possess a coastal enhanced WTC, the improvements reached by GPD+ are more significant for T/P than for Jason. For all other RA with 2-band MWRs (ERS-1, ERS-2, Sentinel-3, SARAL and GFO), GPD+ proves to be a significant improvement over NWM, MWR and the AVISO composite WTC, reducing the SLA variance (both along-track, at crossovers, function of distance from coast and function of latitude) by 1-2 cm$^2$ (Fernandes and Lázaro, 2016, 2018).

Many authors have also proven the positive impact of the GPD+ corrections, particularly in coastal studies, e.g. Handoko et al. (2017) in the Indonesia region and Dinardo et al. (2018, 2020) in the German Bight.

Taken as a whole, the GPD+ algorithm possesses the advantage of being able to compute the WTC at a considerable number of along-track points with an invalid/inexistent MWR-derived WTC, therefore leading to the recovery of the SLA signal at these points. The percentage of recovered points when GPD+ is applied in place of the baseline MWR-derived WTC depends on instrument type, band of latitudes covered by the mission (which determines the extent of ice contamination) and instrument

performance. For all ESA missions (ERS-1, ERS-2, Envisat, Sentinel-3) and SARAL, possessing 2-band radiometers and measuring up to latitudes ±81.2º, the percentage of recover data is similar to that of Envisat, in the range of 7% - 15% of the SLA valid points of each cycle. For the reference missions, measuring only up to ±66.7° and already possessing an improved WTC near the coast (all except T/P), this percentage is smaller, from 2 to 4%. For T/P, these values are from 4% to 7%, larger in the second half of the mission. For GFO, measuring up to ±72.0°, the percentage is similar to that of TP. Exceptions occur for various missions over periods of instrument malfunction, when the percentage of recovered points can be considerably larger, up to 100%, as it happens for Envisat and GFO.

Moreover, the GPD+ WTC is a continuous correction in the ocean/land interface region, as well as in the polar regions. The scientific novelty and practical significance for the common satellite altimetry user is that the GPD-corrected SLA dataset can be used for coastal applications, constituting a major step forward for satellite altimetry to become a tool for coastal management.

Despite significant efforts made in the past to improve the WPD calculation at GNSS-station height and the sea-level reduction of the correction to use in satellite altimetry over ocean, the unpredictable way the WPD varies with altitude is still a factor constraining the precise GNSS data reduction procedure, since all other data are provided at sea level. Therefore, the modelling of the 4D variability of the WPD field is under research at UPorto (Vieira et al., 2019c). It is expected that a better knowledge of the WTC variability will improve the GPD+ WTCs aiming at a larger reduction of the sea level variance due to non-oceanic signals, since the whole GNSS data processing upstream to the GPD+ computation is also performed at UPorto.

Upcoming developments include: i) the inclusion of an ameliorate modelling of the WTC vertical variability (Vieira et al., 2019c), leading to a better consistency of the various datasets combined in the OA procedure; ii) the extension of the corrections to all surface types with new estimates over all regions where observations exist, e.g. large lakes and rivers where valid MWR and GNSS can be exploited; iii) and, for the older missions, the replacement of the ERA Interim model by ERA5, the most recent reanalysis by ECMWF (Vieira et al., 2019d).

**Author Contributions.**

MJF and CL developed the methodology and the code. All authors performed the analyses. CL prepared the manuscript with contributions from all co-authors. All authors have read and approved the final paper.

**Competing Interests.**

The authors declare that they have no conflict of interest.

**Acknowledgements.**

The results here provided for Envisat FMR V3.0 were obtained in the scope of the ESA funded project CLS-SCO-17-0034, ENVISAT RA-2 LEVEL 1B ESL AND PROTOTYPE MAINTENANCE SUPPORT, Subcontract to ESA/Contract N° 4000110859/14/I-AM. The authors thank Radar Altimeter Database System (RADS) for providing the GPD+ input altimeter data for all missions except Envisat and SARAL up to cycle 30, Aviso+ (https://www.aviso.altimetry.fr/) for the production and distribution of SARAL PEACHI products up to cycle 30, the European Centre for Medium-Range Weather Forecasts (ECMWF) for making both the ECMWF operational and the ERA Interim models available, and all institutions providing the water vapour products used in this study: National Oceanic and Atmospheric Administration (NOAA) – Comprehensive Large Array-Data Stewardship System (CLASS) and Remote Sensing Systems. SSM/I and SSMIS data are produced by Remote Sensing Systems and sponsored by the NASA Earth Science MEaSUREs Program and are available at www.remss.com.

**Financial Support.**

Telmo Vieira is supported by the Fundação para a Ciência e a Tecnologia (FCT) through the PhD grant SFRH/BD/135671/2018, funded by the European Social Fund and by Ministério da Ciência, Tecnologia e Ensino Superior (MCTES). This research was also supported by Centro Interdisciplinar de Investigação Marinha e Ambiental (CIIMAR) through the project with reference UID/Multi/04423/2019.

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

**Table 1. Total Column Water Vapour (TCWV) availability. For gridded products, two grids per day are made available, each grid comprising the ascending/descending passes. For the swath products, 14-15 orbital swaths per day are available for each instrument. For these latter products, the value provided for the spatial resolution is that of the central pixel (maximum value for pixel size is 130 km).**

| Satellite/Sensor | Spatial Res | Temporal Res. | Availability |
| --- | --- | --- | --- |
| DMSP-F08/SSM/I | 0.25° × 0.25° | 2 grids/day | July 1987–December 1991 |
| DMSP-F10/SSM/I | 0.25° × 0.25° | 2 grids/day | December 1990–November 1997 |
| DMSP-F11/SSM/I | 0.25° × 0.25° | 2 grids/day | December 1991–May 2000 |
| DMSP-F13/SSM/I | 0.25° × 0.25° | 2 grids/day | May 1995–November 2009 |
| DMSP-F14/SSM/I | 0.25° × 0.25° | 2 grids/day | May 1997–August 2008 |
| DMSP-F16/SSM/IS | 0.25° × 0.25° | 2 grids/day | since October 2003 |
| DMSP-F17/SSM/IS | 0.25° × 0.25° | 2 grids/day | since December 2006 |
| DMSP-F18/SSM/IS | 0.25° × 0.25° | 2 grids/day | since October 2009 |
| NOAA-15/AMSU-A | 50 km | 14-15 orbital swaths per day | since July 2003 |
| NOAA-16/AMSU-A | 50 km | 14-15 orbital swaths per day | July 2003–June 2014 |
| NOAA-17/AMSU-A | 50 km | 14-15 orbital swaths per day | July 2003–April 2013 |
| NOAA-18/AMSU-A | 50 km | 14-15 orbital swaths per day | since August 2005 |
| NOAA-19/AMSU-A | 50 km | available on an orbital basis | since May 2009 |
| MetOp-A/AMSU-A | 50 km | 14-15 orbital swaths per day | since May 2007 |
| MetOp-B/AMSU-A | 50 km | 14-15 orbital swaths per day | since April 2013 |
| AQUA/AMSR-E | 0.25° × 0.25° | 2 grids/day | May 2002–October 2011 |
| GCOM-W1/AMSR-2 | 0.25° × 0.25° | 2 grids/day | since May 2012 |
| TRMM/TMI | 0.25° × 0.25° | 2 grids/day | December 1997–March 2015 |
| Coriolis/WindSat | 0.25° × 0.25° | 2 grids/day | since February 2003 |
| GMI | 0.25° × 0.25° | 2 grids/day | since April 2014 |

**Table 2. Calibration parameters (offset, scale factor and linear trend) obtained for all RA missions with an on-board MWR included in the GPD+ database (Fernandes et al., 2019). For Jason-3 (J3) and SARAL (SA) missions, no parameter for the linear trend has been computed due to the short length of their datasets. For explanation on the mission codes, please refer to Table 4.**

| Satellite Altimetry mission | offset ($a$)/offset error (mm) | scale factor ($b$) /scale factor error | linear trend ($c$)/linear trend error (mm/year) |
|---|---|---|---|
| TP | -8.05/0.041 | 0.978/0.0001 | 0.150/0.005 |
| J1 | -5.09/0.089 | 0.987/0.0001 | -0.049/0.006 |
| J2 | -6.25/0.143 | 0.980/0.0002 | -0.178/0.007 |
| J3 | -9.44/0.007 | 0.992/0.0000 | 0.000/0.000 |
| E1 | -12.04/0.127 | 0.964/0.0006 | 0.169/0.041 |
| E2 | -12.28/0.108 | 0.958/0.0004 | 0.050/0.013 |
| EN | -6.82/0.182 | 0.991/0.0004 | -0.0028/0.011 |
| GFO | 4.71/0.232 | 0.993/0.0004 | 0.0153/0.020 |
| SA | -3.70/0.092 | 0.992/0.0007 | 0.000/0.000 |

**Table 3. Data content in each GPD+ WTC NetCDF file, for the time and location of each RA mission measurement (Fernandes et al., 2019).**

| Variable | Description |
|---|---|
| time_01 | time of measurement, UTC seconds since 2000-01-01 00:00:00.0 |
| lat_01 | latitude of measurement, as in the GDR file |
| lon_01 | longitude of measurement, as in the GDR file |
| GPD_wet_tropo_cor _01 | GPD+ wet tropospheric correction (metres) |
| GPD_wet_tropo_cor_qual_01 | validity flag of the GPD+ estimate: 0-valid, 1-invalid |

**Table 4. Mission Code used in the name of the GPD+ Datasets (Fernandes et al., 2019) and their availability.**

| Mission Code | Mission | Start Time | End Period |
|---|---|---|---|
| TP | TOPEX/Poseidon | 1992/09 (cycle 1) | 2005/10 (cycle 481) |
| J1 | Jason-1 | 2002/01 (cycle 1) | 2012/03 (cycle 374*) |
| J2 | OSTM/Jason-2 | 2008/07 (cycle 1) | 2019/10 (cycle 383) |
| J3 | Jason-3 | 2016/02 (cycle 1) | 2020/01 (cycle 145) |
| E1 | ERS-1 | 1991/08 (phase A, cycle 1) | 1996/06 to phase g, cycles 156* or 53** |
| E2 | ERS-2 | 1995/05 (cycle 1) | 2011/05 (cycle 167) |
| EN | Envisat | 2002/05 (cycle 6) | 2012/03 (cycle 113) |
| GFO | GEOSAT Follow-On | 2000/01 (cycle 37) | 2008/09 (cycle 223) |
| C2 | CryoSat-2 | 2010/07 (sub-cycle 4) | 2020/01 (sub-cycle 126) |
| SA | SARAL/AltiKa | 2013/03 (cycle 1) | 2016/07 (cycle 35) |

\* RADS convention

\*\* AVISO convention




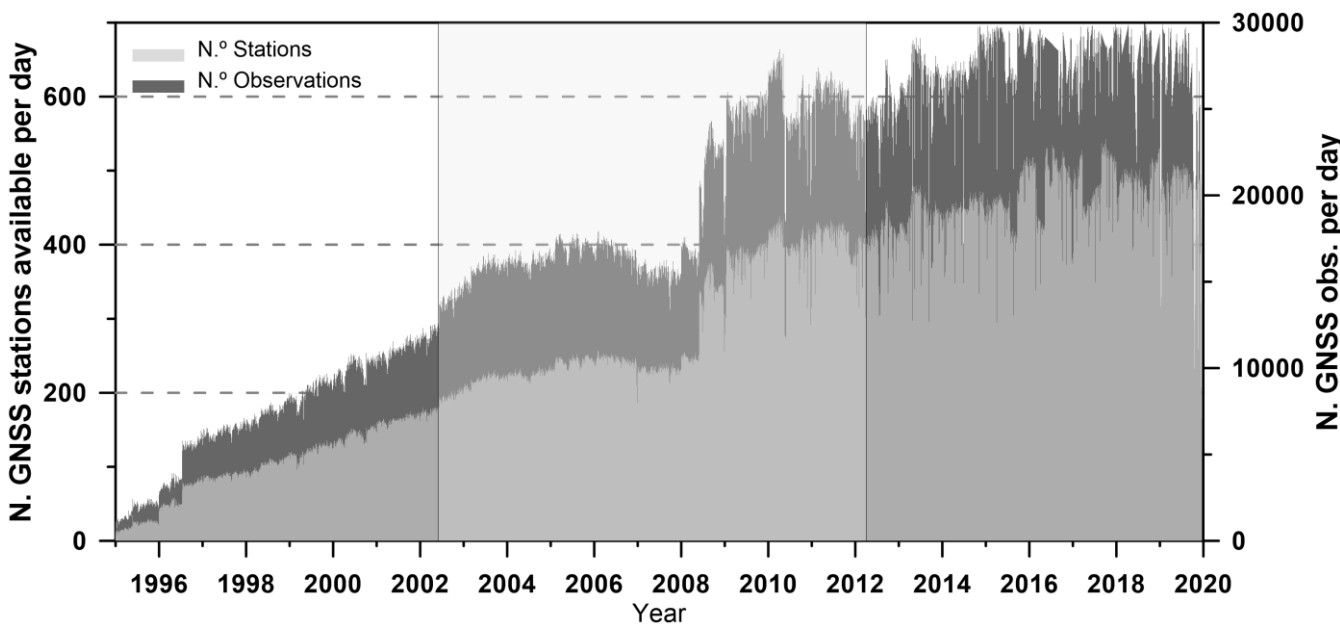

**Figure 1 Number of GNSS stations used in the GPD+ over time (light grey) and number of available GNSS observations per day (dark grey), for the whole RA era. Envisat period (5/2002-3/2020) is shown by the shaded rectangle. All GNSS stations are at a distance from coast less than 100 km.**




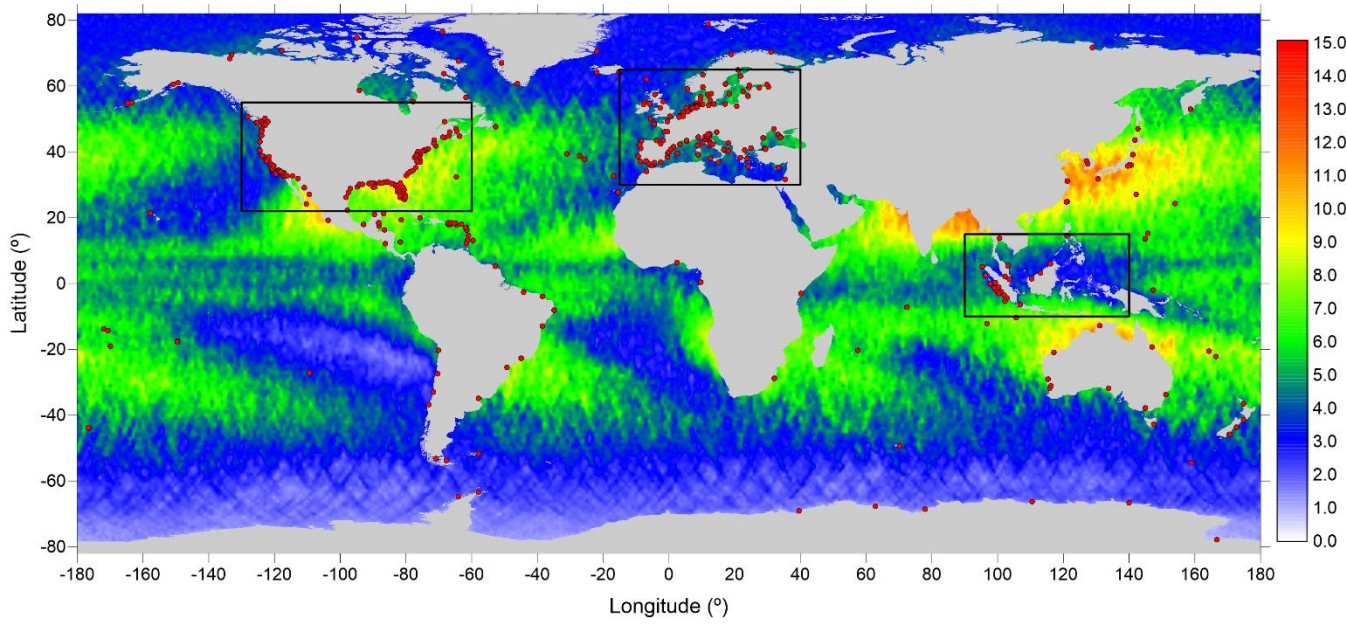

**Figure 2 Location of the coastal and island GNSS stations used in the GPD+ (red dots). Background image shows the standard deviation of the WTC field, in centimetres, computed using ERA Interim extracted for Envisat along-track points for the period November 2010 -November 2011 (cycles 96 to 108). The black rectangles show the regions selected to perform the coastal assessment of the GPD+ WTCs (North American and European coasts and Indonesia region).**


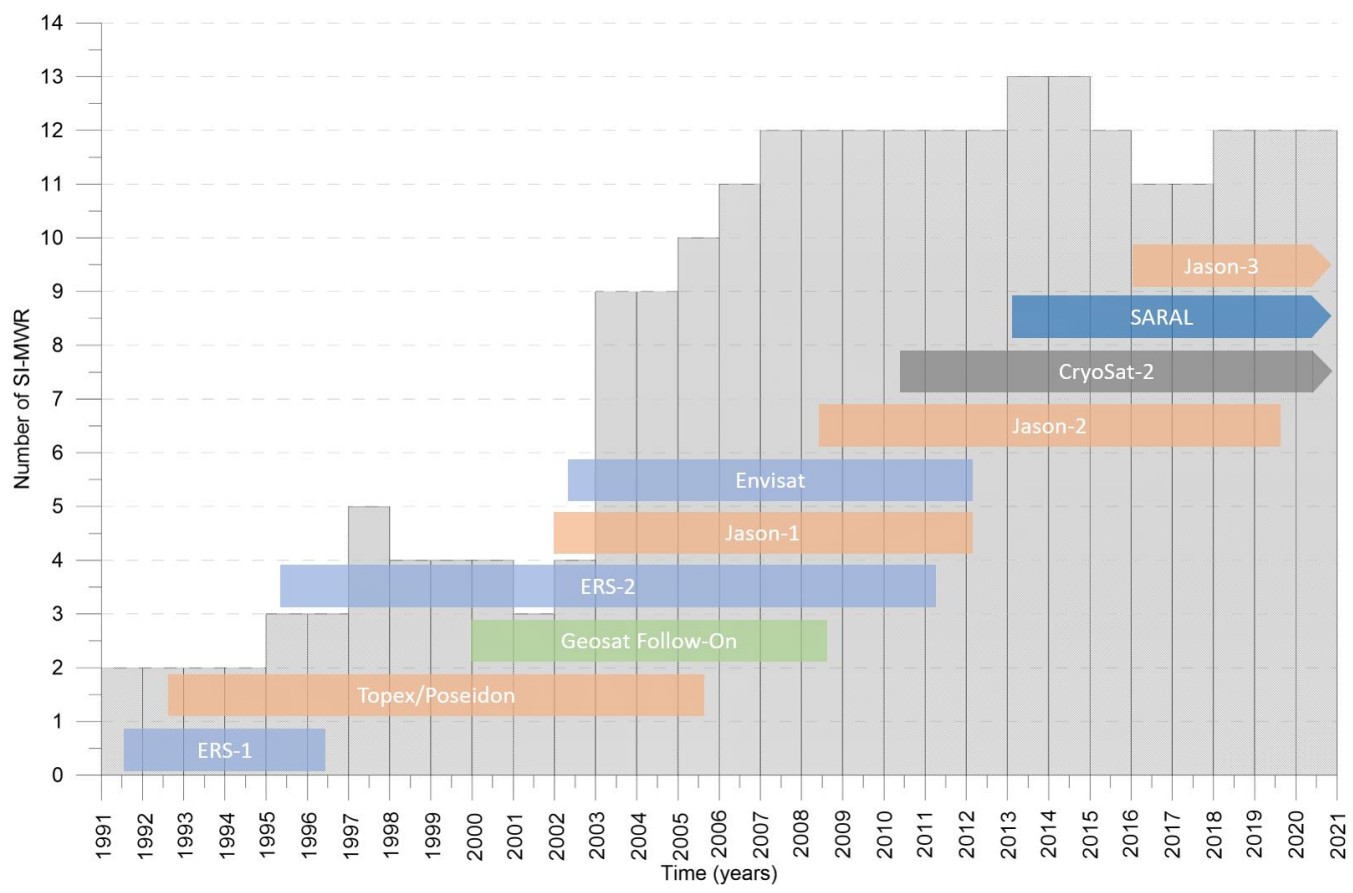


**Figure 3 Number of SI-MWR used in the GPD+ along time and period covered by each RA mission. SARAL, CryoSat-2 and Jason-3 missions are currently operational RA missions.**



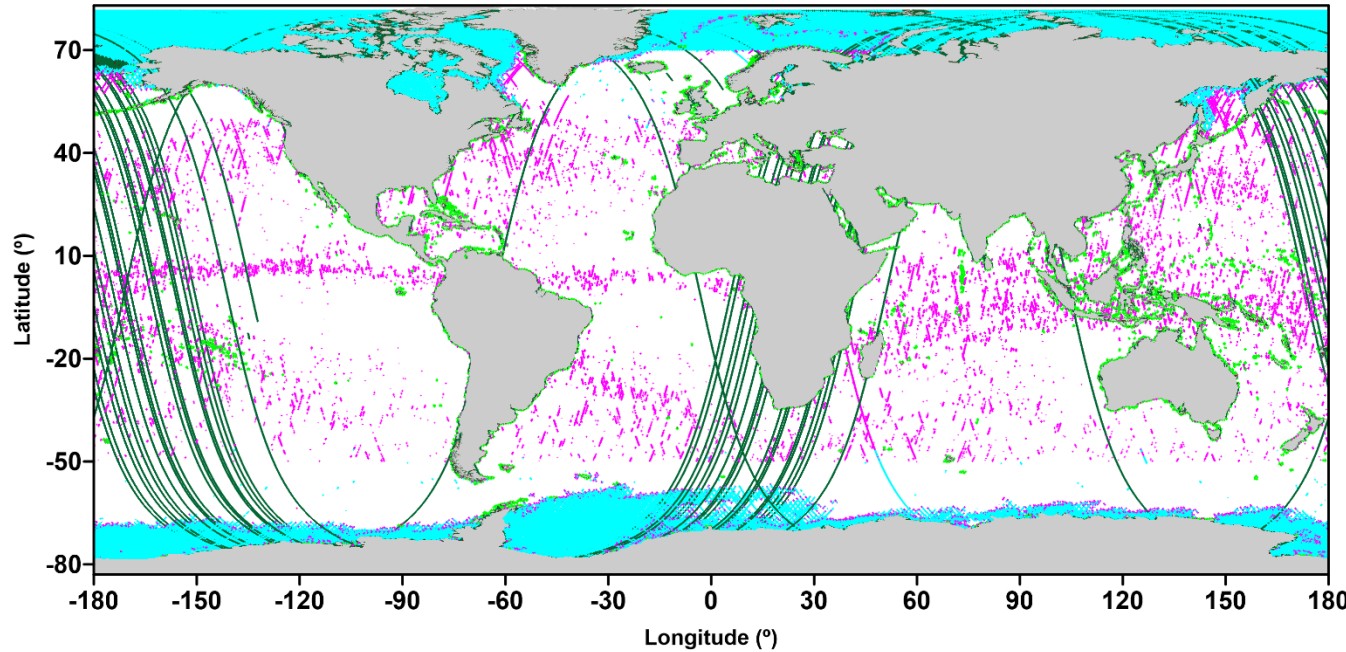

**Figure 4 Invalid MWR-derived WTC for Envisat cycle 12:● correction contaminated due to ice, ● correction contaminated due to rain and outliers; ● points flagged as coastal, may possess a correction contaminated by land; ● no available MWR-derived WTC value (the "fill value" is given). A note must be made that there are several points with available MWR-derived field but with an invalid value and without any error flag, that are detected and flagged by the GPD+ algorithm.**





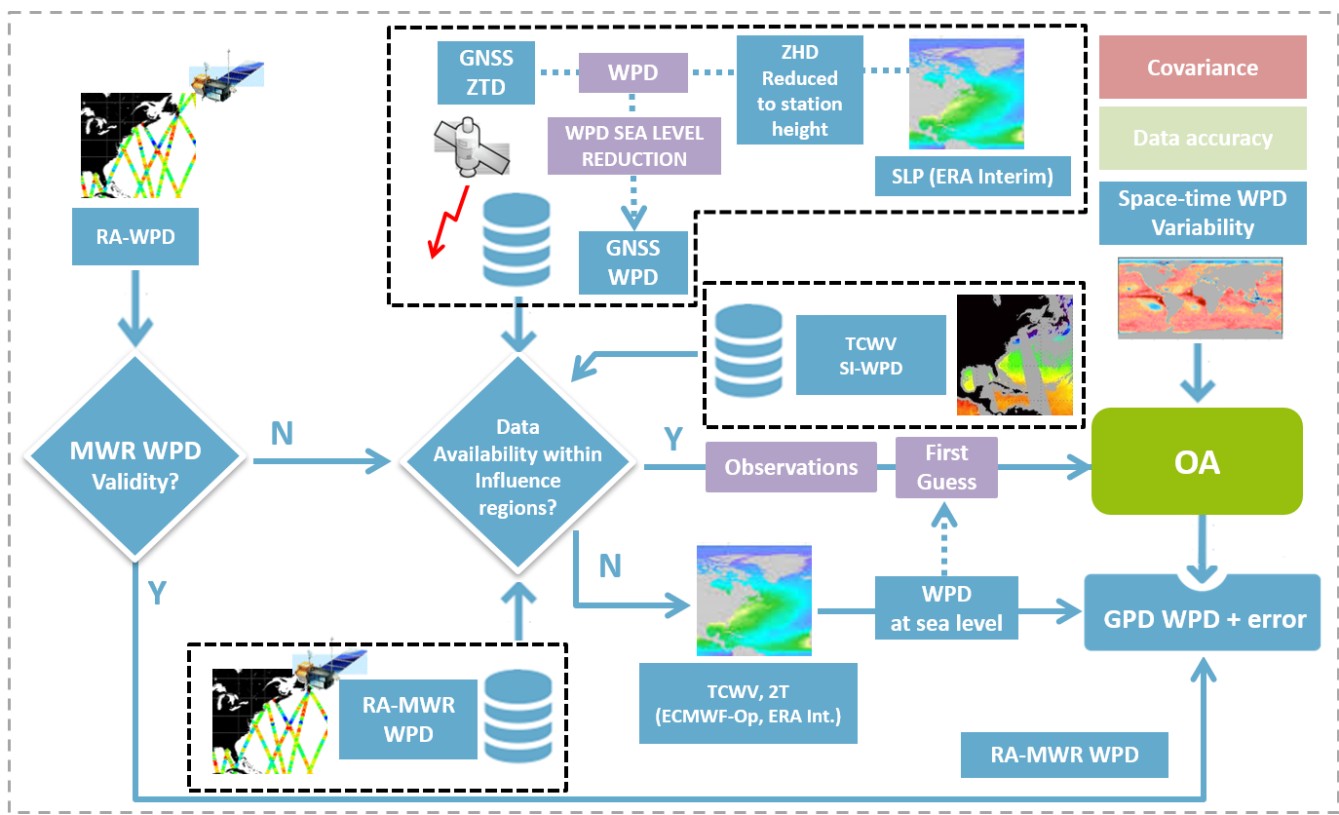

**Figure 5 Fluxogram of the GPD+ algorithm.**




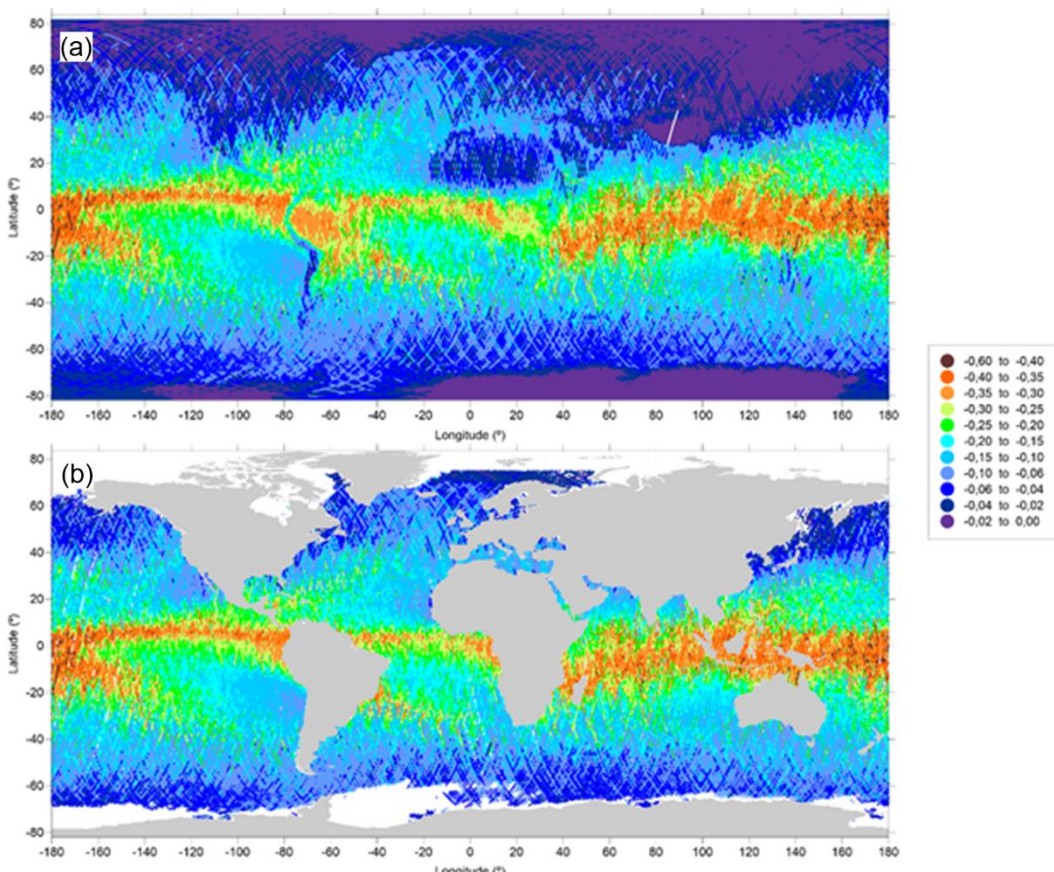


**Figure 6 GPD+ WTC, in metres, for Envisat cycle 012: (a) global coverage and (b) correction over oceanic regions with valid SLA.**

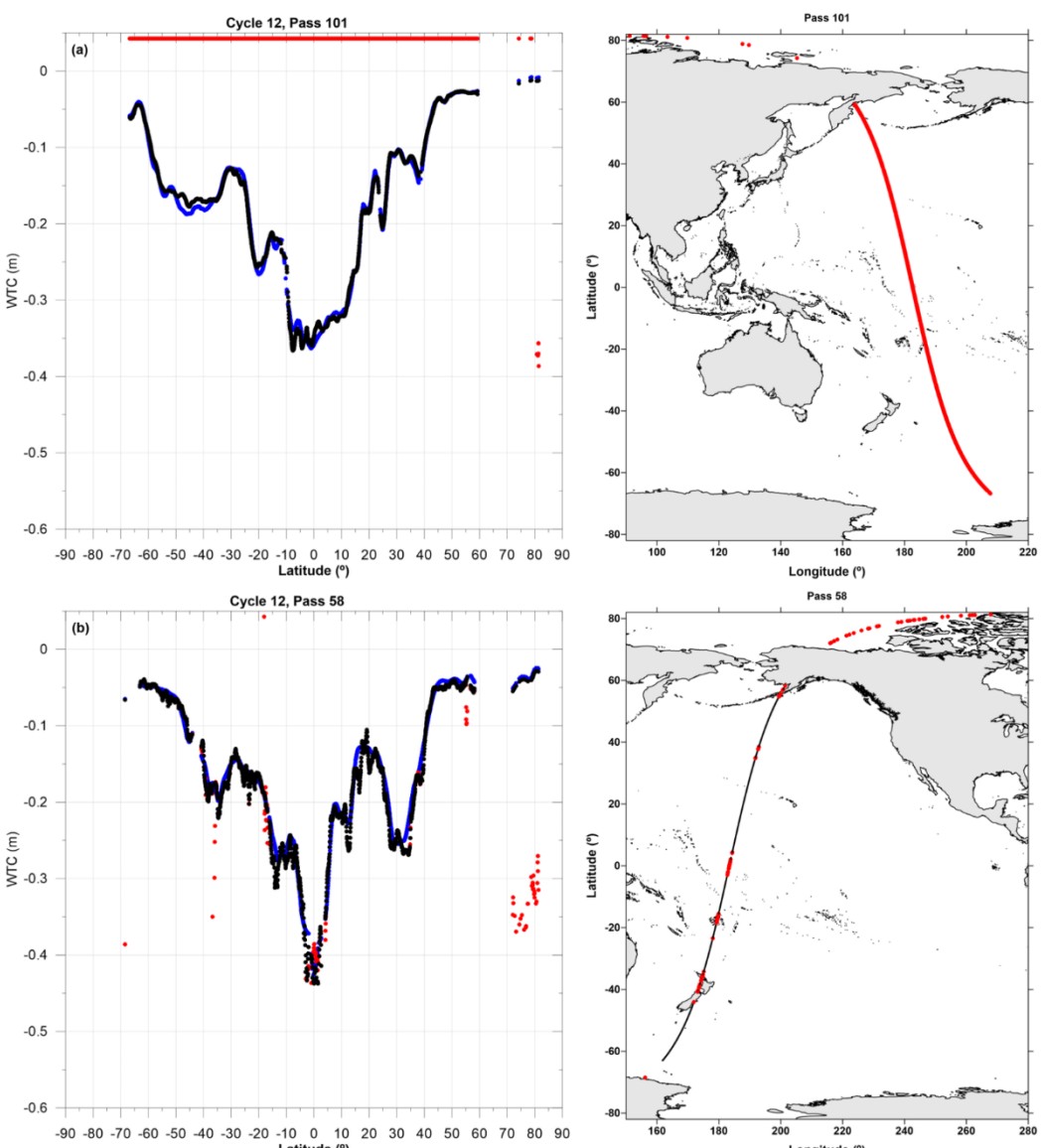

**Figure 7** Left: Along-track WTCs (m) present in the Envisat FMR V3.0 products for Envisat Cycle 12, passes 101 and 58 (panels (a) and (b), respectively): GPD+ (black), MWR-derived WTC (red) and ERA Interim WTC (blue). Panel (a) shows an example of the unavailability of the onboard MWR-derived WTC; panel (b) shows contamination by ice and rain (red points around the Equator) in the MWR-derived WTC. Right: Geographical coverage of the Envisat tracks shown in the left panels (longitude is given in the 0°-360° range to show the entire track). Along-track points with a GPD+ estimate are shown in red, while point where the GPD+ kept the MWR-derived WTC are shown in black.



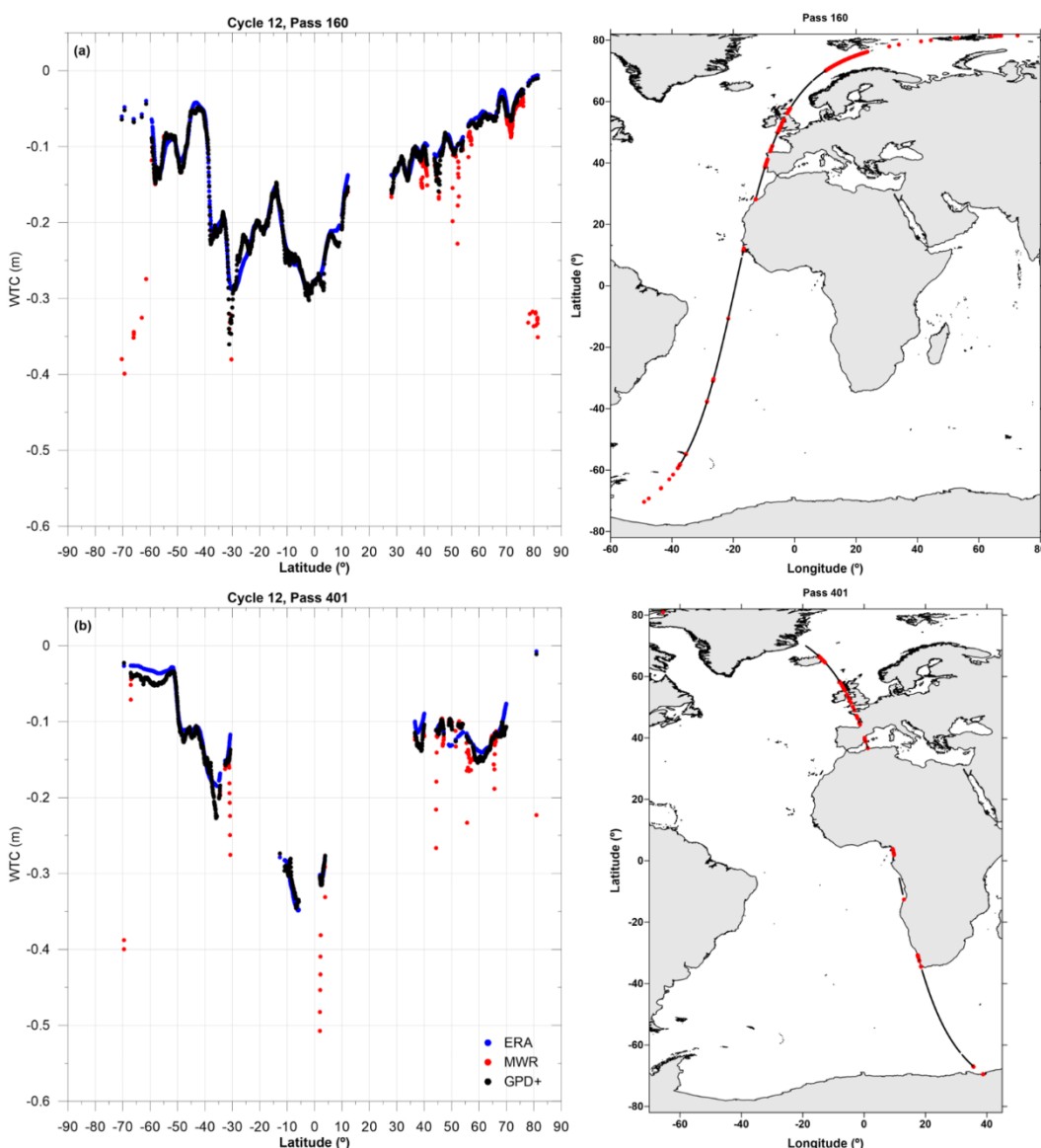

**Figure 8 Same as Fig. 7 for Envisat tracks 160 and 401 (Cycle 12) showing: (a) the existence of outliers (red points located over ocean between latitudes 30ºS and 40ºS); (b) contamination by land proximity.**



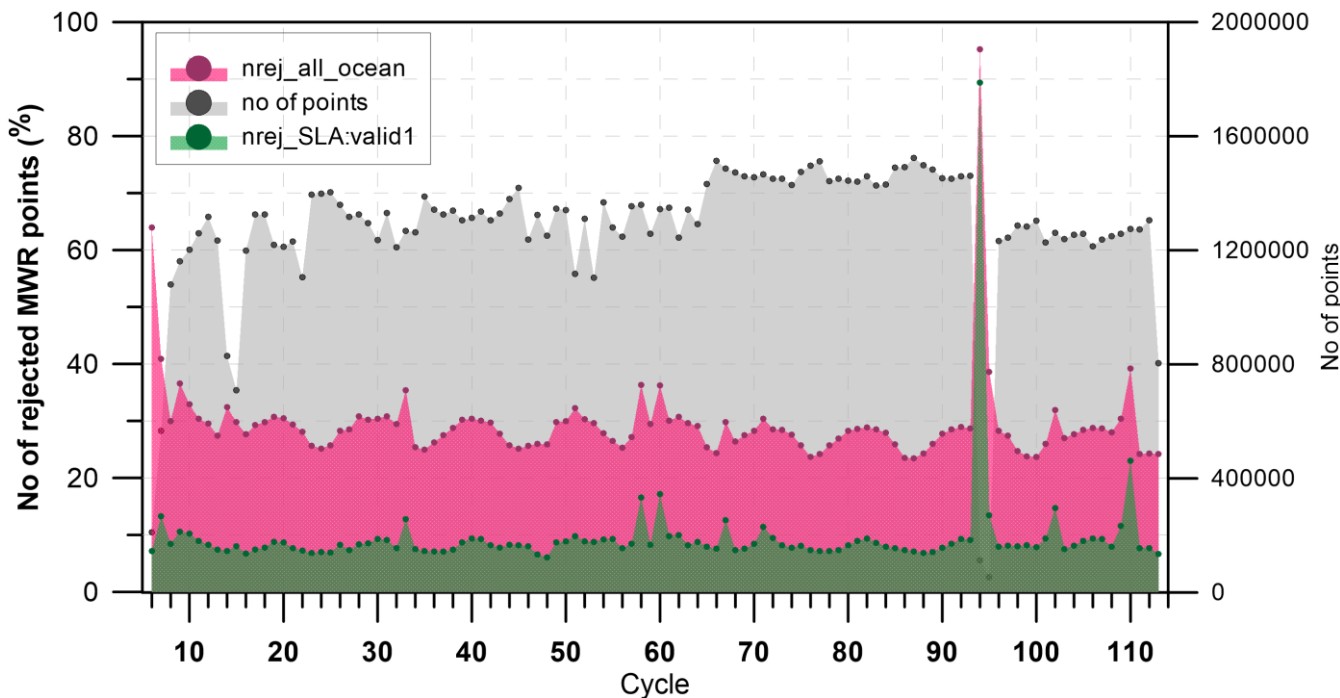

**Figure 9 Summary, for the whole Envisat period, of the percentage of points: (pink) with a rejected MWR-derived WTC, for which a GPD+ estimate has been computed; (green) for which a valid SLA value could be computed after the estimation of the WTC by the GPD+. Also shown in grey is the number of points with valid SLA values per cycle.**

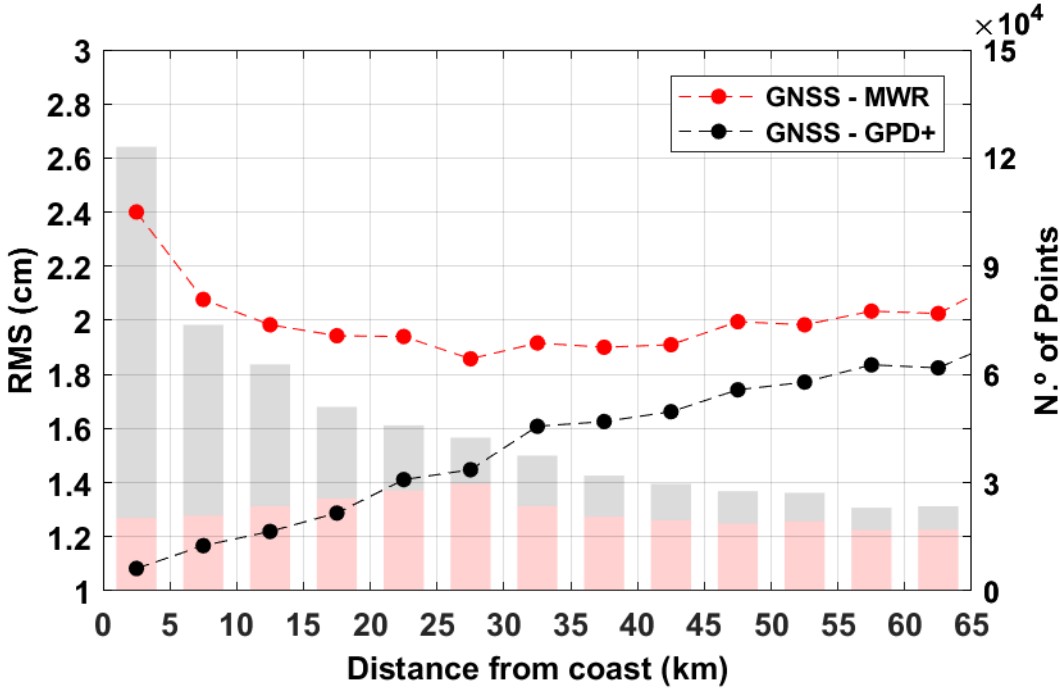

**Figure 10 RMS (in cm) of WTC differences (left axis) and number of altimetry measurements used (right axis) for the Envisat mission, function of distance from coast. Red bars represent the number of measurements used to compute the RMS of the differences GNSS-MWR, while grey bars represent the number of points used to compute the RMS of the differences GNSS-GPD+. In the comparison GNSS – MWR only valid MWR-derived observations have been used.**

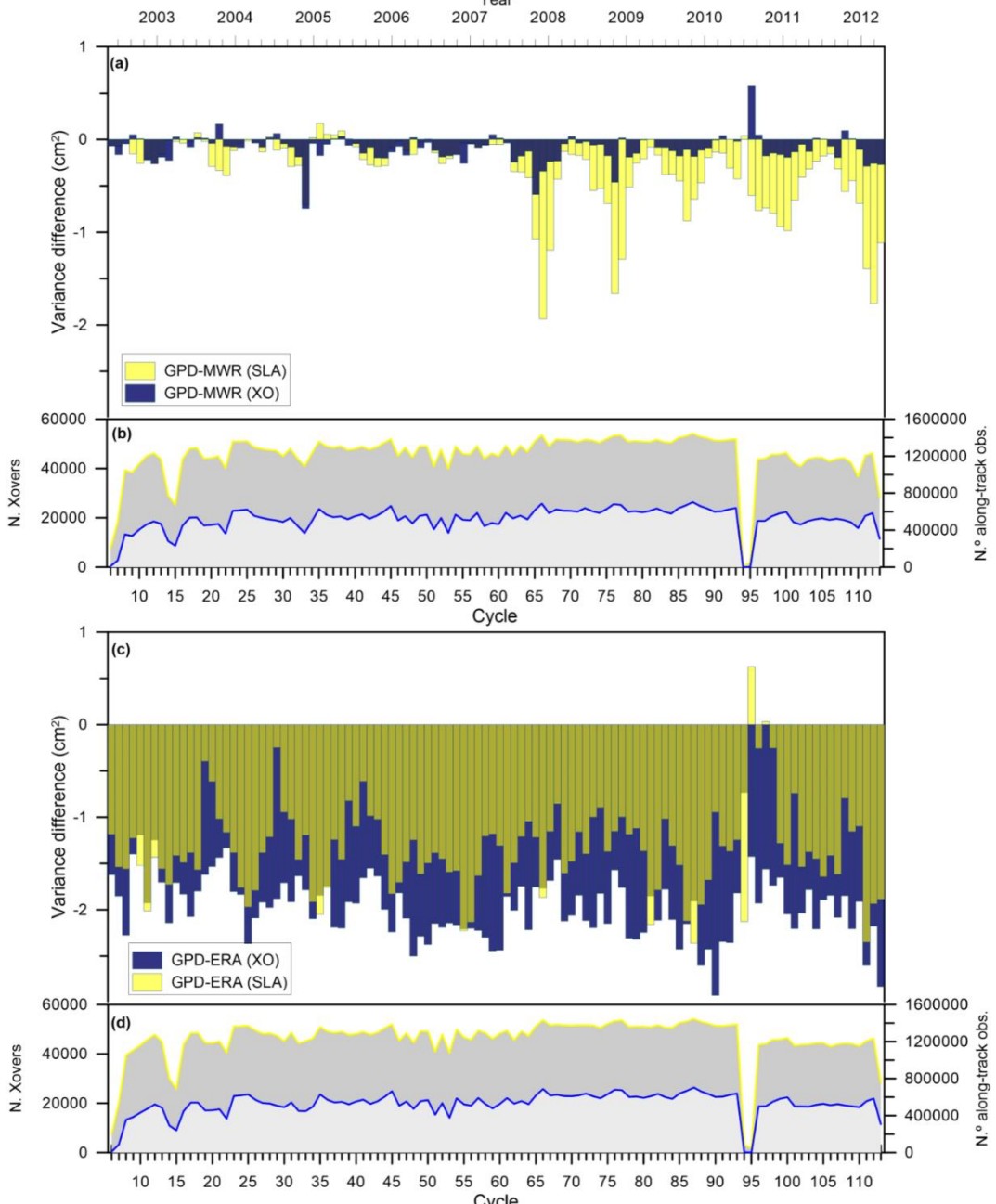

**Figure 11 Temporal evolution of weighted SLA variance differences (cm²) along satellite tracks (yellow) and at crossovers (blue) between (a) GPD+ and the MWR-derived WTCs and (c) between GPD+ and ERA Interim WTCs. Plots (b) and (d) show the number of crossovers ("N. Xovers", blue) and the number of along-track (yellow) pairs used, per cycle, in the GPD-MWR and GPD-ERA analyses, respectively. To facilitate the analysis, both cycle number (bottom x-axis) and time (year, top x-axis) are used.**

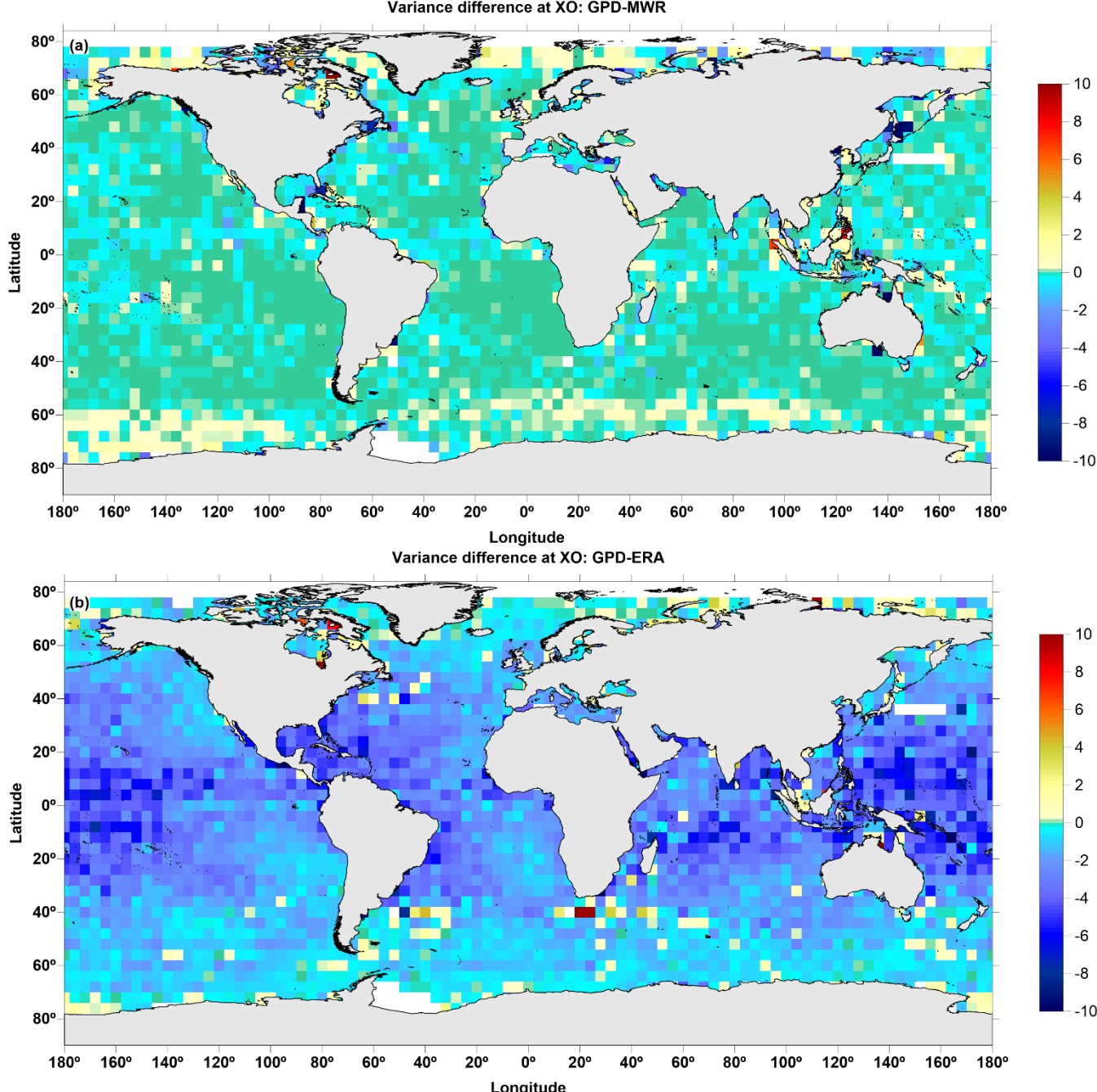

**Figure 12 Spatial distribution of the weighted SLA variance differences (in cm²) at crossovers (XO) between (a) GPD+ and the MWR-derived WTCs and (b) GPD+ and the ERA Interim WTCs for the whole Envisat period (cycles 006 to 113). The green colour represents SLA variance differences around zero. Pixels with no data are shown in white.**

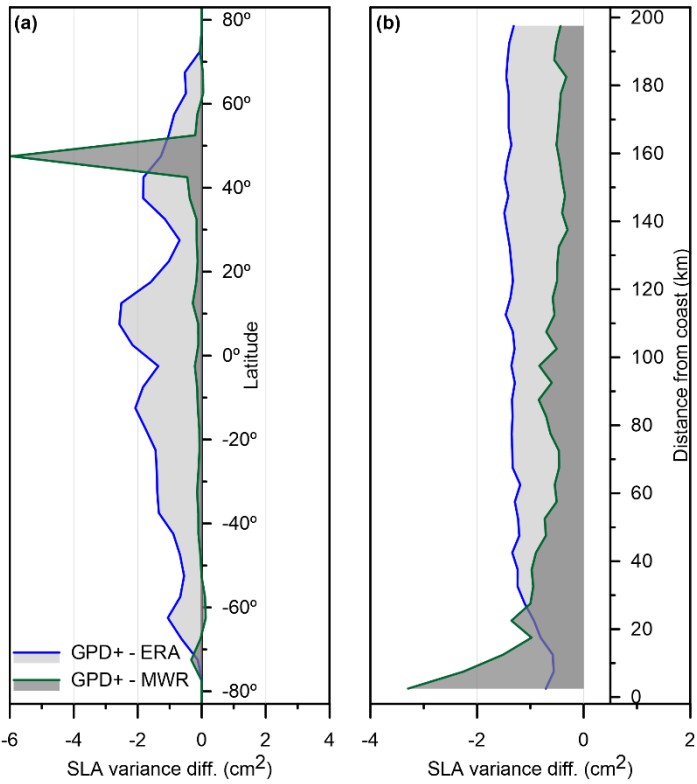

**Figure 13 Variance differences (cm²) of SLA versus latitude (a) and distance from coast (b) between GPD+ and ERA Interim WTCs (blue) and GPD+ and the MWR-derived WTCs (green) for Envisat cycles 006 to 113.**

1005

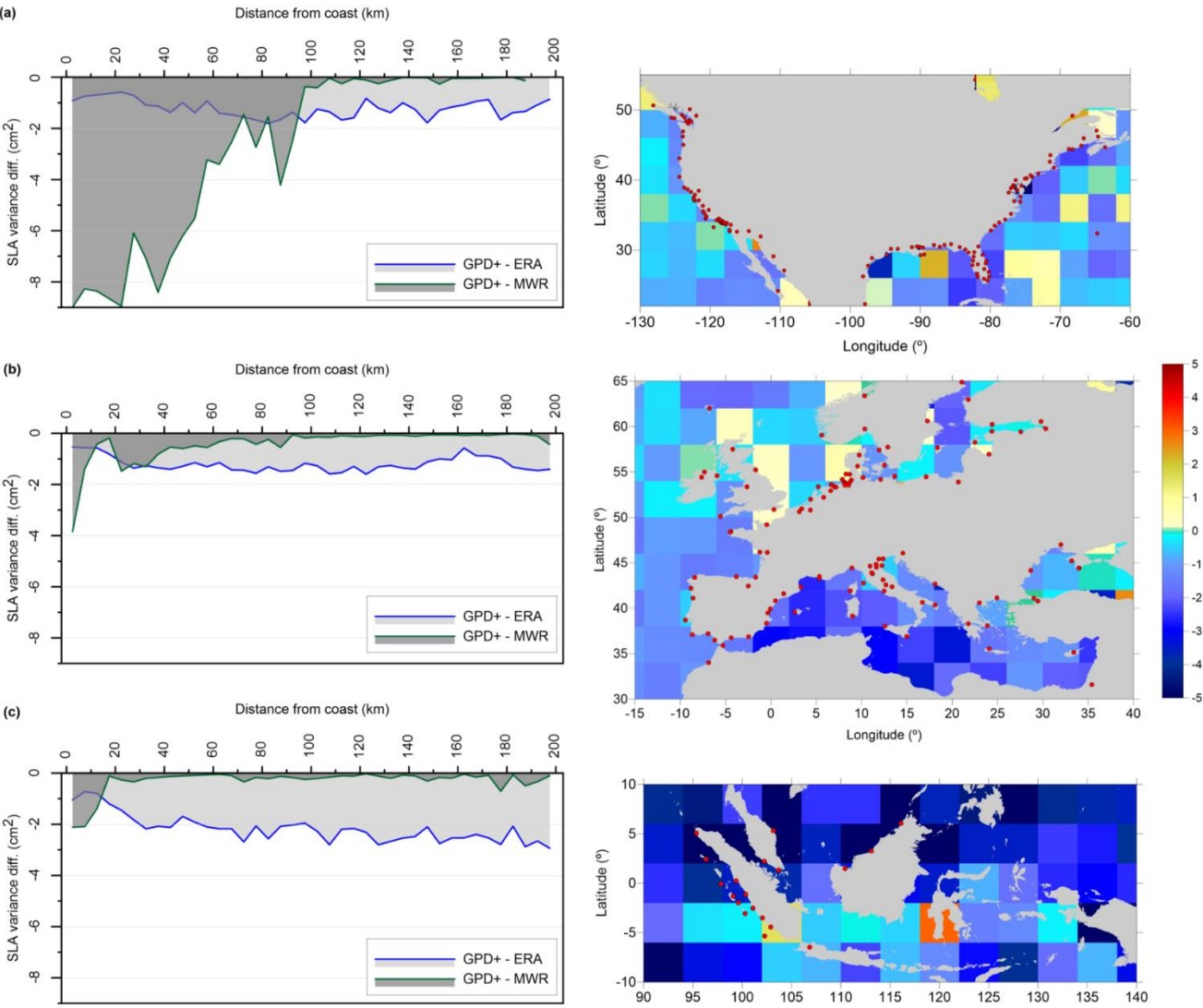

**Figure 14 Variance differences (cm²) of SLA function of distance from coast (left) between GPD+ and ERA Interim WTCs (blue) and GPD+ and MWR-derived WTCs (green) for the whole Envisat period (cycles 006 to 113) for North American coast (a), European coast (b) and Indonesia region (c). In the plot for the North American coast, the y-axis has been clipped to – 9 cm² (minimum value is around -13 cm²). Right panels show the spatial distribution of the weighted SLA variance differences (in cm²), computed along the satellite tracks, between GPD+ and ERA Interim WTCs. The green colour represents SLA variance differences around zero. The GNSS stations used in the computation of the GPD+ WTC are represented as red dots.**

1010