# Peer review of "A coastally improved global dataset of wet tropospheric corrections for satellite altimetry"

_Earth System Science Data, 2019_

## Referee Comment (RC1) · Anonymous Referee #1 · 18 Nov 2019

**MAJOR COMMENTS**

In this paper, the authors present a novel dataset of Wet Troposphere Correction (WTC) to correct the sea level anomaly (SLA) derived from satellite radar altimetry. The dataset is particularly important for coastal altimetry being well known that this correction is the most critical in the coastal zone. The new correction (known as GPD+) computes the Water Path Delay (WPD) for all along-track altimeter points where the default correction (from onboard radiometer) is unusable The method adopts an objective analysis approach to estimate WPD from a number of sources (coastal and island GNSS stations, satellites carrying microwave radiometers, valid on-board MWR measurements). The method is applied to all conventional missions and CryoSat-2. The validation of the dataset is made through statistical analysis of SLA and the metric used

is the reduction of variance.

The author provide a clear description of the datasets apart details specified below in the minor comments. However, the validation of the dataset (that is the part of interest to users) is really poor in showing the improvements, in particular with reference to the coastal zone. The authors titled this paper "A coastally improved global dataset...", unfortunately the reader does not see any zooming in the coastal zone. The metric used is certainly appropriate for open ocean but not for the coastal zone. The results do not provide a clear measure of confidence of the dataset in these challenges area. All plots are global and some plots globally averaged when quantities are showed as a function of distance. Instead, the reader expects to see a selection of relevant coastal regions in the world, based e.g. on bibliography (i.e. areas where users already applied coastal altimetry) or peculiar characteristics (e.g. authors mentioned Indonesia). Moreover, the testing in the coastal zone has to be at 20Hz being the available re-tracked products at this rate. The RADS product is fine for open ocean studies but not in the coastal zone. Also the metrics has to be different, as in the coastal zone we can use tide gauges as an independent measure of SLA. Therefore, by chaning the wet troposphere (default vs GPD+) in the altimetry formula, absolute differences of SLA along the track would show the distance of the coast at which noise increases in the specific region. Comparisons with TGs would show the improvements in terms of statistical indicators ( correlation and rms error).

Having said that there are other important remarks that I would like to highlight.

First, the authors are discussing a product at 1 Hz, when users need a product at 20 Hz in the coastal zone. So this product after publishing would not be usable for the typical non expert coastal users.

Second, I also see insufficient the strategy of showing results related to only one missions. As multi-mission approach is essential in the coastal zone to have more coverage in space and time, the reader expects to see the validation extended to all missions.

Third, one important input for the estimation of an improved WTC in the coastal zone is the presence of GNSS station. The authors provide poor information about distribution in space and time. There is just one figure related to Envisat showing the number of GNSS stations over mission time. The authors have to add same figures for the other missions. Moreover, a map has to add concerning the geographical distribution (areas well covered and areas where no GNSS stations are available. These figures are important for the users that after zooming in their coastal regions of interest can perceive the space and time coverage.

In summary, the paper in the actual version fails to convince the reader that in the coastal zone the new correction cannot be immediately exploited by users (because not at 20 Hz) and that misses a thorough validation in selected coastal areas of investigation (i.e. zooming locally where the user would use SLAs).

Therefore, the paper calls for significant revision in order to fill the gaps in term of exploitability of the product and validation of the correction in the coastal zone.

MINOR COMMENTS Pg 1, abstract: "The results are presented with vague sentences (e.g. GPD+ WTC is the most effective. . ...). The reader expects here to see quantitative results that show the improvement with reference to the state-of-the-art and discussion of these results. In the present version, the abstract is substantially an introduction to the dataset that should be the core with more details, e.g., distance from the coast, etc..

Pg. 1, rows 13-14, "SLA dataset over open ocean accurate to the centimetre-level": The authors in the previous sentences refer to sea level rise (which means mm/yr error level). The reader might be confused with cm level accuracy that is generally a target for oceanography. Moreover, accuracy is not enough for trends, there is also a need of "stability", and here it is the case of wet tropo not drifting over time. Please rephrase properly

Pg. 2, row 44, "with a centimetre-level radial error": Please provide reference where it

is demonstrated.

Pg. 2, row 44, "precise SSH": You used "accurate" before. It depends on what you refer, e.g., global mean sea level requires accuracy; fronts requires precision, etc.)

Pg. 2, row 55, "Chelton et al. (2001).": please refer to recent bibliography (Satellite altimetry over oceans and land surfaces (Detlef Stammer & Anny Cazenave Editors), Earth observation of global changes book series, CRC Press Taylor & Francis, London, UK, 670 pp, doi:10.1201/9781315151779, 2017).

Pg. 3, row 67, "as large as 2.3±0.2 m", is this cited in Fernandes et al. 2014? If not, please provide reference.

Pg. 3, row 67-68, "calculated with millimetre-accuracy, provided the surface atmospheric pressure is known at each location": as we are talking about coastal zone, the authors have here to specify that pressure has to be know at surface level. This pressure is generally retrieved from coarse models that can fail in steep coastal regions.

Pg. 3, row 69, "dry and wet tropospheric corrections (negative values)": why negative ? please explain.

Pg. 3, row 70, "DPD and WPD to the corresponding absolute values": What do you mean with "absolute"? what is the difference between DTC and DPD, WTC and WPD?

Pg. 3, row 73-74, "possessing an absolute value less than 0.50 m.": Please specify how 0,50 is estimated. Please also specify the meaningful of "absolute" vs "relative".

Pg. 3, row 73, "Contrasting": Maybe you mean "in opposite"

Pg. 3, row 79-86, "Radiometers .. 12 km": please explain the different impact of the three radiometers on the retrieved measurements, e.g. with reference with data quality.y Are there differences in the coastal zone in retrieving data ?

Pg. 3, row 88, "precise modelling": I think the word "modelling" is confusing. WTC can be derived from models too. However, here we are talking about "observations".

Pg. 3, row 91, "flagged as invalid, being therefore discarded, or non-existent due to several reasons.": The sentence is vague. Why data are flagged invalid? What is the criteria used ? What re the reasons for missing data ? please explain

Pg.4, row 96, "surface emissivity": Coastal zone has also non homogeneous scattering due to variable waves, winds, surfactant streaks, etc. Are they influencing the retrieval of a valid measurement?

Pg. 4, row 105, "is to describe and grant access": The access to a dataset cannot be an aiming of a paper. I think the authors have to reformulate clearly the main goal of this paper that is presenting and validating a dataset and then elucidate specific single objectives

Pg. 4, rows 111-115, "The main objective": Objectives have to be stated in the introduction. Also description of sections has to be moved in the introduction.

Pg. 4, row 118, "GNSS network of stations", please provide a map of GNSS stations used so the reader can appreciate the global coverage

Pg. 4, row 123, "This way", please add "In"

Pg. 5, row 118, "this way are given at station height". The GPS stations are over land. So you measure the column at land point. It is not clear to me (and probably to most of not expert people) how this value is extrapolated to the ocean

Pg. 6, row 156, "In fact, GPD+ is an upgrade from the GPD methodology": Please better clarify differences between GPD and GPD+. Apparently you say that GPD+ was for coastal zone but now global. Is the reason related to CryoSat-2 ? as it has no radiometer onboard.

Pg. 5, row 158-164: as you provide a table it is redundant here to report names of the mission. It is important to add space and time resolution of single MWR sensors in the table. A matrix has to be added showing the MWR sensors available for each altimetry mission. Again, this is an important figure for the reader. Some comments about
substantial differences between sensors should be recalled here from cited references

Pg. 5, row 173-176, "It is known that, in addition to TCWV, WPD also depends on temperature. Expressions such as Eq. (3) account for an implicit modelling of this dependence. Fernandes et al. (2013b) have shown that this expression leads to similar results as those obtained by adopting formulae that make use of explicit values of atmospheric temperature given e.g. by an NWM." The reader might not understand what you mean here with "Implicit" and "explicit" values. Please show examples of comparisons with WTC derived from NWPs in open ocean and in coastal zone.

Pg. 6, row 179-180, "We recall that the WTC is the symmetric of the wet path delay and the quantity of interest in satellite altimetry" Please rephrase and specify what you mean with "symmetric"

Pg. 6, row 180, "RA data necessary to compute", Please specify the sources you used for corrections, orbit, MSS, etc.

Pg. 7, row 191, "Threshold values used in this criterion depend on the RA mission": Please specify thresholds

Pg. 7, row 194, "at distances from coast": The authors use some editing criteria. I am curious to know what happens when tracks are parallel to the coast , but also some situations, e.g., Indonesia where the altimeter crosses successive land segments due to presence of closest islands.

Pg. 7, row 203, "number of 18 Hz measurements to compute the 1 Hz": is the global product at 1 Hz (i.e. around 7 km spaced for all missions)? While in open ocean it makes sense, I am bit skeptical the coastal zone might benefit from this product if not provided at 18/20 hz. It has been demonstrated that we need high resolution data in the coastal zone (and in fact waveforms are retracked at that rate and SLAs computed at that rate). Otherwise, the user will not be able to exploit the product.

Pg. 7, row 203-204, "For approximately 10% of all oceanic points": What do you mean

with "oceanic domain ? Does it include coastal zone ? at which distance ? The value seems for Envisat only. What about the other missions?

Pg. 8, row 220, "parameters have been obtained for Envisat": Please provide parameters for all missions

Pg. 8, row 240, "For all satellite missions but CryoSat-2 and for each along-track point deemed as invalid": The sentence is unclear, please rephrase

Pg. 9, row 275, "50 km from the ocean": The setting of this value has to ne justified

Pg. 9, row 278, "Figure 4 gives an example of the GPD+ WTC for Envisat's cycle 12". I don't understand the message of this figure. The upper map is substantially unreadable. The lower map is not providing information as the reader would like. Moreover, one cycle per one mission would be only for visual purposes. There is no comments in the paper. The reader expects quantitative results about the improvement. Pg. 10, row 289, "respectively, are provided at 1 Hz.". Previously, the authors mentioned 20 Hz. People using the product in the coastal zone need 20 hz data. I don't understand the utility of publishing a product that then in practice it is not usable from coastal zone users (who are not experts in altimetry). The authors refer to RADS that cannot be considered a "coastal altimetry product". In my opinion, the authors have to satisfy the user requirements if they want to publish this dataset.

Pg. 11, row 315-318: "For results concerning algorithm.": The reader is confused here and reminded to previous paper. Indeed, the reader wants to see statistics of all missions here with the application of the algorithm described here. The authors have to add relevant statistics of all missions.

Pg. 11, row 320, "The GPD+ WTC is here compared to the ECMWF Reanalysis WTC": This kind of comparison make sense in open ocean but not in the coastal zone. The authors provide a title "A coastally improved global dataset…..". They clearly state previously that models fail in the coastal zone and now they use for validation.

Pg. 11, row 335, "Figure 5 shows the GPD+ WTC for some Envisat tracks": The reader expects to see the map showing where the passes are located and identification of the segments where the new corrections improves. The discussion of Figure 5 is not provided. The plots have to be commented in relation to the places touched over ground.

Pg. 11, row 340, "interesting results": please remove being subjective

Pg. 11, row 346, "most of these points are located at high latitudes and in coastal regions": This statement is not demonstrated in the figure. The authors expects to see zooming in coastal regions to see improvements.

Pg. 11, row 361, "for the whole Envisat mission": the authors have to provide th same figure for the other missions too

Pg. 12, row 369, "The results are shown in Fig. 7". The authors state the product is at 1 Hz (7 km) and in the plot show values at less than 5 km

Pg. 13, row 393-395, "Therefore, three SLA datasets of collocated along-track points were derived using the same standard corrections (Sect. 1) but the WTC, which can be the Composite correction present in AVISO CorSSH L2P products (Comp), the GPD+ or the ERA Interim WTCs.". This comparison makes sense only in the open ocean and not in the coastal zone (0-50 km)

Pg. 13, row 406, "Fig.8a": Fig. 8c si not commented in the text. Moreoer, there is a strange behavior around cycle 95

---

## Referee Comment (RC2) · Anonymous Referee #2 · 23 Apr 2020

The paper presents a dataset of wet tropospheric correction applicable to altimetry and the methodology used to product it. The wet tropospheric correction is one of the correction applied to the altimeter range to compute the Sea Level Anomaly. The WTC is traditionnaly provided by on-board microwave radiometer, measuring in appropriate frequencies bands to correct for the excess path delay. The estimation of the WTC from the MWR measurements can be degraded by extreme rains events, ice surface, land contamination in coastal areas, instrument malfunctions. The author proposes a method named GPD+ that intend to improve the MWR-based WTC of operational processing, or propose a correction for mission without MWR on-board (Cryosat-2).

The method consists first in the filtering of the invalid WTC estimation from the operational product and second, by the estimation based on the objective analysis using

external data such as GNSS data, MWR Imaging data (providing water vapour), Numerical Weather Prevision model (ECMWF, ERA interim). The method is applied to almost all conventional missions.

The dataset used for the algorithm is rather well defined. The section about the GNSS dataset lack a discussion about the coverage of this network. Although the paper states otherwise, GNSS stations don't seem to be distributed globally over the globe. The section about the Imaging radiometer seems more dedicated to the filtering step of the method than to the description of the input dataset itself and the added value of this dataset. The NWP dataset is slighlty described, and lack a discussion of the difference between ERA Interim and ECMWF. The paper don't say if one mission can be covered by only one model or if two are needed, if there is a bias between the two models that shall be corrected. Also the paper stated that NWM data are provided as output from GPD+ for northernmost latitudes, but the method to adjust the model to measurements is not clearly defined (is it a simple bias computed over each cycle?).

The algorithm is well described and the workflow provides a clear overview of the processing. The method is assessed for Envisat only in this paper. The paper introduces Full Mission Reprocessing (FMR v3.0) but compare GPD+ dataset to the Composite Correction extracted from L2P products issued from an older reprocessing (FMR v2.1). According to the L2P product handbook available on AVISO, there is no composite correction in the L2P products, only MWR-derived correction. This point must clarified. In the validation section, both corrections (MWR-based and composite) are used for comparisons. it is difficult to follow which version of the Envisat MWR-based correction is used for the generation of the GPD+ and which one is compared to the GPD+.

The number of 30% of invalid WTC data over ocean for Envisat is stated but not justified. This number seems quite high. Moreover, the criteria to select valid SLA points is not discussed. The paper shall define the criteria of validity of the SLA. The L2P products provide a validity flag that could be used. The comparison of GPD+ with GNSS is more a validation of the method than a performance assessment. This section can
gain in clarity in the method used for this comparison. GNSS data are not independant of GPD+ data. Is the GNSS data cited in this section also used in the generation of the GPD+ dataset (data from another network for example)? In the first sections, one of the criteria for rejection of MWR-based correction is the distance to coast, but it is not clear if this criteria is used in this section. The fact that the method is not clear makes the figure 7 difficult to understand.

The paper provides a performance assessment (and not accuracy) of the section 3.2 using analyses of Sea Level Anomaly variances. In this section, the author compares the composite correction which is not part of the L2P products to the GPD+ correction. The method used is to select all valid SLA points, and for the points with the composite outside limits or invalid, the ERA interim WTC value is used. These points shall be discarded from the analysis as they do no represent a fair comparison with the MWR-based correction as the use of the model correction will degrade it. The previous sections have already shown that the GPD+ retrieve some invalid points. The color scales for figure 9 is not well chosen and is difficul to read. Figure 10a shows a strong peak, with not physical values, for latitude around 50°N that is not explained in the paper. For this diagnosis, it is not stated if data cover open-ocean only, or ocean and coastal areas. Moreover the figure shows a reduction of the SLA variance from 200km up to the cost but it is difficult to see the improvement close to the coast.

Although the paper title is "A coastally improved global dataset", there is no real focus on coastal areas. It is not stated clearly in the paper but it seems that the dataset is based on 1Hz data where 20Hz data are more adequate for studies on coastal areas.

Minor comments

Row 118: "GNSS network of stations distributed globally along the coastlines": GNSS stations don't seems to be distributed globally over the globe. A map could be added to show the position of the GNSS stations used for the generation of the dataset.

Row 195: "values of 15km have been used for Jason-1/2/3": 15km seems quite small

for this serie of MWR knowing that they measure at three frequencies, including a 18.7GHz with a large footprint. What is the reason for that?

Row 201: Talking about Envisat datat from latest reprocessing, the author states "30 % of the oceanic points have an invalid WTC value": This seems quite a large number of invalid points when focusing on ocean surface only (with a valid SLA). From Figure 2, it does not look like one third of the points are invalid. How do you explain that number?

Row 209: the author states "Data from the reference missions". For a non-specialist audience, the author should explain which are the reference missions.

Row 219: for the intercalibration processing, the difference at cross-over points with a time-lag of 180 minutes between reference missions and other altimetry missions are computed. Is that time span not too large for WTC ?

Row 229: "In addition, to reduce data discontinuities, ...." : from this sentence it seems that a bias is computed between the MWR and the NWM correction for each cycle. What is the rationale for a simple bias? How is computed that bias?

Row 274-275: "To prevent the loss of points when interpolating to 20 Hz points, in addition to ocean points, the closest land point is included, provided it is within a distance less than 50 km from the ocean." Can you clarify the processing here? what is the closest land point?

Row 320: ""The GPD+ WTC is here compared to the ECMWF Reanalysis WTC (ERA Interim, GDR field mod_wet_tropo_cor_reanalysis_01) and with the WTC present in the AVISO CORSSH L2P products in July 2019 (AVISO, 2017). The latter dataset is usually called Composite Correction ". You state here that you compare the GPD+ to Composite correction, but latter (line 334). But according to the L2P products handbok (https://www.aviso.altimetry.fr/fileadmin/documents/data/tools/hdbk_L2P_all_missions_except_S3.pdf), there is no composite correction in these products. And latter, the author says that he used the field 'rad_wet_tropo_cor_sst_gam_01'. This point shall be clarified

Row 330: "Anomalies in this field have been found, with the field out of limits in a set of points, most of them concentrated on certain passes," : Do you mean that you found anomalies in the ERA interim product for WTC field?

Row 334: "The MWR-based correction used in the generation of these files" : Which files?

Row 342: The author found 30% of points with a rejected MWR-derived WTC. This figure seems quite large. It could be interesting to discuss that number and provides some insights of the repartition within the different causes. It seems this number is estimated over ocean. Does it include coastal regions? Which latitudes?

Row 362: "Only GPD+ estimates retrieved using observations are selected." which observations? MWR? GNSS?

Row 370-376: Methodology difficult to understand

Row 381: "On the contrary": -> Moreover, Additionaly . . .

Row 384: "Accuracy assessment" ==> Performance assessment

Row 420: "third party data": what are those third-party data?

Figure 5: b) and c) look quite similar with land/ice contaminated pass. Outliers are not obvious in c).

Figure 7: why is there an increase in the number of points for the GNSS-GPD+ comparison but not for the GNSS-MWR one?

Figure 9: The green color cannot be seen on the color scale.

Figure 10: What is this peak around latitudes 50°N?

---

## Author Comment (AC1) · 14 Jul 2020

We would like to start by thanking Reviewer#1 for his/her valuable contribution, for the many useful suggestions and corrections, and for his/her constructive comments that led to the improvement of the quality of the manuscript.

The main changes introduced in the manuscript following the comments and suggestions of the two Reviewers can be summarised as:

- Sections containing the Abstract and the Conclusions have been updated to accommodate the new results presented in the revised version of the manuscript.
- Some parts of the text have been moved to new sections or were rewritten/completed to be clearer and more informative.
- Figures 1 as well as figures 11, 12 and 13 have been updated, the latter to include the results for the comparison of the GPD+ WTC with the MWR-derived WTC, instead of that for the Comp WTC, following the concerns raised by Reviewer#2.
- Previous Figure 5 has been divided into Figures 7 and 8 and the geographic location of the Envisat tracks have been added, following the recommendation of Reviewer#1.
- New figures have been added to the revised version (Figures 2, 3 and 14).
- Tables 1 and 4 have been updated, the former to include more information, the latter in the sequence of the last update of the GPD+ database (performed to include more data for the recent missions).
- A new table (Table 2) has been added in the revised version.
- All figures and tables have been renumbered.
- Section 3.2 has been divided into sections 3.2.1 and 3.2.2 describing the global and the regional (coastal) results, respectively, and the text has been extended.
- Reference Vieira et al. (2019c) has been updated, since at the time of this revision it has already been published.
- Reference AVISO (2017) has been removed.
- Five new references (Bevis et al. (1994), Rudenko et al., (2017), Valladeau et al. (2015), Dinardo et al. (2020) and Escudier et al. (2017)) have been inserted in the revised version.

Our point-by-point responses to Reviewer#1 are presented below.

MAJOR COMMENTS

In this paper, the authors present a novel dataset of Wet Troposphere Correction (WTC) to correct the sea level anomaly (SLA) derived from satellite radar altimetry. The dataset is particularly important for coastal altimetry being well known that this correction is the most

critical in the coastal zone. The new correction (known as GPD+) computes the Water Path Delay (WPD) for all along-track altimeter points where the default correction (from onboard radiometer) is unusable The method adopts an objective analysis approach to estimate WPD from a number of sources (coastal and island GNSS stations, satellites carrying microwave radiometers, valid on-board MWR measurements). The method is applied to all conventional missions and CryoSat-2. The validation of the dataset is made through statistical analysis of SLA and the metric used is the reduction of variance.

The author provide a clear description of the datasets apart details specified below in the minor comments. However, the validation of the dataset (that is the part of interest to users) is really poor in showing the improvements, in particular with reference to the coastal zone. The authors titled this paper "A coastally improved global dataset. . .", unfortunately the reader does not see any zooming in the coastal zone.

R.: The paper's aim was to present and describe the GPD+ WTC dataset/database. The authors' intention was to show the improvement in the SLA description in the coastal zone when using the correction in global terms, therefore with no focus on a particular region. The authors refer two papers that show in detail the improvement in the SLA signal description in the coastal region (German Bight and the Indonesia Archipelago) when the GPD+ WTC is used. However, in response to the Reviewer's comment, results for three coastal regions, selected on the one hand, due to the large number of available GNSS stations (North American and European coasts) and, on the other hand, due to the fact of being a challenging region for coastal satellite altimetry (Indonesia region), have been added to the revised version of the manuscript, as an attempt to show the potential of the GPD+ dataset along the coastal waters. Section 3.2.2 has been added in the manuscript to present these results.

The metric used is certainly appropriate for open ocean but not for the coastal zone. The results do not provide a clear measure of confidence of the dataset in these challenges area.

R.: The assessment of the performance of the GPD+ WTC dataset is made using statistical analyses of sea level anomaly variability. The reduction of SLA variance is a metric commonly used to assess the performance of a correction against its counterparts available in the altimetry products accessible to the user. The larger the SLA variance reduction, the better the correction, since its application will lead to an SLA whose variability is more likely to be due to oceanic conditions than to the error in the correction(s). The metric is used to assess the performance of the dataset and not to validate it. Observations adequate to validate the GPD+ WTC dataset over coastal regions are not of sufficient quantity and quality. The vertical distribution of water vapour in the troposphere can be obtained using data from a network of radiosondes. However, datasets from radiosondes possess undesirable inhomogeneities (e.g., vertical range, vertical resolution, temporal regularity, poor continuity), have poor spatial coverage, particularly over coastal zones, and are not collocated with altimeter measurements. Therefore, completeness of observations lacks in these regions.

For these reasons, the authors also performed an assessment of the GPD+ WTC performance using GNSS observations, as explained in Section 3.1 of the manuscript. Former Figure 7 (now Figure 10) illustrates the results, showing that the RMS of the differences GNSS-MWR increases when approaching the coast. On the contrary, the RMS of the differences GNSS-GPD+ decreases and this result is thought to be a clear indicator of the performance of the GPD+ correction.

All plots are global and some plots globally averaged when quantities are showed as a function of distance. Instead, the reader expects to see a selection of relevant coastal regions in the world, based e.g. on bibliography (i.e. areas where users already applied coastal altimetry) or peculiar characteristics (e.g. authors mentioned Indonesia).

R.: The main objective of this paper is to present the GPD+ WTC database to users of the Geophysical or Level 2 altimetry products, i.e., users mainly interested in ocean applications yet wishing to extend their analysis to the coastal regions. Therefore, the authors had opted to show the results for the latest Envisat reprocessing (which have not been published yet), summarised globally and to refer previous published results that have used GPD+ and focused on particular regions (Handoko et al., 2017; Dinardo et al., 2018). However, in response to the reviewer's comments, we have extended the Results section and provided some results for the three coastal regions already referred. Section 3.2.2 has been included in the revised manuscript.

Moreover, the testing in the coastal zone has to be at 20Hz being the available re-tracked products at this rate. The RADS product is fine for open ocean studies but not in the coastal zone.

R.: The rate of the altimetry measurements is not a limitation to the GPD+ methodology. In the scope of a current research project in which the University of Porto (UPorto) is involved, the GPD+ methodology will be used to estimate the WTC for the coastal (and inland water) zone for CryoSat-2 and Sentinel-3 missions. The outcome of this project will be a GPD+ WTC product at high rate (20 Hz), intended to be used for applications over the coastal zone (i.e., no ocean values included for distances larger than ~100 km off the coast) and over inland waters. However, the GPD+ WTCs presented in this manuscript have been computed to be incorporated in altimetry products providing observations at 1 Hz, the rate still most used by the altimetry community databases. They are intended for users who want to have a consistent and continuous WTC correction, from open ocean to coasts (and polar regions as well). The correction can be extended to the coast since a valid WTC value is provided for the first along-track measurement over land. Users can therefore use this measurement to interpolate the valid GPD+ WTC up to the coast, for the location and time instant of the 20 Hz data. Moreover, as the onboard radiometer data are not available at a higher than 7 Hz rate, neither these data nor the third-party data have enough resolution to be provided at 20 Hz. Therefore, and for the time being, the strategy for those users who want to focus on coastal zones, would be to interpolate these 1-Hz data to the location and time instant of the 20 Hz data.

For high-frequency MWR, expected in the future, high-rate WTC are definitely advisable, and the authors intend to exploit this possibility. A sentence has been added to the "Conclusions".

Also the metrics has to be different, as in the coastal zone we can use tide gauges as an independent measure of SLA. Therefore, by chaning the wet troposphere (default vs GPD+) in the altimetry formula, absolute differences of SLA along the track would show the distance of the coast at which noise increases in the specific region. Comparisons with TGs would show the improvements in terms of statistical indicators ( correlation and rms error).

R.: Some analyses previously performed by the authors seem to indicate that comparison with tide gauges is not the best way to validate the WTC, because the differences between

altimetry and tide gauges are large when compared to the SLA variability due to different WTC. For this reason, in this paper, the analysis suggested by the reviewer has been performed using GNSS data. Former Figure 7 (Figure 10 in the revised version) shows an independent comparison between GNSS-derived and MWR-derived WTC, function of distance from coast, for the newly reprocessed Envisat data. This result yields the distance from coast at which this contamination appears. This distance depends on the altimetric mission, due to their different footprint sizes and different MWR retrieval algorithms, varying from 10 to 30 km. For Envisat, this distance is 30 km. This assessment provided us the distance from coast at which an MWR measurement is expected to be contaminated by land. This means that the GPD+ methodology does not use MWR-derived WTC in the last 30 km to the coast, even if they are not flagged as invalid by other rejection criteria, to prevent land contamination in the GPD+ estimates.

We believe that the assessment with GNSS, together with the SLA variance analysis, are clear and sufficient indicators of the GPD+ performance.

Having said that there are other important remarks that I would like to highlight.
First, the authors are discussing a product at 1 Hz, when users need a product at 20 Hz in the coastal zone. So this product after publishing would not be usable for the typical non expert coastal users.
R.: As previously said, these products contain a consistent and continuous WTC correction at 1 Hz rate. Improved criteria have been established in the GPD+ methodology for each mission (e.g. criteria derived from statistical analyses to detect measurements contaminated by ice, land, rain and outliers, based not only on the information available on the points for which a GPD+ estimation is being computed, but also on the information available for neighbouring points) and applied to detect valid/invalid MWR measurements, besides the criteria based on the flags provided in the GDR/RADS/PEACHI products. Moreover, the correction has been calibrated against the SSM/SSMI radiometers, ensuring the long-term stability of the GPD+ WTCs.
As previously mentioned, the correction has been provided continuously over ocean and coastal regions, precisely to be used by non-expert users, working over ocean but wanting to extend their analysis to the coastal regions, without discarding altimeter measurements in the coastal zone as would happen when relying on MWR-derived measurements to compute SLA. As already explained, the correction provided at 1 Hz can be interpolated to 20 Hz data by expert users with enough accuracy. As explained before, this is due to the characteristics of current on-board radiometers.
Typical non expert coastal users who are not able to perform this interpolation procedure and in the absence of MWR data in the coastal strip can rely on a Numerical Weather Model (NWM) derived WTC. However, anomalies in the NWM-derived WTCs have been found in the Envisat FMR V3.0, which are corrected in the GPD+ processing. A sentence clarifying this has been added do the text: "Anomalies in this field have been found, with the field out of limits in a set of points, most of them concentrated on certain passes. This is due to the fact that this correction has been computed from 3D model fields at the altimeter measurement altitude. Therefore, whenever the altimeter-derived surface height is not set (NaN value), the corresponding Model WTC will also be NaN. As our goal is to be able to provide continuous WTC, without data gaps, this field is unsuitable for use in the GPD+ estimations.". Moreover,

as described in this manuscript, NWM-derived WTC are not able to describe the small-scale variability of this field yet.

Second, I also see insufficient the strategy of showing results related to only one missions. As multi-mission approach is essential in the coastal zone to have more coverage in space and time, the reader expects to see the validation extended to all missions.

R.: The primary scope of this paper is the dissemination of the GPD+ database fostering its use among as many people as possible, since in the authors' opinion the GPD+ database is of sufficient quality for both expert and non-expert users. Therefore, the paper focuses mainly on the data description and their usage, and for this reason the Earth System Science Data (ESSD) journal has been chosen. The authors have tried to show the added value of the correction using the results for the newly recomputed Envisat FMR V3.0 data, not yet published before. Results for other missions have been already published by the authors in papers of more scientific nature (cf. references e.g. Fernandes et al. (2015) for results regarding reference and ESA missions, Fernandes and Lázaro (2016) for results for CryoSat-2 and GFO, Fernandes and Lázaro (2018) for Sentinel-3). These works are cited in the manuscript leading the readers to the reference list. The GPD+ WTC are available for all altimetry missions in the UPorto database, except for Sentinel-3A/B as their development is still on course and can therefore be chosen for a multi-mission approach.

Third, one important input for the estimation of an improved WTC in the coastal zone is the presence of GNSS station. The authors provide poor information about distribution in space and time. There is just one figure related to Envisat showing the number of GNSS stations over mission time. The authors have to add same figures for the other missions. Moreover, a map has to add concerning the geographical distribution (areas well covered and areas where no GNSS stations are available. These figures are important for the users that after zooming in their coastal regions of interest can perceive the space and time coverage.

R.: Figure 1 shows the number of GNSS stations used as input in the GPD+ algorithm, function of time. This information does not depend on the mission and therefore one single figure suffices to illustrate that the later the period of the mission, the larger the number of available GNSS stations. Figure 1 has been remade to include information for the whole satellite altimetry era, so the reader can more easily understand this and the Envisat period (2002-2012) is shaded in the figure.

However, the reviewer is right in indicating that the geographical location of the GNSS stations could be of interest to the reader and a new figure (Figure 2) has been added to the revised version of the manuscript.

In summary, the paper in the actual version fails to convince the reader that in the coastal zone the new correction cannot be immediately exploited by users (because not at 20 Hz) and that misses a thorough validation in selected coastal areas of investigation (i.e. zooming locally where the user would use SLAs). Therefore, the paper calls for significant revision in order to fill the gaps in term of exploitability of the product and validation of the correction in the coastal zone.

R.: The authors consider that the Reviewer's comments have undoubtedly improved the paper and therefore are thankful for his/her contribution. The authors expect to have satisfactorily responded to the critiques and/or suggestions raised by the Reviewer.

MINOR COMMENTS

Pg 1, abstract: "The results are presented with vague sentences (e.g. GPD+ WTC is the most effective. . ...). The reader expects here to see quantitative results that show the improvement with reference to the state-of-the-art and discussion of these results. In the present version, the abstract is substantially an introduction to the dataset that should be the core with more details, e.g., distance from the coast, etc..

R.: The ESSD journal encourages the submission of manuscripts describing original research datasets that use can be considered beneficial to Earth system sciences. Therefore, the authors focused on the description of the GPD+ dataset. However, the abstract has been rewritten according to the reviewer's suggestion and quantitative results have been added.

Pg. 1, rows 13-14, "SLA dataset over open ocean accurate to the centimetre-level": The authors in the previous sentences refer to sea level rise (which means mm/yr error level). The reader might be confused with cm level accuracy that is generally a target for oceanography. Moreover, accuracy is not enough for trends, there is also a need of "stability", and here it is the case of wet tropo not drifting over time. Please rephrase properly

R.: Lines 8 to 14 of the abstract have been rephrased and moved to Section 2.1.4 (Radiometer Calibration), to simplify the Abstract (since new information has been added to describe quantitative results, following the Reviewer's suggestion) and because this information is relevant to understand the need for performing the inter-calibration of the radiometers.

Pg. 2, row 44, "with a centimetre-level radial error": Please provide reference where it is demonstrated.

R.: The following reference has been added:

S. Rudenko, K. Neumayer, D. Dettmering, S. Esselborn, T. Schöne and J. Raimondo, "Improvements in Precise Orbits of Altimetry Satellites and Their Impact on Mean Sea Level Monitoring," in *IEEE Transactions on Geoscience and Remote Sensing*, vol. 55, no. 6, pp. 3382-3395, June 2017, doi: 10.1109/TGRS.2017.2670061.

Pg. 2, row 44, "precise SSH": You used "accurate" before. It depends on what you refer, e.g., global mean sea level requires accuracy; fronts requires precision, etc.)

R.: The suggestion has been accepted and the word "precise" has been replaced by "accurate".

Pg. 2, row 55, "Chelton et al. (2001).": please refer to recent bibliography (Satellite altimetry over oceans and land surfaces (Detlef Stammer & Anny Cazenave Editors), Earth observation of global changes book series, CRC Press Taylor & Francis, London, UK, 670 pp, doi:10.1201/9781315151779, 2017).

R: Chelton et al. (2001) is still an important reference. The following reference has been added:

Escudier, P., Ablain, M., Amarouche, L., Carrère, L., Couhert, A., Dibarboure, G., Dorandeu, J., Dubois, P., Mallet, A., Mercier, F., Picard, B., Richard, J., Steunou, N., Thibaut, P., Rio, M.-H., and Tran, N.: Satellite radar altimetry: principle, accuracy & precision, in: Satellite Altimetry Over Oceans and Land Surfaces, edited by: Stammer, D. and Cazenave, A., CRC Press Taylor & Francis, London, UK, 670 pp, doi:10.1201/9781315151779, 2017

Pg. 3, row 67, "as large as 2.3±0.2 m", is this cited in Fernandes et al. 2014? If not, please provide reference.

R.: Yes, these values are given in Fernandes et al. (2014).

Pg. 3, row 67-68, "calculated with millimetre-accuracy, provided the surface atmospheric pressure is known at each location": as we are talking about coastal zone, the authors have here to specify that pressure has to be know at surface level. This pressure is generally retrieved from coarse models that can fail in steep coastal regions.

R.: For oceanic coastal points, the DTC must be computed at sea level from sea level pressure data. If the DTC is provided in the altimetric products at the level of the model orography, as is usually the case, which can depart significantly from sea level at coastal zones, then the value of the dry tropospheric correction should be corrected as described in Fernandes et al. (2013a) (cited in this manuscript). In the same paper, it is also shown that current models are accurate enough to compute the DTC with this accuracy, including the coastal zones, provided adequate procedures are adopted.

The sentence "calculated with millimetre-accuracy, provided the surface atmospheric pressure is known at each location" has been changed to "over the ocean it can be calculated with millimetre-accuracy, provided the sea level atmospheric pressure is known at each location".

Pg. 3, row 69, "dry and wet tropospheric corrections (negative values)": why negative ? please explain.

R.: The measured distance between the satellite and the sea surface, or altimeter range $R_{obs}$, is computed from the following equation, neglecting atmospheric refraction:

$$R_{obs} = c\,\frac{\Delta t}{2}$$

where $c$ represents the velocity of light in vacuum and $\Delta t$ is the two-way travel time of a radar pulse between the satellite antenna and the sea surface. The velocity of the altimeter pulses is reduced in a refractive medium as the atmosphere. Therefore, when the signal passes through the troposphere, the propagation velocity of the altimeter pulses is smaller than $c$. This means that the $R_{obs}$ computed from equation above will be longer than the true range. To correct for this overestimation in the measured range, both the DTC and the WTC are negative.

Pg. 3, row 70, "DPD and WPD to the corresponding absolute values": What do you mean with "absolute"? what is the difference between DTC and DPD, WTC and WPD?

R.: Following the previous answer, we can define the effect of the troposphere on the altimeter signals, which appears as an extra delay in the measurement of the signal traveling from the satellite to receiver, as the tropospheric path delay, which can be divided into the dry and wet components, called the dry path delay (DPD) and the wet path delay (WPD), respectively. Each delay component contributes to an error (path length) in the measured distance that must be corrected for. The corrections needed to consider these delays – the dry tropospheric correction (DTC) and the wet tropospheric correction (WTC) – have therefore the same magnitude as the DPD/WPD and the opposite (negative) sign, and must be subtracted from the range estimated assuming the free-space value for the speed of light.

Hereupon, the term "absolute" is used to refer to the modulus of the DTC and WTC.

Pg. 3, row 73-74, "possessing an absolute value less than 0.50 m.": Please specify how 0,50 is estimated. Please also specify the meaningful of "absolute" vs "relative".

R.: As explained in the previous response, the term "absolute" is used to refer to the modulus of the corrections, which are negative.

In the computation of the water vapour range correction, passive microwave estimates of columnar water vapour from satellite radiometers are used. The maximum value of 50 cm for the wet path delay is known from decades of observations from satellite passive microwaves (e.g., Special Sensor Microwave/Imager (SSM/I) on board the United States Air Force Defense Meteorological Satellite Program (DMSP) satellites). Considering the global dynamic range of columnar water vapor, 0.5–7 g cm$^{-2}$, the wet tropospheric path delay varies from 3 to 45 cm, with standard deviations covering the range from 3 to 6 cm. A very thorough description of the underlying theory and principles of the wet tropospheric correction estimation can be found in:

Chelton, D. B., Ries, J. C., Haines, B. J., Fu, L. L., Callahan, P. S.: Satellite Altimetry. In Satellite Altimetry and Earth Sciences: A Handbook of Techniques and Applications; Fu, L.L., Cazenave, A., Eds.; Academic: San Diego, CA, USA; Volume 69, 1–131, 2001.

Pg. 3, row 73, "Contrasting": Maybe you mean "in opposite"
R.: Accepted and changed.

Pg. 3, row 79-86, "Radiometers .. 12 km": please explain the different impact of the three radiometers on the retrieved measurements, e.g. with reference with data quality.y Are there differences in the coastal zone in retrieving data ?

R.: According to the literature, and as mentioned in the paper, radiometer footprints depend on instrument and frequency. So, footprint size is the key factor in the coastal zone. In addition to the known footprint of each radiometer and according to several analyses performed by the authors (e.g., Fernandes et al., 2015, Fernandes and Lázaro, 2016) land contamination for these missions occurs at distances from coast less than 15 km, while for ESA's missions, T/P and GFO, this value is around 30 km. All this information is given in the paper.

Pg. 3, row 88, "precise modelling": I think the word "modelling" is confusing. WTC can be derived from models too. However, here we are talking about "observations".

R.: The suggestion has been accepted. The word "modelling" has been replaced by "estimation".

Pg. 3, row 91, "flagged as invalid, being therefore discarded, or non-existent due to several reasons.": The sentence is vague. Why data are flagged invalid? What is the criteria used ? What re the reasons for missing data ? please explain

R.: The reasons why the microwave measurements are flagged as invalid in the coastal zones are given in the sentences that follow the referred one. In the coastal region, the measurements of the MWR are in general contaminated by land, due to the large diameter of the footprint of the instrument. The WTC retrieval algorithms are based on sea surface emissivity conditions, which is valid only for open-ocean conditions since surface emissivity can be highly variable when the coastal land contribute to the returning signal. This cause a failure of the algorithms that retrieve the WTCs from the onboard microwave radiometer

measurements, resulting in their absence. Also, the algorithms can retrieve the WTCs but their values are considered invalid and are, therefore, flagged by the retrieval algorithms. The invalid WTC values are exemplified in red in Figures 7 and 8 (former Figure 5). If used, invalid SLA values would consequently be obtained. For those altimeter points for which the MWR-derived WTC values are missing, no SLA values can be computed unless the user decides to use WTC values from the model. The estimation of the WTC in these points that has been made possible by GPD+, therefore allowing the computation of SLA, is one of the advantages of the methodology.

Pg.4, row 96, "surface emissivity": Coastal zone has also non homogeneous scattering due to variable waves, winds, surfactant streaks, etc. Are they influencing the retrieval of a valid measurement?

R.: The microwave radiation measured by an on-board MWR, expressed as brightness temperature, corresponds to the sum of three contributions: atmosphere, surface and the cosmic background. Regarding the surface contribution, it depends on the surface temperature and emissivity properties. All WTC retrieval algorithms are based on sea surface emissivity models, so they do not consider the very strong (emissivity higher than 0.9) and variable non-ocean radiation. The problem of the non-homogeneous scattering as mentioned by the reviewer appears in the altimeter (active sensor) measurements. Any surface (different from calm waters) induces a non-homogeneous scattering, influencing the retrieval of altimeter measurements.

Pg. 4, row 105, "is to describe and grant access": The access to a dataset cannot be an aiming of a paper. I think the authors have to reformulate clearly the main goal of this paper that is presenting and validating a dataset and then elucidate specific single objectives
R.: The sentence has been rewritten.

Pg. 4, rows 111-115, "The main objective": Objectives have to be stated in the introduction. Also description of sections has to be moved in the introduction.
R.: The authors are here stating the main objective of the methodology and not of the study itself. To avoid any confusion to the reader, this sentence has been rewritten. The description of the sections has been moved to the Introduction, as suggested by the Reviewer.

Pg. 4, row 118, "GNSS network of stations", please provide a map of GNSS stations used so the reader can appreciate the global coverage
R.: Figure 2 has been added to the manuscript as suggested.

Pg. 4, row 123, "This way", please add "In"
R.: Accepted and changed.

Pg. 5, row 118, "this way are given at station height". The GPS stations are over land. So you measure the column at land point. It is not clear to me (and probably to most of not expert people) how this value is extrapolated to the ocean
R.: The handling of the GNSS observations is described in Section 2.1.1. After the computation of the GNSS-derived WTC, at the level of the station height, the WTC are reduced to sea level, the quantity of interest for satellite altimetry, using the height reduction procedure

(exponential decay with height) proposed by Kouba (2008), cited in this paper. This height reduction is fully described in e.g. Fernandes et al. (2013a, 2015).

Pg. 6, row 156, "In fact, GPD+ is an upgrade from the GPD methodology": Please better clarify differences between GPD and GPD+. Apparently you say that GPD+ was for coastal zone but now global. Is the reason related to CryoSat-2 ? as it has no radiometer onboard.

R.: The GPD methodology was developed to compute the WTC only for coastal points, where WTCs derived from the on-board MWR are usually invalid. In its former version, the GPD methodology used as input GNSS-derived WTCs and valid MWR measurements only. Later, the methodology was updated to estimate the WTC for CryoSat-2. Since this mission does not possess an on-board MWR, it was necessary to estimate the WTC not only for coastal regions, but also for open ocean. To do this, the GPD methodology was improved to use data from the scanning imaging radiometers, which are available over coastal regions as well as open ocean, as another input data source. This later version was called GPD Plus (GPD+). The sentence has been rephrased for clarity:

"In fact, GPD+ is an upgrade from the GPD methodology, which was developed to compute the WTC only for coastal points, relying only on GNSS and valid on-board MWR measurements. Motivated by the need to compute an improved correction for CS-2, the SI-MWR data set was included and the focus of the correction extended to open ocean.".

Pg. 5, row 158-164: as you provide a table it is redundant here to report names of the mission. It is important to add space and time resolution of single MWR sensors in the table. A matrix has to be added showing the MWR sensors available for each altimetry mission. Again, this is an important figure for the reader. Some comments about substantial differences between sensors should be recalled here from cited references

R.: We believe the information concerning the data providers for the different SI-MWR missions should be kept in the paper, however the spatial and time resolutions of the SI-MWR missions have been added to Table 1 as requested. Since the number of SI-MWR sensors varies with time, a figure showing their availability along time for each satellite altimetry mission has been added (current Figure 3). Also, a sentence summarizing the main differences between the data types has been added to the manuscript:

"Two types of TCWV products have been used: Level-2 swath products in HDF-EOS2 format (near real time products, 14-15 orbital swaths per day available for each instrument) from all data sources except RSS, and Level-2 gridded products (two grids per day, each containing the ascending/descending passes) in binary format from RSS.".

Pg. 5, row 173-176, "It is known that, in addition to TCWV, WPD also depends on temperature. Expressions such as Eq. (3) account for an implicit modelling of this dependence. Fernandes et al. (2013b) have shown that this expression leads to similar results as those obtained by adopting formulae that make use of explicit values of atmospheric temperature given e.g. by an NWM." The reader might not understand what you mean here with "Implicit" and "explicit" values. Please show examples of comparisons with WTC derived from NWPs in open ocean and in coastal zone.

R.: The WTC can be calculated from using the expression given in Bevis et al. (1994), given below:

$$WTC = -\left(0.101995 + \frac{1725.55}{T_m}\right)\frac{TCWV}{1000}$$

where $T_m$ is the weighted mean temperature of the atmosphere. This expression shows that the WTC explicitly depends on the temperature. Equation (3) given in the manuscript does not depend explicitly on temperature as the former. The results requested by the Reviewer are given in Fernandes et al. (2013b), which show that, after sensor inter-calibration, crucial to guarantee datasets consistency, the WTC derived from both methods are equivalent, with differences within ± 2 mm. This result has been added to the manuscript and the following sentence has been included in the revised manuscript:

"The authors show that after sensor inter-calibration, a crucial step to guarantee datasets consistency, the WTC derived from both methods are equivalent, with differences within ± 2 mm.".

Also, the reference Bevis et al. (1994) has been added to the manuscript to direct the reader to the appropriate literature, in case of interest:

Bevis, M., Businger, S., Chiswell, S., Herring, T.A., Anthes, R.A., Rocken, C., Ware, R.H. (1994), GPS Meteorology – Mapping Zenith Wet delays onto precipitable water. Journal of Applied Meteorology, 33, 379-386.

Pg. 6, row 179-180, "We recall that the WTC is the symmetric of the wet path delay and the quantity of interest in satellite altimetry" Please rephrase and specify what you mean with "symmetric"

R.: The term "symmetric" has been replaced by "absolute value" and the sentence referred by the Reviewer has been rewritten as: "It is recalled that the WPD is the absolute value of the WTC, the quantity of interest in satellite altimetry.".

Pg. 6, row 180, "RA data necessary to compute", Please specify the sources you used for corrections, orbit, MSS, etc.

R.: The models and corrections used to derive the SLA datasets are provided in the altimetric products (RADS and Envisat FRM V3.0). To derive these datasets, used to analyse the SLA variance reduction, the same corrections and models are kept unchanged except the WTC. In other words, an SLA dataset is computed using a set of selected models and corrections and the WTC from ERA, then another SLA dataset is computed using the same models and corrections and the MWR-derived WTC, and so on. We do not consider necessary to enumerate all the models and corrections used to generate the SLA datasets, since these SLA datasets have been generated only to perform the statistical assessment, i.e., have been used only as a mathematical tool.

Following the Reviewer's comment, however, the following sentence has been introduced in the revised version of the manuscript:

"The criteria to select valid SLA are those recommended in the literature and adopted in the standard RADS processing (Scharroo et al., 2012, cited in this manuscript) and include: application of thresholds for all involved fields (satellite orbit above reference ellipsoid, altimeter range, all range and geophysical corrections), altimeter ice and rain flag (whenever set) and SLA within ±2m.".

Pg. 7, row 191, "Threshold values used in this criterion depend on the RA mission": Please specify thresholds

R.: The threshold values are specified in the text that follows the referred sentence (lines 193-196). Values of 30 and 15 km have been set for ESA missions, GFO and T/P, and for the Jason series of satellites and SARAL, respectively.

Pg. 7, row 194, "at distances from coast": The authors use some editing criteria. I am curious to know what happens when tracks are parallel to the coast , but also some situations, e.g., Indonesia where the altimeter crosses successive land segments due to presence of closest islands.

R.: The distance that is inspected by the algorithm is the distance from the point to the closest land point. If a track is parallel to the coast and the distance from its point to the coast is less than the threshold value, all points will be flagged as invalid by the methodology, even if they are not flagged as invalid in the original products. This guarantees that non-flagged invalid MWR-derived WTCs contribute to the estimations.

Pg. 7, row 203, "number of 18 Hz measurements to compute the 1 Hz": is the global product at 1 Hz (i.e. around 7 km spaced for all missions)? While in open ocean it makes sense, I am bit skeptical the coastal zone might benefit from this product if not provided at 18/20 hz. It has been demonstrated that we need high resolution data in the coastal zone (and in fact waveforms are retracked at that rate and SLAs computed at that rate). Otherwise, the user will not be able to exploit the product.

R.: As already explained, the GPD+ WTC database has been computed for GDR products, which are used by most non-expert users. Over open ocean regions, the MWR-derived WTC is the best choice to account for the wet path delay in the altimeter measurements, and this correction is usually available in these regions. Users that want to extend the use of these products in the coastal zone must rely on model-derived WTC since the former is usually invalid or absent in the coastal zone. Discontinuities may therefore occur between both corrections. The GPD+ WTC, which preserves the valid MWR-derived WTC over open ocean and improves the WTC estimation in the coastal region, has the advantage of being a continuous correction in the transition open ocean/coastal zone. As already explained, expert users can interpolate the GPD+ WTC for the location and epoch of the high-rate altimeter measurements, benefiting this way of an improved WTC in coastal zones. To prevent the loss of points when interpolating to 20 Hz points, in addition to ocean points, the WTC for the closest land point, computed at sea level, is included as explained in the manuscript. Provided the necessary funding is allocated, the GPD+ WTC can be computed for high-rate altimetry products.

In the revised version of the paper, results highlighting the improvement in the Envisat SLA datasets when the GPD+ WTC is used in coastal regions have been added. A summary of the results for the other missions have been included in the revised manuscript. The readers are advised to refer to the cited references from the authors.

Pg. 7, row 203-204, "For approximately 10% of all oceanic points": What do you mean with "oceanic domain ? Does it include coastal zone ? at which distance ? The value seems for Envisat only. What about the other missions?

R.: This percentage is computed using all points over ocean with valid SLA values (i.e., along-track points with all available corrections but the MWR-derived WTC, which is computed using GPD+), including coastal regions.

The values given in the manuscript are typical for ESA's missions. Results for other missions have been added in the conclusions section:

"The percentage of recovered points when GPD+ is applied in place of the baseline MWR-derived WTC, depends on instrument type, band of latitudes covered by the mission (which determines the extent of ice contamination) and instrument performance. For all ESA missions (ERS-1, ERS-2, Envisat, Sentinel-3) and SARAL, possessing 2-band radiometers and measuring up to latitudes ±81.2⁰, the percentage of recover data is similar to that of Envisat, in the range of 7% - 15% of the SLA valid points of each cycle. For the reference missions, measuring only up to ±66.7⁰ and already possessing an improved WTC near the coast (all except T/P), this percentage is smaller, from 2 to 4%. For T/P, these values are from 4% to 7%, larger in the second half of the mission. For GFO, measuring up to ±72.0⁰, the percentage is similar to that of TP. Exceptions occur for various missions over periods of instrument malfunction, when the percentage of recovered points can be considerably larger, up to 100%, as it happens for Envisat and GFO.".

Pg. 8, row 220, "parameters have been obtained for Envisat": Please provide parameters for all missions

R.: The calibration parameters for all satellite altimetry missions possessing an MWR are now provided in Table 2. Subsequent tables have been renumbered.

Pg. 8, row 240, "For all satellite missions but CryoSat-2 and for each along-track point deemed as invalid": The sentence is unclear, please rephrase

R.: The sentence has been rewritten to: "For the altimetry missions carrying an on-board MWR (all but CryoSat-2), a GPD+ WTC estimate is calculated for all along-track points with an MWR-derived WTC deemed as invalid, using valid WTC observations from different sources at the nearby location and within a time interval, defined by the spatial and temporal radiuses of influence used in the computation.".

Pg. 9, row 275, "50 km from the ocean": The setting of this value has to ne justified

R.: The justification has been included: "To prevent the loss of points when interpolating to 20 Hz points, in addition to ocean points, the closest point over land is included, provided it is within a distance less than 50 km from the ocean. This guarantees that observations over ocean necessary to compute the WTC for this location are still available within the radiuses of influence centred on the point. The WTC estimated for the closest points over land are also estimated at sea level.".

Pg. 9, row 278, "Figure 4 gives an example of the GPD+ WTC for Envisat's cycle 12". I don't understand the message of this figure. The upper map is substantially unreadable. The lower map is not providing information as the reader would like. Moreover, one cycle per one mission would be only for visual purposes. There is no comments in the paper. The reader expects quantitative results about the improvement.

R.: Figure 4 is presented to show, as an example, the availability of the GPD+ WTC globally (Panel (a)). Panel (b) shows the correction over ocean only, to be used in satellite altimetry. The idea is to show the global coverage, and therefore one of the advantages, of the GPD+ WTC. This explanation is given in lines 278-279 of the original manuscript.

Pg. 10, row 289, "respectively, are provided at 1 Hz.". Previously, the authors mentioned 20 Hz. People using the product in the coastal zone need 20 hz data. I don't understand the utility of publishing a product that then in practice it is not usable from coastal zone users (who are not experts in altimetry). The authors refer to RADS that cannot be considered a "coastal altimetry product". In my opinion, the authors have to satisfy the user requirements if they want to publish this dataset.

R.: We believe that the GPD+ WTC satisfies the requirements of the users who want to base their analyses on the GDR/RADS products. In what concerns the availability of the GPD+ WTC for coastal purposes only, please refer to previous answers.

Pg. 11, row 315-318: "For results concerning algorithm.": The reader is confused here and reminded to previous paper. Indeed, the reader wants to see statistics of all missions here with the application of the algorithm described here. The authors have to add relevant statistics of all missions.

R.: We would like to emphasise that the purpose of this paper is not to describe in detail the results for all missions as that has already partly been done in previous papers. Moreover, a paper with an exhaustive description of the results for all missions would necessarily be very long and tedious. Here, we believe the focus should be on the benefits of using these products. Therefore, we detail the results for Envisat, not presented before, and provide a summary of the results for all other missions in the conclusions.

Pg. 11, row 320, "The GPD+ WTC is here compared to the ECMWF Reanalysis WTC": This kind of comparison make sense in open ocean but not in the coastal zone. The authors provide a title "A coastally improved global dataset. . ...". They clearly state previously that models fail in the coastal zone and now they use for validation.

R.: Actually, the authors assessed the performance of the GPD+ WTC by comparing it with those WTC available to the users in the altimetry products, to show the improvement attained when the GPD+ WTC is used in the SLA datasets generation, instead of using the MWR- or NWM-derived WTCs. The word "validation" has therefore been changed to "assessment" throughout the text whenever it was used incorrectly.

Pg. 11, row 335, "Figure 5 shows the GPD+ WTC for some Envisat tracks": The reader expects to see the map showing where the passes are located and identification of the segments where the new corrections improves. The discussion of Figure 5 is not provided. The plots have to be commented in relation to the places touched over ground.

R.: Figure 5 has been divided into Figures 7 and 8 and now includes the geographical coverage of the selected tracks, as we agree with the Reviewer that this information is necessary. The discussion of these figures has been included.

Pg. 11, row 340, "interesting results": please remove being subjective
R.: The sentence has been rewritten.

Pg. 11, row 346, "most of these points are located at high latitudes and in coastal regions": This statement is not demonstrated in the figure. The authors expects to see zooming in coastal regions to see improvements.

R.: Results for three different coastal regions have been added in the revised version of the manuscript.

Pg. 11, row 361, "for the whole Envisat mission": the authors have to provide th same figure for the other missions too

R.: We have already explained that it is not possible to present detailed results for all missions, neither it would be relevant to repeat results already published before.
Similar figures for most missions (T/P, Jason-1, Jason-2, ERS-1, ERS-2, Envisat, GFO and SARAL) are provided in Vieira et al. (2019) and in Fernandes and Lazaro (2018) for Sentinel-3 (both cited in the paper). The following sentences have been introduced in the paper:
"For other missions, results have been presented in Vieira at al. (2019) and in Fernandes and Lázaro (2018) and are summarised here. For the 2-band radiometers, land contamination on the MWR observations occurs for points at distances from coast smaller than 25-30 km (ERS-1 and ERS-2), 20-25 km (Sentinel-3) and 15-20 km (GFO and SARAL), the latter in agreement with the smaller radiometer footprint of the SARAL MWR. Similar analysis shows that land contamination is observed up to 25-30 km from the coast for T/P and Jason-1 and up to 20-25 km for Jason-2 and Jason-3. These numbers are function both of the instrument footprint size and of the efficiency of the criteria used to detect valid/invalid MWR observations, since in these plots only MWR values that passed all validation criteria, except for the distance from coast, have been used. In summary, for each mission, these analyses show the distances from coast up to which the MWR observations are contaminated by land and must be discarded. Moreover, they also show that GPD+ is efficient in removing this effect.".

Pg. 12, row 369, "The results are shown in Fig. 7". The authors state the product is at 1 Hz (7 km) and in the plot show values at less than 5 km

R.: Figure 7 (now Figure 10) shows the RMS of WTC differences in bins of distance from coast. While along-track points are separated by 7 km, the points closest to land can be at any distance from the coast, even at distances less than 5 km.

Pg. 13, row 393-395, "Therefore, three SLA datasets of collocated along-track points were derived using the same standard corrections (Sect. 1) but the WTC, which can be the Composite correction present in AVISO CorSSH L2P products (Comp), the GPD+ or the ERA Interim WTCs.". This comparison makes sense only in the open ocean and not in the coastal zone (0-50 km)

R.: The GPD+ WTC has been compared with the other WTCs available in the altimeter products provided by RADS, GDR, PEACHI and AVISO for use in both open-ocean and coastal regions. The Comp WTC is the result of the methodology developed by AVISO, to improve the WTC in the coastal region, therefore we consider that the comparisons shown are reasonable. However, following the concerns of Reviewer#2, who were right pointing out that the Composite WTC available at the time of our analysis in AVISO products has not been computed using this new Envisat FMR V3.0, we decided to show in the revised version of the manuscript the assessment of the GPD+ WTC by comparing it with the ERA- and MWR-derived WTCs, which are the actual corrections provided in these products. Therefore, Section 3 has been rewritten accordingly.

Pg. 13, row 406, "Fig.8a": Fig. 8c si not commented in the text. Moreoer, there is a strange behavior around cycle 95

R.: Figure 11 has been changed to include a new panel (b) the number of points used in the MWR and GPD+ WTCs, since it is different from the number of points used in the comparison with ERA (shown in panel (d) of the same figure). This is explained in the manuscript. Reference to both panels (b) and (d) of Figure 11 (previous Figure 8) have been added in the text.

In October 2010, a new orbit configuration (30-day repeat cycle) for Envisat was implemented, corresponding to a change from Envisat Phase b to Phase c. As a consequence, a large amount of data was lost in the period corresponding to cycles 94 and 95. This information has been added in the revised paper.

---

## Author Comment (AC2) · 14 Jul 2020

We would like to start by thanking the Reviewer for his or her careful reading of our manuscript and for taking the time to assess it.

The main changes introduced in the manuscript following the comments and suggestions of the two Reviewers can be summarised as:

- Sections containing the Abstract and the Conclusions have been updated to accommodate the new results presented in the revised version of the manuscript.
- Some parts of the text have been moved to new sections or were rewritten/completed to be clearer and more informative.
- Figures 1 as well as figures 11, 12 and 13 have been updated, the latter to include the results for the comparison of the GPD+ WTC with the MWR-derived WTC, instead of that for the Comp WTC, following the concerns raised by Reviewer#2.
- Previous Figure 5 has been divided into Figures 7 and 8 and the geographic location of the Envisat tracks have been added, following the recommendation of Reviewer#1.
- New figures have been added to the revised version (Figures 2, 3 and 14).
- Tables 1 and 4 have been updated, the former to include more information, the latter in the sequence of the last update of the GPD+ database (performed to include more data for the recent missions).
- A new table (Table 2) has been added in the revised version.
- All figures and tables have been renumbered.
- Section 3.2 has been divided into sections 3.2.1 and 3.2.2 describing the global and the regional (coastal) results, respectively, and the text has been extended.
- Reference Vieira et al. (2019c) has been updated, since at the time of this revision it has already been published.
- Reference AVISO (2017) has been removed.
- Five new references (Bevis et al. (1994), Rudenko et al., (2017), Valladeau et al. (2015), Dinardo et al. (2020) and Escudier et al. (2017)) have been inserted in the revised version.

We have responded to all the comments and suggestions raised by the Reviewer as follows.

The paper presents a dataset of wet tropospheric correction applicable to altimetry and the methodology used to product it. The wet tropospheric correction is one of the correction applied to the altimeter range to compute the Sea Level Anomaly. The WTC is traditionnaly provided by on board microwave radiometer, measuring in appropriate frequencies bands to correct for the excess path delay. The estimation of the WTC from the MWR measurements can be degraded by extreme rains events, ice surface, land contamination in coastal areas,

instrument malfunctions. The author proposes a method named GPD+ that intend to improve the MWR-based WTC of operational processing, or propose a correction for mission without MWR on-board (Cryosat-2).

The method consists first in the filtering of the invalid WTC estimation from the operational product and second, by the estimation based on the objective analysis using external data such as GNSS data, MWR Imaging data (providing water vapour), Numerical Weather Prevision model (ECMWF, ERA interim). The method is applied to almost all conventional missions.

The dataset used for the algorithm is rather well defined. The section about the GNSS dataset lack a discussion about the coverage of this network. Although the paper states otherwise, GNSS stations don't seem to be distributed globally over the globe.

R: The GNSS stations used in the methodology belong to several international GNSS networks (IGS, EPN, SuomiNet); some stations from national networks have also been used by the authors (e.g., in Indonesia, German Bight, etc.). The Reviewer is right saying that GNSS stations providing atmospheric products are not well distributed over the globe. The authors only say that GNSS stations all over the world are used, provided their atmospheric products are made available to users.

The section about the Imaging radiometer seems more dedicated to the filtering step of the method than to the description of the input dataset itself and the added value of this dataset.

R: By filtering step the reviewer may refer to the calibration step. This is a very relevant step, as it is important to ensure that the corrections are stable in time and do not introduce spurious trends in the SLA. In the revised version, information about the SI-MWR products have been inserted as well as the added value of this dataset.

The NWP dataset is slighlty described, and lack a discussion of the difference between ERA Interim and ECMWF. The paper don't say if one mission can be covered by only one model or if two are needed, if there is a bias between the two models that shall be corrected. Also the paper stated that NWM data are provided as output from GPD+ for northernmost latitudes, but the method to adjust the model to measurements is not clearly defined (is it a simple bias computed over each cycle?).

R: As ECMWF operational model has undergone several updates, this model does not have the same accuracy over time. For example, the RMS of the differences between MWR-derived WTC and NWM-derived WTC, in points with valid MWR values, is in the range 1.2-1.4 cm after 2004. Before that date, the RMS of differences increases as we go back in time, reaching 2.8-3.0 cm in 1995. On the contrary, ERA Interim is fairly uniform, with RMS of WTC differences with respect to the MWR WTC in the range 1.2-1.4 cm, including the period of the first altimeter decade. For this reason, for all missions with data before 2004 (T/P, Jason-1, ERS-1, ERS-2, Envisat and GFO), ERA Interim is used in GPD+, while for the most recent missions the Operational model is adopted. More details can be found in Legeais et al. (2014), cited in this manuscript. A sentence clarifying this has been added in the revised version of the paper.

The question concerning the adjustment of the WTCs derived from the model and GPD+ is addressed below.

The algorithm is well described and the workflow provides a clear overview of the processing. The method is assessed for Envisat only in this paper. The paper introduces Full Mission Reprocessing (FMR v3.0) but compare GPD+ dataset to the Composite Correction extracted from L2P products issued from an older reprocessing (FMR v2.1). According to the L2P product handbook available on AVISO, there is no composite correction in the L2P products, only MWR-derived correction. This point must clarified. In the validation section, both corrections (MWR-based and composite) are used for comparisons. it is difficult to follow which version of the Envisat MWR-based correction is used for the generation of the GPD+ and which one is compared to the GPD+.

R.: The Composite correction has been developed also aiming at getting a WTC with validity extended up to the coast. According to personal communication of colleagues from CLS, the AVISO products usually adopt this correction. However, it is difficult to find a proper reference for this product. Moreover, the Reviewer is right when states that the MWR-derived WTC used to generate the Composite WTC is different from the one provided in FMR V3.0 dataset. Therefore, in the revised manuscript we dropped the comparison with the Composite WTC for Envisat. Now, GPD+ WTC is compared with both the ERA-derived WTC and the MWR-derived WTC (from Envisat V3.0), the later therefore has replaced the comparison with the Composite correction. In the present comparisons between GPD+ and MWR, the points for which the MWR observations are not set (NaN values) or are out of the limits of the WTC range (-50 cm to 0 cm) in the GDR products, have been discarded from the analysis. This has been clarified in the paper.

The number of 30% of invalid WTC data over ocean for Envisat is stated but not justified. This number seems quite high.
R.: This question is addressed below.

Moreover, the criteria to select valid SLA points is not discussed. The paper shall define the criteria of validity of the SLA. The L2P products provide a validity flag that could be used.
R.: The following sentence has been introduced:
"The criteria to select valid SLA are those recommended in the literature and adopted in the standard RADS processing (Scharroo et al., 2012, cited in this manuscript) and include: application of thresholds for all involved fields (satellite orbit above reference ellipsoid, altimeter range, all range and geophysical corrections), altimeter ice and rain flag (whenever set) and SLA within ±2m.".

The comparison of GPD+ with GNSS is more a validation of the method than a performance assessment. This section can gain in clarity in the method used for this comparison. GNSS data are not independent of GPD+ data. Is the GNSS data cited in this section also used in the generation of the GPD+ dataset (data from another network for example)? In the first sections, one of the criteria for rejection of MWR-based correction is the distance to coast, but it is not clear if this criteria is used in this section. The fact that the method is not clear makes the figure 7 difficult to understand.
R.: As stated in the paper (line 353 of original manuscript), GNSS data are not independent from the GPD+ WTC, as they have been used in their computation. Nevertheless, the analysis of the root mean square (RMS) value of the GNSS-derived and GPD+ WTC differences, function of distance from coast, is valuable to inspect the correction in coastal regions since it

allows us to derive a threshold value of the distance to coast where the radiometer correction starts to become invalid (even if not flagged as invalid in the original GDR). Once this threshold value is obtained, it can and should be used in the GPD+ algorithm. This has been done for the Envisat FMR V3.0 used in this paper (more details are given below). The authors have detailed the methodology used in this comparison in the revised paper.

The paper provides a performance assessment (and not accuracy) of the section 3.2 using analyses of Sea Level Anomaly variances. In this section, the author compares the composite correction which is not part of the L2P products to the GPD+ correction. The method used is to select all valid SLA points, and for the points with the composite outside limits or invalid, the ERA interim WTC value is used. These points shall be discarded from the analysis as they do no represent a fair comparison with the MWRbased correction as the use of the model correction will degrade it.

R.: As stated before, the assessment with respect to the Composite WTC has been replaced by the corresponding comparison with the MWR-derived WTC. Since in this comparison we cannot assess the performance of the GPD+ WTCs in the points where these corrections are not set, we have now removed these points from the analyses. We believe that the assessment of the GPD+ through comparison with the model WTC is important for the users, who must rely on model data when the correction from the on-board radiometer is invalid/absent.

The previous sections have already shown that the GPD+ retrieve some invalid points.

R.: The GPD+ methodology retrieves a WTC estimate for all points with an invalid MWR-derived WTC. In the absence of observations (valid MWR-, GNSS- and SI-MWR-derived WTCs), the GPD+ output is the first guess (ERA Interim WTC) adjusted to the valid MWR-derived WTCs, as explained here. Therefore, all GPD+ estimates are valid. The GPD+ products provide a flag identifying the model-derived WTCs.

The color scales for figure 9 is not well chosen and is difficul to read.

R.: In Figure 9 (now Figure 12), blueish colours have been chosen to show an improvement from the use of the GPD+ WTC in the computation of SLA, compared to the use of other WTC correction, while the yellow to red colours show a degradation of the SLA dataset when the GPD+ WTC is used. The green colour is used for differences around zero. Since this colour can be difficult to see in the colour scale, a note has been added to the caption of the figure.

Figure 10a shows a strong peak, with not physical values, for latitude around 50°N that is not explained in the paper.

R.: This comment is addressed below.

For this diagnosis, it is not stated if data cover open-ocean only, or ocean and coastal areas. Moreover the figure shows a reduction of the SLA variance from 200km up to the cost but it is difficult to see the improvement close to the coast.

R.: For the diagnoses described in the paper, global data have been used (please refer to former Figures 9 and 10 (Figure 12 and 13 of the revised paper), which show that data for the whole ocean, including coastal regions, have been used). This has been written clearly in the revised paper. The improvement in the closest 20 km to the coast can be as high as 3 cm$^2$ for

the GPD+ and MWR-derived WTCs comparison and 1 cm² for the comparison with ERA. We believe that in both cases the improvement in SLA dataset is significant, by reducing the SLA error introduced by the WTC in a few centimetres.

Although the paper title is "A coastally improved global dataset", there is no real focus on coastal areas. It is not stated clearly in the paper but it seems that the dataset is based on 1Hz data where 20Hz data are more adequate for studies on coastal areas.

R.: In response to the Reviewer's concern, results for three coastal regions selected, on the one hand, due to the large number of available GNSS stations (North American and European coasts) and, on the other hand, due to the fact of being a challenging region for coastal satellite altimetry (Indonesia region), have been added to the revised version of the manuscript, as an attempt to show the potential of the GPD+ dataset along the coastal waters. Section 3.2.2 has been added in the manuscript to present these results.

The rate of the altimetry measurements is not a limitation to the GPD+ methodology. In the scope of a current research project in which the University of Porto (UPorto) is involved, the GPD+ methodology will be used to estimate the WTC for the coastal (and inland waters) zone for CryoSat-2 and Sentinel-3 missions. The outcome of this project will be a GPD+ WTC product at high rate (20 Hz), intended to be used for applications over the coastal zone (i.e., no ocean values included for distances larger than ~100 km off the coast). However, the GPD+ WTCs presented in this manuscript have been computed to be incorporated in altimetry products providing observations at 1 Hz rate, still the most used by the altimetry community databases. They are intended for users who want to have a consistent and continuous WTC correction, from open ocean to coasts (and polar regions as well). The correction can be extended up to the coast since a valid WTC value is provided for the first along-track measurement over land. Users can therefore use this measurement to interpolate the valid GPD+ WTC up to the coast, for the location and time instant of the 20 Hz data. Moreover, as the onboard radiometer data are not available at a higher than 7 Hz rate, neither these data nor the third-party data have enough resolution to be provided at 20 Hz. Therefore, and for the time being, the strategy for those users who want to focus on coastal ones, would be to interpolate these 1-Hz data to the location and time instant of the 20 Hz data.

For high-frequency MWR, expected in the future, high-rate WTCs are definitely advisable, and the authors intend to exploit this possibility. A sentence has been added in the section with the conclusions.

Minor comments

Row 118: "GNSS network of stations distributed globally along the coastlines": GNSS stations don't seems to be distributed globally over the globe. A map could be added to show the position of the GNSS stations used for the generation of the dataset.

R.: Following the suggestion of the Reviewer, a figure showing the location of the coastal and island GNSS stations used in this study has been added in the revised version (Figure 2).

Row 195: "values of 15km have been used for Jason-1/2/3": 15km seems quite small for this serie of MWR knowing that they measure at three frequencies, including a 18.7GHz with a large footprint. What is the reason for that?

R.: The quoted values refer to those used currently in the GPD+ processing. For the Jason series of satellites, a smaller value has been adopted since the WTC provided in their products

are already improved in the coastal regions using the methodology developed by Brown (2010), cited in the text. However, the assessment of the MWR-derived WTCs through their comparison with GNSS-derived WTCs in the coastal zone has shown the existence of contaminated measurements for distances larger than 15 km off the coast. The result from this assessment for Envisat (30 km), shown in current Figure 10, has already been used in the GPD+ processing described in this paper. The same value was obtained for E1, E2 and GFO, which is the value in use in the GPD+ processing. For the reference missions (T/P, J1, J2 and J3), however, the assessment using GNSS data has shown land contamination up to 25 km off the coast. Therefore, the threshold value currently set (15 km) will be updated in the forthcoming GPD+ processing for these missions. For SARAL and Sentinel-3, the outcome of this assessment were thresholds of 15 and 25 km, respectively, that have been already implemented in GPD+. It should be emphasized that the distance from coast is a rejection criterion applied after a set of other criteria, such as the radiometer land flag, that, if efficient, should have already rejected land contaminated points. So, these distances must be large enough to ensure the rejection of contaminated points, but also conservative to avoid rejection of good MWR observations.

Row 201: Talking about Envisat datat from latest reprocessing, the author states "30 % of the oceanic points have an invalid WTC value": This seems quite a large number of invalid points when focusing on ocean surface only (with a valid SLA). From Figure 2, it does not look like one third of the points are invalid. How do you explain that number?
R.: Altimeter data are acquired along satellite tracks only, therefore at low latitudes the distance between adjacent tracks is maximum, with large diamond-shaped regions without altimeter points. Therefore, the quantity of points sampled by the altimeter varies with latitude, being maximum over polar regions where WTCs are usually invalid. Also, the points within a strip of width 30-50 km along the coasts have usually invalid WTCs. Additionally, as depicted in Figure 2 (now Figure 4), there are generally full tracks with invalid WTCs. The percentage of ocean points with invalid WTC for Envisat cycle 12 is 29.5%. The corresponding number when only points with valid SLA are selected is 10.9%. These figures have been added to the text.

Row 209: the author states "Data from the reference missions". For a non-specialist audience, the author should explain which are the reference missions.
R.: The explanation has been added in the revised version of the manuscript.

Row 219: for the intercalibration processing, the difference at cross-over points with a time-lag of 180 minutes between reference missions and other altimetry missions are computed. Is that time span not too large for WTC ?
R.: As explained in Fernandes et al. (2013b), cited in the text, this value has been chosen to guarantee the existence of enough crossovers to perform the analysis i.e., it is the best compromise between the number of crossovers and the minimum time interval.

Row 229: "In addition, to reduce data discontinuities, : : :." : from this sentence it seems that a bias is computed between the MWR and the NWM correction for each cycle. What is the rationale for a simple bias? How is computed that bias?

R.: The bias is computed, for each cycle, as the mean difference between MWR and model WTC, for all points where the former correction has been considered valid. The rationale behind this comes from the observation that the differences between the WTC from MWR and that from models, in addition to the small scales observed in Figures 7 and 8 in the revised version, have long-wavelengths from yearly to decadal signals. See for example in the figure below, the mean cycle differences between Sentinal-3A (S3) MWR and ECMWF operational (cyan) and between S3 MWR and ERA5 (blue). Although these differences are small (only a few mm) the application of a mean bias per cycle helps to reduce these small discontinuities.

[Figure]

The method used to adjust the NWM-derived and GPD+ WTCs has been described in more detail in the revised version of the manuscript.

Row 274-275: "To prevent the loss of points when interpolating to 20 Hz points, in addition to ocean points, the closest land point is included, provided it is within a distance less than 50 km from the ocean." Can you clarify the processing here? what is the closest land point?
R.: This means that, for each track crossing a coastal zone, a GPD+ WTC at sea level estimate is also computed for the first altimeter measurement point located over land. This WTC estimate and the previous one over ocean, allow the user to perform the interpolation of the WTC field if high-rate data are to be used. The sentence has been rewritten to "To prevent the loss of points when interpolating to 20 Hz points, in addition to ocean points, the closest point over land is included, provided it is within a distance less than 50 km from the ocean. This guarantees that observations over ocean necessary to compute the WTC for this location are still available within the radiuses of influence centred on the point. The WTC estimated for the closest points over land are also estimated at sea level." to become more clearer to the reader.

Row 320: ""The GPD+ WTC is here compared to the ECMWF Reanalysis WTC (ERA Interim, GDR field mod_wet_tropo_cor_reanalysis_01) and with the WTC present in the AVISO CORSSH L2P products in July 2019 (AVISO, 2017). The latter dataset is usually called Composite Correction ". You state here that you compare the GPD+ to Composite correction, but latter (line 334). But according to the L2P products handbook

([https://www.aviso.altimetry.fr/fileadmin/documents/data/tools/hdbk_L2P_all_missions_exce](https://www.aviso.altimetry.fr/fileadmin/documents/data/tools/hdbk_L2P_all_missions_except_S3.pdf)
[pt_S3.pdf](https://www.aviso.altimetry.fr/fileadmin/documents/data/tools/hdbk_L2P_all_missions_except_S3.pdf)), there is no composite correction in these products. And latter, the author says that
he used the field 'rad_wet_tropo_cor_sst_gam_01'. This point shall be clarified

R.: As stated before, the comparison with the Composite WTC has been removed.
The MWR-based correction used in the generation of the GPD+ Envisat files is the
'rad_wet_tropo_cor_sst_gam_01' field provided in Envisat FMR V3.0 GDR dataset, based on
a five-input algorithm, according to reference:
Collecte Localisation Satellites (CLS). Surface Topography Mission (STM) SRAL/MWR L2
Algorithms Definition, Accuracy and Specification; S3PAD-RS-CLS-SD03-00017; CLS:
Ramonville St-Agne, France, 2011.

Row 330: "Anomalies in this field have been found, with the field out of limits in a set of points,
most of them concentrated on certain passes," : Do you mean that you found anomalies in the
ERA interim product for WTC field?
R.: Yes, please see sentence below, added to the paper:
"Anomalies in this field have been found, with the field out of limits in a set of points, most of
them concentrated on certain passes. This is due to the fact that this correction has been
computed from 3D model fields at the altimeter measurement altitude. Therefore, whenever
the altimeter-derived surface height is not set (NaN value), the corresponding model-derived
WTC will also be NaN. As our goal is to be able to provide continuous WTC, without data
gaps, this field is unsuitable for use in the GPD+ estimations."

Row 334: "The MWR-based correction used in the generation of these files" : Which files?
R.: The sentence has been rewritten in the revised version of the manuscript to make it clearer
to the reader.

Row 342: The author found 30% of points with a rejected MWR-derived WTC. This figure
seems quite large. It could be interesting to discuss that number and provides some insights
of the repartition within the different causes. It seems this number is estimated over ocean.
Does it include coastal regions? Which latitudes?
R.: The analysis is global, including coastal zones and the whole range of latitudes. This can
be verified from Figure 2 (Figure 4 in the revised version), which shows an example of all
points with invalid MWR-derived WTCs. This figure allows the reader to inspect the causes
that led to the occurrence of all invalid WTCs. A sentence emphasizing that the reader, when
analysing Figure 6 (Figure 9 in the revised version) can also refer to Figure 2 has been
included. Moreover, the percentage of points contaminated due to each cause has been
included in the text for Envisat cycle 12 (the same cycle used to generate Figure 2).

Row 362: "Only GPD+ estimates retrieved using observations are selected." Which
observations? MWR? GNSS?
R.: WTC from along-track MWRs, SI-MWRs and GNSS stations are considered observations.
The referred sentence has been rewritten as "Only GPD+ estimates retrieved using
observations (valid MWR-, GNSS- and SI_MWR- derived WTCs) are selected, GPD+
estimates based on model have been discarded from this analysis.".

Row 370-376: Methodology difficult to understand

R.: The authors have included a more detailed explanation of the methodology.

Row 381: "On the contrary": -> Moreover, Additionaly …
R.: The suggestion has been accepted.

Row 384: "Accuracy assessment" ==> Performance assessment
R.: The suggestion has been accepted.

Row 420: "third party data": what are those third-party data?
R.: Third-party data are WTC observations, other than those from the on-board MWRs. The explanation has been included in the manuscript, where the term is used for the first time.

Figure 5: b) and c) look quite similar with land/ice contaminated pass. Outliers are not obvious in c).
R.: In general, all tracks have land and ice contamination. We decided to keep both figures because the referred tracks cover different oceans and therefore show different WTC variability. This has been highlighted in the text. Also, the discussion of former Figure 5, which has been divided into figures 7 and 8 to add plots showing the geographical coverage as recommended by Reviewer#1, has been extended in the text. The caption of Figures 7 and 8 includes now a brief description of the issues in the MWR-derived WTC.

Figure 7: why is there an increase in the number of points for the GNSS-GPD+ comparison but not for the GNSS-MWR one?
R.: As the tracks gets closer to the coast, the MWR-derived WTC become invalid or are inexistent, therefore the number of valid MWR-derived WTC diminishes. The GPD+ methodology computes a WTC estimate for these along-track points, therefore allowing SLAs to be computed at these locations and epochs. Therefore, the number of valid WTCs in the coastal region increases, being this one of the advantages of the GPD+ methodology.

Figure 9: The green color cannot be seen on the color scale.
R.: The green colour is used for the SLA variance differences with values around zero. A note has been added in the caption of the figure to help the reader interpreting this result.

Figure 10: What is this peak around latitudes 50_N?
R.: The peak in latitudes 50ºN is related to the large reduction in SLA variance when the GPD+ WTC is used instead of the Comp WTC or the MWR-derived WTCs. This can be seen in the original Figure 9, as dark blue pixels (GPD+ WTC performs better than Composite WTC or MWR-derived WTC) are found in the westernmost coastal regions of the oceanic basins (e.g., in the Gulf of Saint Lawrence or in the Sea of Okhotsk sea).

---

## Author Response (AR2)

**Posted by Anonymous Referee #3**

The authors would like to thank Reviewer#3 for his/her careful consideration of the manuscript and for giving helpful comments. The authors have carefully considered and answered the Reviewer's comments and worked to include most of these comments in the revised version of the manuscript. Please find below the responses to the reviewer's comments, in blue. Text added to the revised version of the manuscript has been highlighted in green.

Review essd-2019-171: A coastally improved global dataset of wet tropospheric corrections for satellite altimetry by Clara Lázaro, Maria Joana Fernandes, Telmo Vieira, and Eliana Vieira (Data description paper)

The authors describe the GPD+ data set, which is an alternative wet tropospheric correction available for almost all radar altimeter missions since 1991. The wet tropospheric correctiom is necessary for the derivation of sea level from radar altimetry. In general best results are obtained with a correction derived from on-board micro-wave radiometer measurements. However, part of this data is missing, mainly at polar and coastal regions. For the GPD+ correction, an optimal interpolation scheme merges GNSS-, SSM/I-, SSM/IS- and data from numerical weather models at locations without valid on-board measurements. Furthermore an enhanced outlier detection for the MW radiometer data is applied and the on-board radiometers are cross-calibrated with the SSMI-data since they are prone to drifts. The improvements by the GPD+ correction are assessed for the newly processed ENVISAT data set. Here, the number of valid sea level measurements is increased by 10% with respect to the original radiometer derived correction. The GPD+ correction decreases the global sea level variance at cross-over points and along-track with respect to the model derived correction. With respect to the original radiometer correction the cross-over point variance is slightly decreased and the along-track variance is variability is clearly decreased from end of 2007 onwards. The improvement is mainly originating from the last 100 km and especially pronounced for the last 30 km off shore. For the ENVISAT mission the GPD+ correction is also available for inland waters.

The paper reviews the measurement principles of radar altimetry and the relevance of the tropospheric corrections. The methodology and the different input data sources are described in detail. It summarizes various previously published papers that assess the GPD+-quality for different missions and includes numbers for global reduction of variance and increase of retained SLAs related to the use of GPD+. Possibilities to further improve the data set are discussed. I am missing a more detailed description of the calibrations of the on-board MW radiometers of the reference missions relative to the third-party SSM/I and SSM/IS data. What are the formal errors of this fit and what is the effect on global mean sea level trend?

R.: Following the reviewer's suggestion, we have included in Table 2 the formal errors associated with each calibration parameter. However, the details concerning the calibration of the MWR on board satellite altimetry missions, as well as those of the SI-MWR on board other remote sensing satellites, relative to the third-party SSM/I and SSM/IS data have been already provided in Fernandes and Lázaro (2016), cited in Section 2.1.4 Radiometer Calibration (see line 257). Therefore, we think there is no need to repeat them here, as there is no change to the methodology to report. However, to emphasize

this and orientate the readers to the correct reference, the sentence "For more details concerning the calibration of the radiometers, the readers are advised to see Fernandes and Lázaro (2016)." was added at the end of the above-mentioned section.

When considering each altimeter mission alone, the effect of the calibration on global mean sea level trend is that due to the parameter designated by "linear trend", which indicates if the dataset is well aligned over time with the altimeter reference missions and with SSM/I and SSM/IS. For the Envisat mission, this parameter is negligible. It is quite difficult, however, to predict the effect on sea level trend when multi-mission data are merged since the result is highly dependent on the adopted processing and period of analysis. This is an exciting topic for further investigation that, in our opinion, is out of the scope of this paper, which intends, but is not limited to, to support potential users of the GPD+ WTC dataset. Therefore, we expect to exploit the impact of the sensor calibration in future studies or, alternatively, expect that other researchers could use the GPD+ database to develop their own analysis.

Since the wet tropospheric correction is critical for sea level retrieval from radar altimetry and GPD+ has proven to offer advantages over the conventional corrections this review of the data set is very valuable for users of altimeter data. Therefore I would recommend the publication of this paper after minor corrections.
* * *
Detailed Comments:
The term 'reference missions' should be explained

R.: The definition has been included  in the text that now reads:
"Data from the reference missions for sea level investigations such as the T/P and Jason series (and soon Sentinel-6) have been calibrated against those of the Special Sensor Microwave Imager (SSM/I) and the SSM/I Sounder (SSM/IS) by selecting matching points from each pair of missions operating simultaneously with a difference in time and location less than 45 minutes and 50 km, respectively (Fernandes et al., 2013b)."

The Envisat GPD+ is also provided over inland waters. Have inland GNSS stations been included in the processing?

R.: The GPD+ version used to generate the Envisat FMR V3.0 only computes the WTC for ocean and coastal points. In the absence of a GPD+ WTC estimation over land, and consequently over inland waters, the WTC output from GPD+ is the one derived from ERA Interim. This guarantees that the product is continuous over ocean (including coastal zones) and non-ocean surfaces. In the scope of a current research project in which the University of Porto (UPorto) is involved, the GPD+ methodology will be improved to provide an estimation of the WTC for inland water regions (for CryoSat-2 and Sentinel-3 missions) as well. As a result, we expect to have, in the future, a GPD+ WTC product at high rate (20 Hz), intended to be used for applications over these regions. In the new implementation, all available WTC sources, including GNSS stations, will be used.
To avoid misunderstandings, the text in lines 333-337 (R1 version) has been corrected as follows:
"This has already been done for Envisat, therefore ensuring that Envisat GPD+ products are continuous over ocean and non-ocean surfaces. Future versions of the GPD+ correction for the remaining missions will cover all surface types as well. In addition, it is envisaged to improve the GPD+ methodology so that GPD+ WTCs will be estimated over non-oceanic regions, provided WPD observations exist (e.g. from MWR over large lakes or from GNSS)."

Page 15 and following:
The measure of the quality of the GPD+correction in this paper is the reduction of sea level variance for the global mean and for different regions and coastal distances. Since neither the absolute wet path

delay nor its variability are evenly distributed over the global ocean it might be useful to related the local decrease of variance to the absolute values (e.g. looking at explained variances).

R.: We understand the reviewer's viewpoint here, and we do agree that showing the percentage of explained variance could be advantageous to show the improvement achieved when the GPD+ WTC is used instead of the model-derived one, in global terms, and instead of the MWR-derived WTC over coastal regions. We believe this could be beneficial to show the improvement of the GPD+ WTC particularly over regions of low WTC variance values. We really have considered this suggestion, but soon we have realized that: - to follow the suggestion, a new data processing would have been necessary, preventing the completion of this review in due time; - several figures would have to be updated; - a major part of Section 3 "Results and discussion" would have to be rewritten in order to accommodate the new results. We sincerely appreciate this valuable suggestion, and we intend to change our processing scheme in the future according to this suggestion.

Page 17, 518-521:
I do not understand these arguments. You might want to rephrase the paragraph.

R.: The GPD+ uses as input valid MWR-derived WTCs. If anomalous WTCs have not been correctly flagged as invalid in the products and are not, during the GPD+ processing stage, flagged as invalid by the GPD+ rejection criteria, they will be used in the GPD+ as valid measurements and will consequently degrade the GPD+ WTC estimates. In the authors' opinion, the MWR-derived WTCs are the best source to account for the wet path delay in Satellite Altimetry, since they are collocated both in space and time as the altimeter measurements. Therefore, they should be preserved whenever valid.
We explain that GPD+ corrections are computed for all along-track altimeter points, including those for which an MWR-derived WTC is not present in the products (NAN values). In the analysis presented in this manuscript, statistics are computed only for those locations for which an MWR-derived WTC exists and has been flagged as invalid, provided its value is within the expected WTC range, i.e., is in the [-0.5 m, 0 m] range. Thus, points for which the MWR-derived WTCs are not defined (i.e., default value or NaN) or are out of this range, are discarded from the analysis presented here, since a collocated MWR-derived WTC value is not available for comparison. However, WTCs for these points with missing/out of range MWR-derived WTCs are computed by GPD+ and available in the output GPD+ products, since the methodology relies on other WPD sources apart from those from the MWRs. What we want to say in this paragraph is that since GPD+ estimates WTCs for all points in this situation and these estimates are not included in the analyses presented here, these analyses provide underestimated results for GPD+. In other words, these points would be discarded in the SLA computation due to the absence of a valid MWR-derived WTC, but since a GPD+ WTC is estimated for these points, they can contribute to the generation of the SLA dataset. This is one of the main advantages of the GPD+ methodology. Having explained the arguments, which we believe are valid, we have decided to keep the text as it is, and just added a brief explanation of what are "missing" MWR-derived WTCs:
"Over the Southern Ocean, for latitudes 80ºS-60ºS, some degradation is visible when the GPD+ is used. This could probably be due to the existence of ice contamination in the radiometer-derived (both along-track and image) WTCs. However, it is recalled that, over this region, the MWR-derived WTC is usually missing (default value or NaN) or out of range, and that these points, for which a GPD+ estimate would be computed otherwise, have been removed from the analysis. Therefore, it must be emphasized that these results for the comparison GPD+ and MWR-derived WTCs provide underestimated results for the GPD+."

Page 17, 541-545, page 18, 583-584:
the authors claim that GPD+ retains more valid sea level measurements close to the coast with respect to the MW radiometer correction. However, this is only shown for the global ocean. I would recommend

to include numbers of the percentage of missing/retained data in coastal areas in the text and to add the corresponding curves in figure 13 and figure 14.

R.: The GPD+ methodology preserves the MWR-derived WTCs, provided they have been flagged as valid, and computes a WTC estimate for all points with an invalid MWR-derived WTC. In the analysis presented in this manuscript, as previously explained, the GPD+ WTCs are compared with all MWR-derived WTCs with a value within the WTC range. In other words, this analysis compares pairs of GPD+ and MWR-derived WTCs, therefore we do not have the figures requested by the reviewer. However, since GPD+ computes a WTC estimate for those points with an MWR-derived WTCs out of the WTC range and with missing MWR-derived WTCs, which otherwise wouldn't be used in the computation of the SLA dataset, it seems legitimate to state that GPD+ leads to a larger set of SLA measurements. Besides this, we should not ignore the fact that MWR-derived WTCs in the closest 30 km to the coast are generally set as invalid, due to land contamination (cf. Figure 10), and are not available to compute the SLA in the coastal regions. Since a GPD+ WTC estimate is computed for each one of these points, provided observations are available, the number of points with a valid WTC to compute the SLA increase.

Figure 4:
• in my printout I can not discern the light and dark green colors

R.: The authors do not understand why the Reviewer was not able to discern the light and dark green colours in Figure 4. We have downloaded the Rev1 PDF documents from ESSD page, uploaded by us, and confirmed that both colours are discernible. Points in light green are mostly concentrated around continental coasts and, over ocean, around islands, while points in dark green, representing points with MWR-derived WTC unavailability (for which the "fill value" is given), occur over the entire passes shown in the figure. Maybe a different PDF file has been provided to the Reviewer?

Figures 7/8:
• figure caption is confusing

R.: The caption of Figure 7 has been rewritten as:
"Figure 7 Left: Along-track WTCs (m) present in the Envisat FMR V3.0 products for Envisat Cycle 12, passes 101 and 58 (panels (a) and (b), respectively): GPD+ (black), MWR-derived WTC (red) and ERA Interim WTC (blue). Panel (a) shows an example of the unavailability of the onboard MWR-derived WTC; panel (b) shows contamination by ice and rain (red points around the Equator) in the MWR-derived WTC. Right: Geographical coverage of the Envisat tracks shown in the left panels (longitude is given in the 0°-360° range to show the entire track). Along-track points with a GPD+ estimate are shown in red, while points where the GPD+ kept the MWR-derived WTC are shown in black."
The sentence "The improvement in the description of the WTC signal in terms of small spatial scales when compared to the ERA Interim WTC (in blue) is clear in the panel (a) of Fig. 7 (e.g. around latitudes 10°S-10°N).", from the previous caption, has been moved to the manuscript (Section 3.1) to simplify the figure's caption.

• in my printout I can not discern the black and blue curves

R.: Once again, the authors do not understand why the Reviewer was not able to discern the blue and black curves in Figures 7 and 8. Both curves are discernible in the Rev1 PDF documents that have been uploaded by the authors and are currently available in the ESSD system. There could be, however, a superposition of both curves when the GPD+ and NWM WTCs are equal, but this situation seldom occurs in the provided plots and both curves are generally different in the cases presented in the figures.

Figure 11a:

for the GPD+ correction there is a pronounced decrease of along-track SLA variance from the end of 2007 onwards. This seems to coincide with the number of available GNSS stations/observations. Is there a similar effect for the Jason-1 missions?

R.: From mid/late 2007 onwards, there is a pronounced decrease of along-track SLA variance with respect to the MWR that we believe is due to an increase in the number of MWR-derived WTC observations flagged as invalid/out of range in the Envisat products, instead of being due to an increase in the number of available GNSS stations. If the increase in the number of GNSS stations used by GPD+ was the reason for the decrease in the variance of the differences (GPD-MWR), a similar result would be seen in the comparison GPD-ERA and Figure 11(c) shows that this is not the case. According to the Envisat validation report for the year 2012 (cf. https://www.aviso.altimetry.fr/fileadmin/documents/calval/validation_report/EN/annual_report_en_2012.pdf), which provides results for the whole Envisat mission, despite the MWR on board Envisat exhibits nearly 100% of availability throughout the mission period, MWR corrections are missing in the products over certain mission periods due to data generation problems at ground segment level. Consequently, for these periods, the land/sea radiometer flag has been set to land over ocean points, meaning that MWR data are missing. The percentage of missing MWR WTCs over ocean increased in the referred period (cf. Figure 9 of this manuscript), which can also be seen in the figure extracted from the reference given above. A summary of the MWR unavailability periods for the whole Envisat mission is also available in the ENVISAT Microwave Radiometer Assessment Report for Cycle 112: https://earth.esa.int/sppa-reports/envisat/mwr/cyclic/2012-02-19/rapport_cycle_env_20120219.pdf

[Figure]

Figure 1 - Percentage of missing MWR WTCs over ocean (extracted from https://www.aviso.altimetry.fr/fileadmin/documents/calval/validation_report/EN/annual_report_en_2012.pdf).

Besides the unavailability of the MWR-derived WTC, which could be the result of the aging of the onboard radiometer (also pointed out in the MWR quality reports), we should also consider the loss of the altimeter S-band, in the beginning of 2008, that has led to an increase in the number of invalid SLA measurements.

Therefore, we believe that the decrease in the SLA variance is due to the globally poorer performance of both the MWR and the Envisat altimeter towards the end of the mission. Therefore, no similar results have been found for the Jason series.

The following text (in green) has been added to the text:

"Using the GPD+ WTC instead of the MWR-derived WTC (Fig. 11a) leads, in the along-track analysis, to an improvement in the variance of the oceanic signal of 0.3 $cm^2$ on average, increasing in the second half of the period where values of 2 $cm^2$ can be reached in some cycles, most probably due to the globally poorer performance of both the MWR and the altimeter towards the end of the mission."

Figure 12:
• the caption says: weighted SLA variance differences, what kind of weighting is applied?

R.: This weighting is applied to overcome the fact that the number of available along-track points increases substantially as the latitude increases. More information has been added where the expression "weighted variance" appears for the first time. These lines now read:
"Differences between each pair of SLA data sets are computed along track and at crossovers and the weighted variance estimated for the time span of the whole Envisat period, with latitude-dependent weights (i.e., weights are function of the co-sine of latitude)."

• the variance reduction between the 'MWR and the GPD+ correction is much more pronounced for the along-track SLAs than for the cross-over points. Wouldn't it be more convincing to show the patterns for the along-track SLAs? In addition, I would be interested to see where the peak in figure 13 originates from that is showing up around 50°N.

R.: We agree that the variance reduction between the GPD+ and MWR-derived WTCs is much more pronounced in the results for the along-track analysis than for those obtained for the cross-over analysis. Both diagnoses are complementary, since the variance analysis for crossover points allows the evaluation of SLA error for scales lower than 10 days, while the SLA variance analysis performed for along-track points includes all the temporal scales.
However, the variance reduction at crossover analysis is generally considered a better indicator of the improvement in the various terms used in the SLA computation, the WTC included, and thus we opted for showing the results for the crossover analysis.
The increase in the reduction of the SLA variance at latitudes around 50°N is associated with a better description of the WPD field in the coastal regions northwards of the regions where the western boundary currents flow (off Newfoundland and in the Sea of Okhotsk). This result, although more noticeable in the spatial plots derived from the along-track analysis and not shown in the paper, can also be seen in the results from the crossover analysis. The peak in Figure 13 originates from the occurrence of pixels with large negative variance GPD- MWR differences (dark blue colour) visible in Figure 11 at longitudes ~65°W-50°W (off Newfoundland) and 140°E-150°E (Sea of Okhotsk).
The following figure shows similar results as those provided in Figure 12 for crossovers but computed for along-track results. For both differences GPD-MWR and GPD-ERA, generally better results are obtained for the along-track analysis.

[Figure]

You might consider to add a link to the CTOH GNSS Product Handbook

R.: The authors believe that the best source for the GPD+ dataset is the archive available on the homepage of the Satellite Altimetry Group, University of Porto, publicly available at the repository provided in this manuscript, since this is the one that provides the most recent GPD+ reprocessing outputs. Moreover, the manuscript under review should constitute the most trustworthy and complete GPD+ product handbook.

**Summary of relevant changes included in the revised version (R2) of the manuscript**

The authors have carefully considered and answered the Reviewer's comments and worked to include most of these comments in the revised version of the manuscript. The main changes introduced in the manuscript can be summarised as:

- Table 2 now shows the formal errors associated with each calibration parameter.
- Some parts of the text were rewritten/completed to be clearer and more informative.
- Caption of Figure 7 has been made clearer. Consequently, one sentence of the former caption has been moved to Section 3.1.
- One sentence has been added at the end of Section 2.1.4.
- Minor typos have been corrected.

[revised manuscript text omitted]